# Illuminating phenotypic drug responses of sarcoma cells to kinase inhibitors by phosphoproteomics

Chien-Yun Lee [ID][1], Matthew The [ID][1], Chen Meng [ID][2], Florian P Bayer[1], Kerstin Putzker [ID][3], Julian Müller [ID][1], Johanna Streubel[1], Julia Woortman [ID][1], Amirhossein Sakhteman[1], Moritz Resch[1], Annika Schneider [ID][1], Stephanie Wilhelm [ID][1] & Bernhard Kuster [ID][1,2,4 ✉]

## Abstract

Kinase inhibitors (KIs) are important cancer drugs but often feature polypharmacology that is molecularly not understood. This disconnect is particularly apparent in cancer entities such as sarcomas for which the oncogenic drivers are often not clear. To investigate more systematically how the cellular proteotypes of sarcoma cells shape their response to molecularly targeted drugs, we profiled the proteomes and phosphoproteomes of 17 sarcoma cell lines and screened the same against 150 cancer drugs. The resulting 2550 phenotypic profiles revealed distinct drug responses and the cellular activity landscapes derived from deep (phospho) proteomes (9–10,000 proteins and 10–27,000 phosphorylation sites per cell line) enabled several lines of analysis. For instance, connecting the (phospho)proteomic data with drug responses revealed known and novel mechanisms of action (MoAs) of KIs and identified markers of drug sensitivity or resistance. All data is publicly accessible via an interactive web application that enables exploration of this rich molecular resource for a better understanding of active signalling pathways in sarcoma cells, identifying treatment response predictors and revealing novel MoA of clinical KIs.

**Keywords** Drug Response; Kinase Inhibitor; Phosphoproteomics; Proteomics; Sarcoma
**Subject Categories** Cancer; Pharmacology & Drug Discovery; Proteomics

## Introduction

Molecularly targeted drugs including many kinase inhibitors (KIs) have revolutionised the clinical management of several cancers. This advancement was enabled by a precise understanding of the oncogenic mechanism and the development of a bespoke molecule that addresses it. However, this simple paradigm is challenged by the realisation that the molecular composition of cancer cells is highly heterogeneous and dynamically changes over time. In addition, most KIs show polypharmacology that is influenced by the cellular context the drug operates in. Therefore, understanding the mode of action (MoA) of drugs in a particular cellular system is essential for its successful use as a medicine (Davis, 2020; Schenone et al, 2013; Tulloch et al, 2018).

One way to address this challenge in a systematic fashion is to record the genotypes, transcriptional activity and protein expression signatures (proteotypes) (Aebersold and Mann, 2016) of cancer cell lines at baseline and to correlate these signatures with drug phenotypic responses. These correlations may reveal direct relationships between genes/proteins and the actions of small molecules, which may serve as molecular markers for drug sensitivity or resistance. Much work in this direction has been performed at the level of genomic mutations and measuring the transcriptomes of cancer cells (Barretina et al, 2012; Garnett et al, 2012; Rees et al, 2016; Seashore-Ludlow et al, 2015). More recently, these investigations have been extended to the level of the proteome. To date, the proteotypes of >1000 cancer cell lines have been recorded and correlated with drug sensitivity data (Frejno et al, 2020; Frejno et al, 2017; Gholami et al, 2013; Lawrence et al, 2015). This approach has proven useful in several cases by providing insights into specific cancer biology and illuminating the cellular MoAs of small molecules. Many of these datasets have also become important publicly available resources for cancer research (Barretina et al, 2012; Corsello et al, 2020; Frejno et al, 2017; Garnett et al, 2012; Gholami et al, 2013; Goncalves et al, 2022; Nusinow et al, 2020; Rees et al, 2016; Reinhold et al, 2012; Roumeliotis et al, 2017; Seashore-Ludlow et al, 2015; Tsherniak et al, 2017).

Despite the above advances, the relationships between the levels of protein post-translational modifications (PTM) and drug responses have only received limited attention (Frejno et al, 2020; Gosline et al, 2022). This is somewhat surprising as decades of cancer research has established that the activity of many cancer pathways is regulated by PTMs, most notably by protein phosphorylation. Consequently, KIs have become successful cancer drugs and, to date, >70 are approved for use in humans and certain other diseases (Cohen et al, 2021). That said, the full range polypharmacologies of KIs is only beginning to emerge

[1]Chair of Proteomics and Bioanalytics, Technical University of Munich, Freising, Germany. [2]Bavarian Biomolecular Mass Spectrometry Center (BayBioMS), Technical University of Munich, Freising, Germany. [3]Chemical Biology Core Facility, EMBL Heidelberg, Heidelberg, Germany. [4]German Cancer Consortium (DKTK), partner site Munich and German Cancer Research Center (DKFZ), Heidelberg, Germany. ✉E-mail: kuster@tum.de

(Klaeger et al, 2017) and their impact on the phosphoproteome is not systematically understood (Gosline et al, 2022; Kauko et al, 2020; van Alphen et al, 2020).

Most of the aforementioned studies have focussed on profiling diverse sets of cancer cell lines covering major cancer entities for which many cell line models are available. In contrast, rare cancers are often under-represented even though the unmet medical need is just as high. Sarcomas fall within this category. They are not only rare (~1% of all human cancers; ~15% of all paediatric cases) and lack effective therapeutic options (Grunewald et al, 2020), but also often exhibit a low mutational burden (Nacev et al, 2022). Genomic profiling of sarcoma cells or tissues can be beneficial for the diagnosis and classification of certain subtypes, but information about actionable genomic alterations is often limited (Gounder et al, 2022; Grunewald et al, 2020; Nacev et al, 2022). This may in part be because sarcomas are extremely diverse and more than 70 histological subtypes have been identified to date (Gounder et al, 2022; von Mehren et al, 2020). For example, in a recent genomic profiling study of 2138 sarcoma cases, only one-third contained at least one actionable alteration (Nacev et al, 2022). A few kinases have been identified as drivers of cell growth in sarcoma cells (Bai et al, 2012) and four KIs have been approved for the treatment of sarcomas (Imatinib, Pazopanib, Sunitinib and Avapritinib) mostly because on the basis of inhibiting the tyrosine kinases KIT and PDGFR. Several further KIs are in clinical evaluation (Wilding et al, 2019), implying a considerable potential for repurposing clinically approved KIs for treating sarcoma.

In this work, we systematically asked the question which phosphorylation-regulated signalling pathways may operate in sarcoma cells and which clinically approved KIs may, therefore, be repurposed for the treatment of sarcomas. To do so, a drug screen comprising 150 clinical drugs (including 139 KIs) in 17 diverse sarcoma cell lines was performed and complemented with recording their corresponding (treatment-naive) baseline proteomes and phosphoproteomes. Integration of the phenotypic and proteomic data identified, for instance, several KIs that are effective in most sarcoma lines. By computing the activity landscapes of the cell lines (Frejno et al, 2020), activated kinases and pathways could be identified. In addition, the analysis highlighted a group of proteins and phosphorylation sites that may be used as markers for sensitivity or resistance to a single KI or a group of KIs sharing the same targets. As only few examples can be discussed in the current manuscript, all data is accessible via an interactive web interface in ProteomicsDB (Lautenbacher et al, 2022; Samaras et al, 2020; Schmidt et al, 2018) (https://www.proteomicsdb.org/sarquarium/). The data, analysis and computational tools provide a valuable resource for further investigations into how the cellular proteotype shapes cellular phenotype of drug responses.

# Results

## Selection of sarcoma cell lines and clinical drugs

The selection of cell lines for this study was guided by three major considerations: (1) They already showed response in prior phenotypic drug screens. Here, we relied on published literature in which 63 human sarcoma lines were screened against 100 FDA-approved oncology agents, 345 investigational agents and including 52 KIs (Teicher et al, 2015). (2) To ensure the feasibility of the current work, cell lines were

prioritised for complementarity in drug response towards the above 52 KIs assuming that phenotypic response diversity would be reflected by the molecular diversity of the same cell lines. (3) All cell lines are commercially available and a detailed description of the rational and criteria for cell line selection can be found in "Methods". Further information on the resulting 17 cell lines representing 12 diverse histological subtypes can be found in Dataset EV1. Four subtypes are represented by two cell lines because they showed very distinct drug responses to KIs even though they are classified as the same subtype. The choice of compounds (total of 150, Dataset EV2) focused on KIs because of the focus of the current study on the phosphoproteome. The screening deck contained all 71 approved KIs (at the time of writing) and 52 phase III drugs. These 123 KIs are particularly valuable as they may be repurposed and/or recommended for treatment in a compassionate use setting. Further, 16 phase I, II, or II/III KIs were included to cover complementary target proteins (Klaeger et al, 2017). In addition, four chemo drugs and one epigenetic drug already used for treating sarcomas today were included, as well as six non-KIs used for treating other types of cancer.

## Datasets

Three systematic datasets were generated (Fig. 1; see "Methods" for details). First, all cell lines were phenotypically screened in a dose-dependent fashion (11 doses plus vehicle control) against all cancer drugs (in triplicate), generating 2550 dose–response profiles with excellent reproducibility (median coefficient of variation, CV of 1.74%; Dataset EV3; Fig. EV1A). The quantitative data (EC50) correlated reasonably well with publicly available drug response data (Teicher et al, 2015) for the 16 cell lines and 55 compounds that are common between the two ($R = 0.4$–$0.7$; Appendix Fig. S1). Second, baseline (i.e., treatment-naive) full proteomes were recorded for the same 17 cell lines quantifying 9200–10,200 protein groups per cell line (Fig. EV1B; Dataset EV4). Five of these lines had also been profiled by the CCLE consortium (Nusinow et al, 2020) and the relative protein quantities determined in their study (based on TMT reporter ion intensity) correlated well with the ones determined here (based on MS1 intensity of TMT-labelled peptides; $R = 0.53$–$0.66$; Appendix Fig. S2). The proteomes of the four cell lines, G401, RD, RD-ES and SW1353 cells were measured in three independent biological replicates which reproduced and correlated well (>90% of all proteins reproduced with CVs below twofold; Pearson correlations between $R = 0.89$–$0.92$, Fig. EV1C; Dataset EV4; Appendix Fig. S3). Third, baseline (i.e., treatment-naive) phosphoproteomes were recorded for the same 17 cell lines quantifying 10,200–27,200 phosphorylation sites (p-sites) per cell line (Fig. EV1B; Dataset EV5). Again, the phosphoproteomes of the above four cell lines were measured in three biological replicates and 60–90% of all phosphopeptides reproduced with CVs below twofold and showed correlated coefficients of between $R = 0.68$- and 0.93 (Fig. EV1C; Dataset EV5; Appendix Fig. S4). These datasets formed the basis for all further analysis and only such cases are highlighted or discussed in the manuscript for which observed effects were substantially larger than the technical variation (i.e. >0.3 ΔAUC for drug response; >tenfold p-peptide or protein abundance; minimum of detection in eight cell lines). When exploring the data in ProteomicsDB (see below), we advise readers to be cautious when investigating abundance or drug response differences between cell lines that are close to the determined

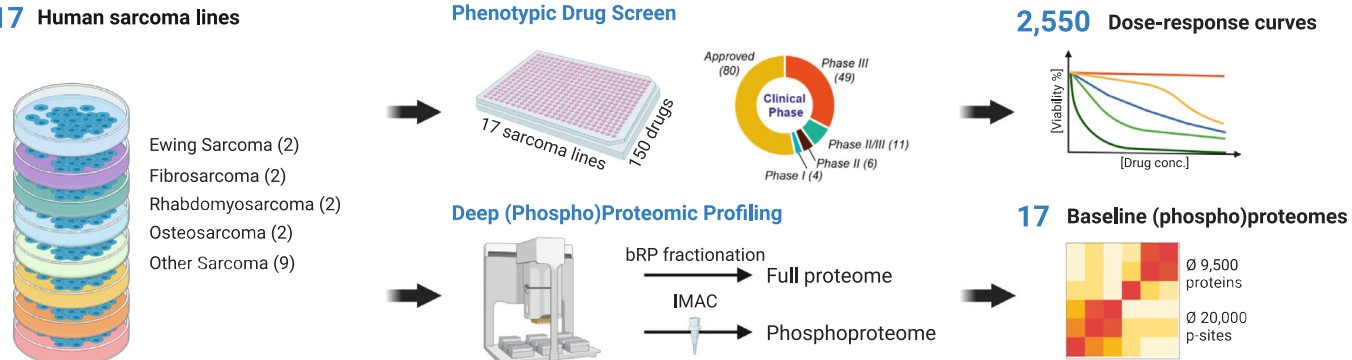

**Figure 1. Schematic representation of the experimental and data analysis workflows.**

Seventeen sarcoma cell lines representing 11 histological subtypes were used as models to investigate the relationships between their proteotypes and drug phenotypes. Three datasets were generated for this cell line panel: First, response (viability) to 150 cancer drugs (grouped by state of clinical evaluation). Second, baseline protein expression (following lysis, tryptic digestion and peptide off-line fractionation by high pH reversed-phase liquid chromatography and online LC–MS/MS analysis. Third, baseline phosphorylation following phosphopeptide enrichment by immobilised metal affinity chromatography and online LC–MS/MS analysis. p-site phosphorylation site.

technical reproducibility of the proteomic and drug screening workflows.

## Sarcoma cell lines exhibit distinct responses to KIs

All dose–response viability curves from the phenotypic drug screen can be viewed in a section of ProteomicsDB called "Sarquarium" (https://www.proteomicsdb.org/sarquarium/). This study very substantially increased the content of the database for drug viability data and is a valuable resource for the community. To explore similarities and differences in drug responses, the phenotypic data were clustered (unsupervised) by the area under the curve (AUC; Fig. 2A; Dataset EV3). It is apparent that many drugs did not substantially inhibit the growth of sarcoma lines (Fig. 2A, red areas). This may be expected because many of the KIs target specific kinases and signalling pathways, e.g., EGFR or HER2, that may not be oncogenic drivers in these cells. This specific case is supported by e.g., public datasets (Barretina et al, 2012; McDonald et al, 2017) for nine of the sarcoma lines used here, for which no oncogenic driver alteration has been found for EGFR or HER2 (Dataset EV1).

In contrast, drugs targeting cellular mechanisms of general importance for cell survival, such as DNA replication, transcription and the cell cycle, generally potently inhibited the proliferation of most cell lines (blue areas). These drugs include the cell cycle inhibitors Trabectedin (a natural product that covalently modifies guanine bases of DNA and is a first-line chemodrug for the treatment of sarcoma), Dinaciclib (a multi-CDK inhibitor, blocking the cell cycle), Volasertib (a PLK1 inhibitor interfering with mitosis) as well as a series of multi-kinase inhibitors (Fig. 2A,B). Volasertib is an interesting case because it recently received orphan drug designation by the US Food and Drug Administration (FDA) for the treatment of rhabdomyosarcoma (FDA-Orphan-Drug-Designations-and-Approvals, 2020). PLK1 serves essential functions throughout the M phase of the cell cycle, including regulation of mitotic exit and cytokinesis. Previous studies suggested that PLK1 inhibition by Volasertib promotes the degradation of PAX3-FOXO1, a fusion protein frequently found in rhabdomyosarcoma

patients, thereby inhibiting cancer growth in PDX models (Abbou et al, 2016). Our data showing that Volasertib is effective across many sarcoma lines may provide a rational for additional clinical trials in other sarcoma subtypes. The designated PI3K inhibitor Copanlisib also showed high potency against most of the sarcoma lines (Fig. 2A, red label on the right). However, most of the other PI3K inhibitors in the screen did not show substantial effects, implying that yet unknown off-target(s) of Copanlisib may be responsible for its inhibitory effect on sarcoma cells (Fig. EV2A).

Other clusters of drugs in Fig. 2A represent more differentiated responses of cell lines. One such cluster groups the receptor tyrosine kinases (RTKs) VEGFR, PDGFRB and FGFR. Other clusters contain drugs targeting PI3K, MAP2K, or mTOR, indicating that these cell lines share molecular characteristics that make them responsive to these drugs (Fig. 2A). Surprisingly, Pazopanib, an approved multi-kinase inhibitor for the treatment of sarcomas, was not very effective in any of the sarcoma lines contained in this panel (Fig. 2C). In contrast, selected cell lines showed full, medium or no response to Trametinib (a MAP2K inhibitor). This heterogeneity of response was observed for MAP2K inhibitors in most of the cell lines. When filtering the data for EC50 values of <100 nM, AUCs of <0.9 and relative inhibition effect >50% to focus on the most potent drugs, it became apparent that most cell lines responded to 3–10 KIs and that A204 cells (a rhabdoid line) were susceptible to KIs (Figs. 2D and EV2B; Dataset EV3). These cells responded to several RTK inhibitors, multi-kinase inhibitors and mTOR inhibitors, implying high activity of several oncogenic signalling axes in these cells. While the above data and analysis shows that phenotypic screens can be powerful for identifying advanced clinical compounds with repurposing potential, the reasons why certain cell lines do or do not respond to these agents often remain elusive.

## Molecular activity landscapes explain drug responses and suggest effective drugs

To better understand the above drug responses at the molecular level, the phenotypic drug profiles were integrated with the

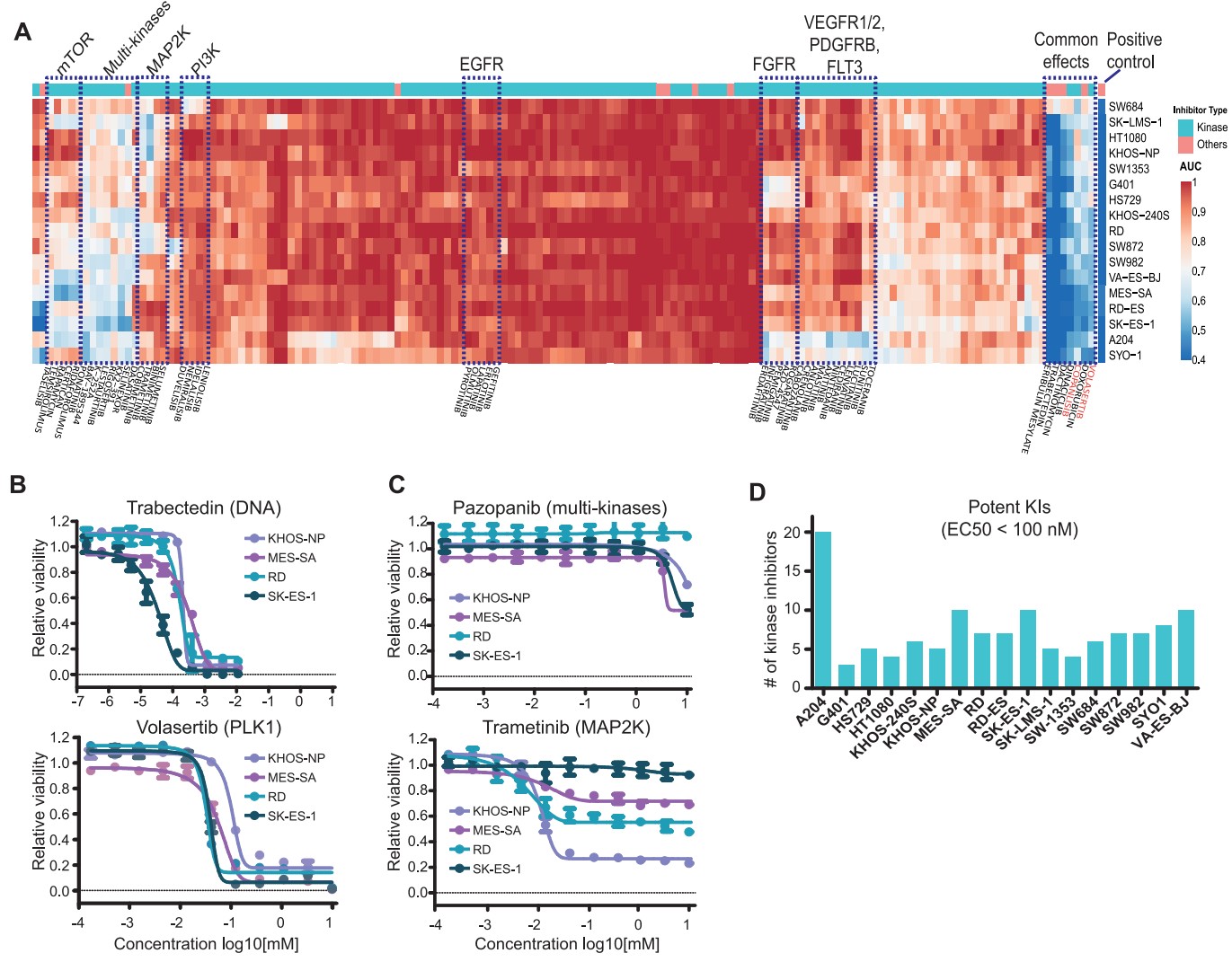

**Figure 2. Phenotypic drug screening revealed common and distinct drug responses of sarcoma cell lines.**

(A) Heatmap summarising the responses of 17 sarcoma lines (rows) to 150 clinical cancer drugs (columns). Dotted line boxes highlight clusters of drugs representing common targets or effects. AUC area under the curve. (B, C) Examples for cell viability dose–response curves from panel (A). Measurements were in technical triplicates and error bars denote the standard deviation (SD). (D) Numbers of kinase inhibitors that affect the viability of cell lines with EC50 values of smaller than 100 nM (AUCs <0.9 and relative inhibition effect >50%). Source data are available online for this figure.

proteomes and phosphoproteomes of treatment-naive (termed baseline from here on) cell lines (Datasets EV4 and EV5). The simple rationale for this analysis is that drugs targeting specific kinases in certain pathways would be expected to be more effective in cell lines driven by the presumably elevated activity of these kinases and pathways at baseline. To test this hypothesis, we calculated an activity score for each kinase in each cell line by combining kinase abundance (from full proteomes; mandatory criterion), with at least one of the following three criteria: kinase phosphorylation abundance, kinase activation loop abundance and kinase-substrate phosphorylation abundance (all from phospho-proteomes; see "Methods" for details). A similar strategy was developed in a previous study by the authors (Frejno et al, 2020), and we extended it here to include kinase activation loop

phosphorylation as reported by others for the INKA score (Beekhof et al, 2019). This way, we computed activity scores for 383 kinases across all cell lines. This information can be visualised as a heatmap representing the activity landscapes of the cell lines (Fig. 3A; Dataset EV6). It is apparent that the molecular heterogeneity of the different cell lines is reflected in their activity landscapes. While some cell lines exhibited high scores for many kinases, e.g., VA-ES-BJ (epithelioid sarcoma), G401 (kidney rhabdoid tumour), KHOS-NP (osteosarcoma), the Ewing sarcoma line RD-ES, appeared comparatively "silent" in kinase activity. Interestingly, RD-ES is the only cell line in this study showing mixed adherent and suspension characteristics in culture. It also has by far the lowest number of phosphorylation sites identified among all sarcoma lines (Fig. EV1B). This may suggest that RD-ES

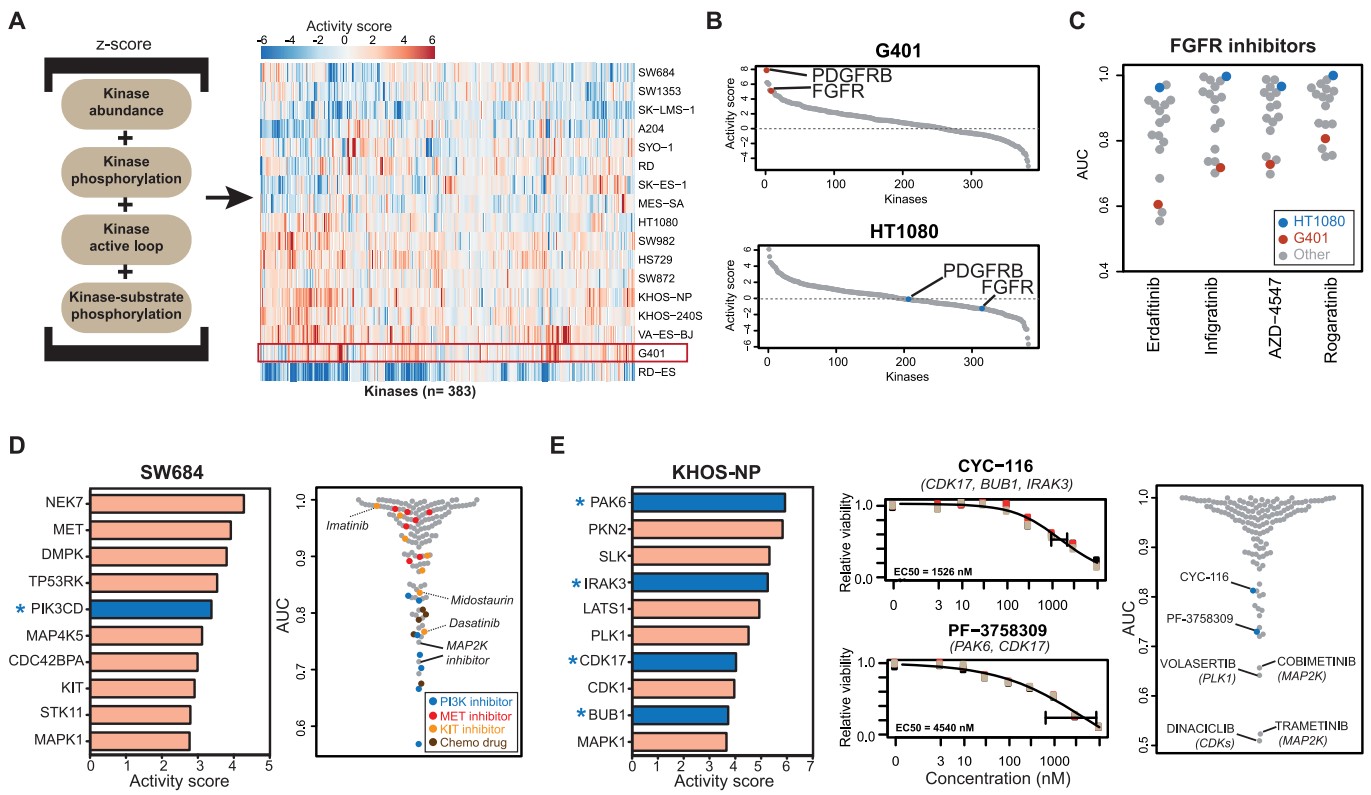

**Figure 3. Connecting molecular activity landscapes and drug responses.**

(A) Activity landscapes of sarcoma lines (right panel) computed from the proteomic data based on the intensities of the molecular features shown in the left panel (see "Methods" for details). (B) Distribution of activity scores of protein in G401 and HT1080 cells showing differences in activity scores for FGFR and PDGFRB. (C) Relative viability (AUC) of sarcoma cell lines in response to four FGFR inhibitors. (D) Left panel: list of kinases ranked by activity score in SW684 cells. The asterisk denotes a kinase that is a target of the most potent kinase inhibitor. Right panel: relative sensitivity of SW684 cells to all drugs in the screen. (E) Left panel: same as panel (D) but for KHOS-NP cells. Middle panel: dose–response curves for the phase I/II inhibitors CYC-116 and PF-3758309 (the gene names of the targets of the drugs are indicated in parenthesis). Measurements were in technical triplicates and error bars denote the standard deviation (SD). Right panel: relative sensitivity of KHOS-NP cells to all drugs in the screen. Source data are available online for this figure.

cells are not strongly driven by kinase-regulated signalling pathways.

Ranking kinases by their activity scores enabled forming hypotheses for each cell line as to which drug may be able to inhibit its proliferation. These hypotheses could be experimentally substantiated in many cases (Figs. 3B and EV3). For example, PDFGRB and FGFR scored highly in G401 cells (a rhabdoid tumour line), suggesting that these two kinases are essential for cell growth (Fig. 3B). In contrast, activity scores of the same two kinases were low in HT1080 cells (a fibrosarcoma line), suggesting that the proteins may not be essential for proliferation. Examination of the phenotypic drug response data of four FGFR and 11 PDGFR inhibitors indeed confirmed that G401 cells are among the most sensitive, whereas HT1080 did not respond to these drugs (Figs. 3C and EV3A). This result aligns well with previous work showing that co-inhibition of these two kinases leads to growth inhibition of rhabdoid tumour lines (Wong et al, 2016). A further example is shown in Fig. EV3B for MAP2K inhibitors. Here, RD-ES cells had very low activity scores for MAP2K (a kinase that directly phosphorylates MAPK1 and MAPK3) and these cells, in contrast to most others, also showed no response to MAP2K inhibitors.

When breaking down the kinase activity score into protein abundance and substrate phosphorylation, the value of the phosphoproteomics information became apparent in many cases. For example, AURKA abundance was not particularly high in the Ewing sarcoma line SK-ES-1, but AURKA activity was high, as evidenced by the high ranks of AURKA substrates (Fig. EV3C). In line with this hypothesis, SK-ES-1 cells were the most sensitive to the AURKA inhibitors Alisertib and Barasertib. Similarly, DDR2 showed high activity by substrate phosphorylation levels but not by protein abundance in the rhabdoid line A204 (Fig. EV3D) (Hinson et al, 2013). Even though there are no designated DDR2 inhibitors in the compound library used here, several KIs have DDR2 as a potent off-target (Klaeger et al, 2017). Consequently, A204 cells responded much stronger to potent DDR2 inhibitors (Nintedanib, Dasatinib, Ponatinib, Lucitanib) than most other KIs (Fig. EV3D).

Taking this concept a step further suggests that the activity scores of kinases in cell lines would enable the rational choice of an effective drug for this cell line. To test this idea, we first focussed the analysis on kinases that are targets of at least one of the (clinical) compounds in our screen (Klaeger et al, 2017;

Samaras et al, 2020). Subsequently, these kinases were ranked by their activity scores. For instance, in SW684, a fibrosarcoma line that showed little response towards most of the drugs used in this study (Fig. EV2B), PI3K and MAPK1 were among the ten highest kinase activity scores (Fig. 3D). Indeed, PI3K inhibitors and drugs targeting MAP2K (the direct upstream kinase of MAPK1) were among the most effective drugs for this cell line. Conversely, the computed activity scores for CDK1 and PKN1 were very low (Fig. EV3E) and, consequently, Dabrafenib (inhibitor of BRAF and CDK1) and Tofacitinib (PKN1) were not potent in reducing the viability of SW684 cells. Dinaciclib (a multi-CDK inhibitor) also only had a small phenotypic effect on SW684 cells, In fact, the computed CDK1 activity in SW684 cells was the lowest of all cell lines and SW684 cells were the least Dinaciclib-responsive of all 17 sarcoma lines (Fig. EV3E, right panel).

Another example is the osteosarcoma line KHOS-NP that also showed minimal response to most KIs in the drug screen. Still, the top ten list of active kinases enabled choosing the phase I compounds CYC-116 and PF-3758309 as likely effective drugs on the basis of their ability to inhibit PAK6, CDK17, BUB1 and IRAK3 (Fig. 3E). The latter information was provided by ProteomicsDB (https://www.proteomicsdb.org/analytics/selectivity) which contains target information for a large number of clinical KIs (Klaeger et al, 2017). Albeit not very potent in absolute terms, these two were among the best compounds for inhibiting the growth of KHOS-NP cells. In addition, this cell line responded well to the MAP2K inhibitors Trametinib and cobimetinib as well as Dinaciclib (a multi-CDK inhibitor) and Volasertib (a PLK1 inhibitor). This is in line with the fact that MAPK1, CDK1, CDK17 and PLK1 also have high-activity scores in this cell line. The above results demonstrate that the activity landscapes of kinases computed from the proteomes and phosphoproteomes of cell lines can, in many cases, either explain drug sensitivity toward certain inhibitors or provide a rationale for choosing a particular KI to address the kinase activity driving the growth of sarcoma lines.

It is important to note that a high or low kinase activity score computed from the proteome and phosphoproteome data may not necessarily be relevant for the phenotype measured in the drug screen (here cell viability) as these kinases may serve other important functions in cells. Two examples are shown in Fig. 3D where e.g., MET and KIT appear to be highly active in SW684 cells. Yet, none of the MET inhibitors used has an effect on cell viability. The same is true for KIT inhibitors, while the reason why Dasatinib and Midostaurin show some effect on cell viability is because both drugs target many other kinases.

### Proteome expression profiles identify drugs acting at preferred steps of the cell cycle

Exemplified by the cell cycle, we asked if the proteome expression data can be used to identify drugs that act preferentially in a particular cellular process. To explore this, we estimated the proportion of cells in a certain cell cycle phase form the bulk proteome based on the differential expression of proteins across the cell cycle reported in previous studies (Kelly et al, 2022; Ly et al, 2017). Then, we correlated the expression of these 119 "periodic proteins" with the response to 87 drugs (minimum AUC <0.8) (Fig. EV3F). For 10 out of 14 drugs (with Pearson correlation <−0.5), the estimated proportion of cells in a particular stage of the cell

cycle agreed the one that the respective drug has been reported to arrest cells (Dataset EV7). Examples include Dinaciclib, a CDK2,5,9 inhibitor that blocks cells in S-phase and is most negatively correlated in Cluster 2. The same was observed for Doxorubicin, an inhibitor of DNA replication that happens in S-phase (Cluster 2). The correlation for Dasatinib was strongest in Cluster 4 (early M phase) but given the many targets of this drug, it is difficult to attribute this effect to a particular kinase. In cluster 5, we observed a large number of drugs with positive correlations (resistance) (Dataset EV7). Interestingly, eight of these are PI3K/mTOR inhibitors, indicating that cells in M phase are particularly resistant to inhibition of these pathways.

### Drugs with related MoA share markers of drug response

The results of the phenotypic drug screen showed that the cellular responses towards inhibitors that target the same kinases were relatively similar across the sarcoma lines (Fig. 2A). This indicates that the cellular MoAs of these drugs in the sensitive cell lines would also be similar. Therefore, we reasoned that such similar responses might be explained by the abundance of the same (group of) proteins or p-sites involved in the pathways targeted by the drugs. If so, these proteins or p-sites may be used as markers of sensitivity (or resistance) for a particular drug or a group of related drugs. To explore this systematically, we used an extended sparse multiblock partial least-square regression (SMBPLSR) algorithm (Frejno et al, 2020). SMBPLSR clusters drugs showing similar responses across cell lines and correlates the abundance of multiple proteins and p-sites with the responses toward these drugs. Twelve such clusters of drugs were found (Dataset EV8), and three are illustrated in Fig. 4. The first cluster contains four MAP2K inhibitors and markers for drug sensitivity, including proteins and p-sites known to be members of the MAPK pathway, which validates the approach (Figs. 4A, left panel and EV4A). These include the expression of SPRY2 and S100A16, as well as two p-sites on GIGYF2 (middle panel) (Hanafusa et al, 2002; Higashi et al, 2010; Zhu et al, 2016). Figure 4A also shows that the higher the abundance of the calcium-binding protein S100A16 is in a cell, the higher the sensitivity to the MAP2K inhibitor Cobimetinib.

The second cluster contains six PI3K and mTOR inhibitors (Fig. 4B). Among the sensitivity markers is phosphorylation on T33 of RRM2 (ribonucleotide reductase family member 2; Fig. EV4B). RRM2 catalyses the conversion of ribonucleotides (NDPs) to deoxyribonucleotides (dNDPs), the essential building blocks for DNA synthesis. Phosphorylation of T33 of RRM2 has been shown to promote the degradation of RRM2 protein to maintain genome integrity and DNA repair in G2/M phase (D'Angiolella et al, 2012). A previous study has also shown that mTORC1 is a determinant for G2/M checkpoint recovery, and inhibition of the mTORC1 pathway delays mitotic entry after DNA damage (Hsieh et al, 2018). Therefore, we hypothesise that high phosphorylation of RRM2 T33 may be a negative regulator of activated mTOR signalling. The strongest resistance marker for the same drugs is phosphorylation of T394 of Lamin A (LMNA) (Fig. 4B). Lamin A is an intermediate filament protein supporting the nuclear envelope and phosphorylation of Lamin A has been found throughout interphase and mitosis (Torvaldson et al, 2015). However, specific phosphorylation of T394 has not been detected during interphase (Kochin et al, 2014). Several G2/M marker proteins such as

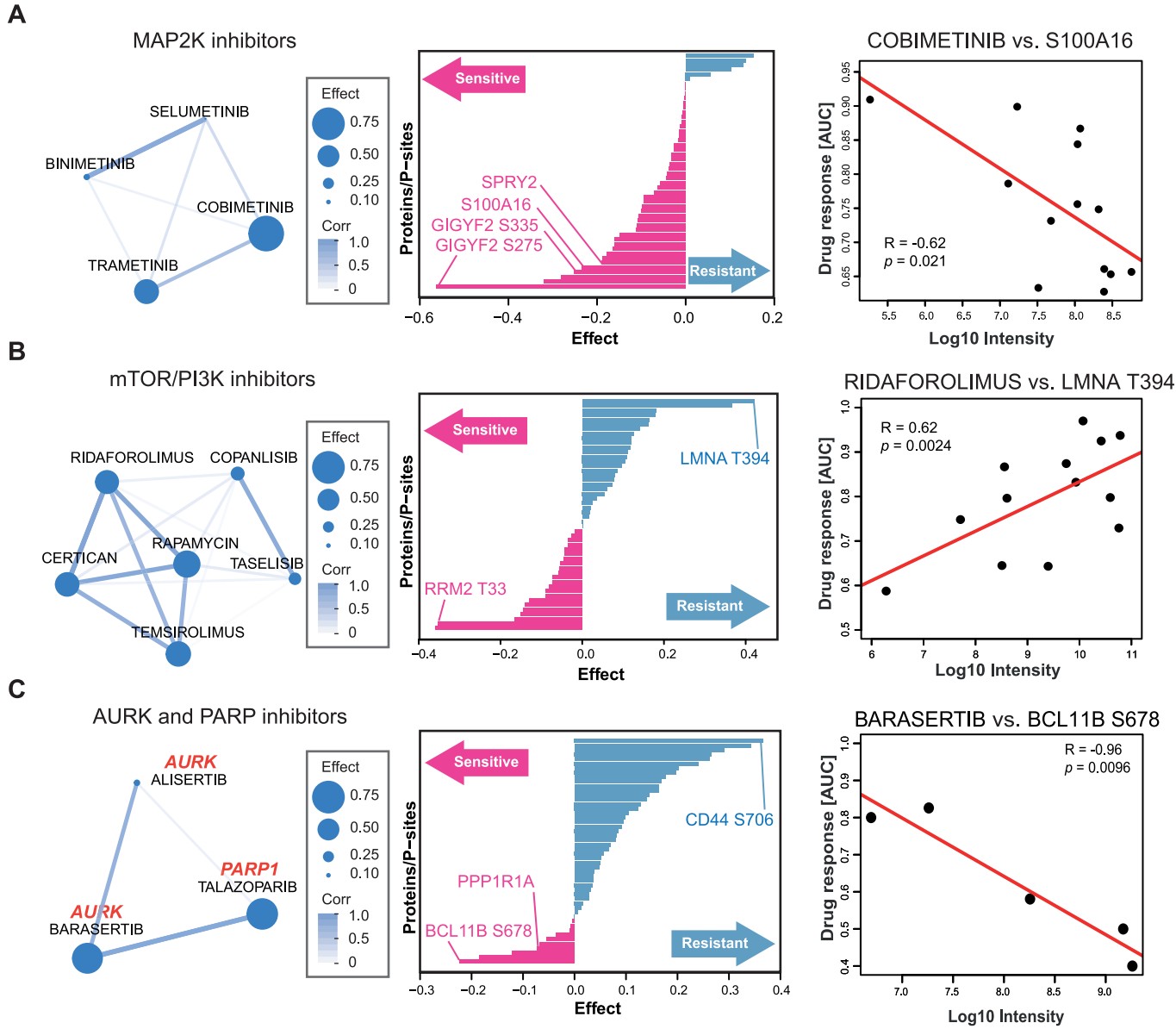

**Figure 4. Drugs with related MoAs share common markers of drug response.**

(A) Left panel: cluster of drugs resulting from multiblock partial least-square regression (SMBPLSR) analysis containing MAP2K inhibitors. Middle panel: ranked list of proteins and phosphorylation sites from the SMBPLSR analysis as indicators of drug sensitivity (pink) or resistance (blue). Right panel: correlation between the extent of phenotypic response towards Cobimetinib and the abundance of S100A16. R Pearson correlation coefficient, p P value. (B) same as (A) but for mTOR/PI3K inhibitors. (C) Same as (A) but for AURK and PARP inhibitors.

AURKB, TOP2A/B and CDCA8 showed statistically significant associations with response to the mTOR inhibitor Ridaforolimus (Appendix Fig. S5). Mining a recent large-scale kinase-substrate relationship study (Johnson et al, 2023) suggested that BUB1, a mitotic checkpoint serine/threonine-protein kinase, EEF2K, AAK1, MST1, or PASK may be upstream kinases of T394. However, none of these kinases correlated statistically significantly with T394 abundance or phenotypic response to mTOR inhibitors in terms of kinase abundance or computed activity. Hence, it remains unclear exactly how Lamin A phosphorylation links to G2/M of the cell cycle and resistance to mTOR inhibition.

The third cluster contains AURK and PARP inhibitors (Fig. 4C). PARP inhibitors are approved drugs for treating ovarian and fallopian tube cancer (Ashworth and Lord, 2018). In earlier work, it has been observed that the amplification of AURKA was associated with sensitivity to PARP inhibitors in 39 ovarian cancer cell lines (Ihnen et al, 2013), indicating an underlying mechanistic link between the AURK and PARP pathways. We identified phosphorylation at S678 on BCL11B as a sensitivity marker which was not observed at the protein level (Figs. 4C, right panel and EV4C). BCL11B is a zinc finger transcription factor up-regulated by the oncogenic fusion protein EWS/FLI, creating a constitutively active

transcription factor that is present in 85% of all Ewing sarcomas (Bailly et al, 1994). Interestingly, the expression of another sensitivity marker, PPP1R1A (protein phosphatase 1 regulatory subunit), has recently been reported to be induced by EWS/FLI in Ewing sarcoma lines as well (Fig. EV4D) (Orth et al, 2022). As the two Ewing sarcoma lines in our panel were susceptible to AURK and PARP inhibitors in our and prior screens (Fig. EV4E) (Garnett et al, 2012; Teicher et al, 2015), the data suggests that the sensitivity to AURK and PARP inhibitors in this cluster might be triggered by an EWS/FLI-regulated signalling pathway (Ihnen et al, 2013). Conversely, high phosphorylation of CD44 (a cell surface receptor with multiple functions) at S706, and much more so CD44 protein levels, were found to be a resistance marker (Fig. EV4F,G). CD44 expression levels were high in most sarcoma cell lines except for the two Ewing sarcoma lines (Fernandez-Tabanera et al, 2022; Teicher et al, 2015; Tsherniak et al, 2017). This observation suggests a negative association of CD44 abundance with active AURK and PARP signalling. However, the underlying mechanism has yet to be investigated.

As an interesting side note, we also identified a cluster containing Eribulin, Rigosertib, and KX2-391 (Dataset EV8). Eribulin is a clinically approved chemodrug for many types of cancer, including soft tissue sarcomas. It inhibits microtubule polymerisation, which blocks mitotic spindle formation, leading to mitotic arrest and cell death by apoptosis (van Vuuren et al, 2015). KX2-391 is a non-ATP competitive substrate-pocket-directed SRC inhibitor and inhibition of microtubule polymerisation has recently been identified as a second MoA (Smolinski et al, 2018). Rigosertib is a designated PLK1 inhibitor but previous studies have reported that an impurity in the preparation of Rigosertib also has anti-microtubule polymerisation effects (Baker et al, 2020; Jost et al, 2020). Hence, the reason that these drugs were grouped together, most likely results from their unintended effect on microtubule polymerisation.

## Correlating phenotypes and proteotypes identifies markers of drug response

To discover and prioritise proteins and p-sites associated with drug response, we applied Elastic net regression (Zou and Hastie, 2005) (see "Methods" for details) to 9.7 million associations resulting from 150 drugs, 12,000 proteins and 53,000 phosphosites. These can be explored in ProteomicsDB and analysis recapitulated many known and identified many potential new proteins and p-sites involved in the targeted pathways. One example shown in Fig. 5A is Cobimetinib, a MAP2K inhibitor used in combination with other drugs for the treatment of advanced melanoma (Boespflug and Thomas, 2016). Our analysis recapitulated numerous proteins and p-sites in the MAPK signalling pathway that act as sensitivity markers. This included MAP3K1 phosphorylation at S266, activation loop phosphorylation of MAPK1 at Y187, as well as downstream substrates such as phosphorylated ERRFI1, GYGFI, PXN, FOSL1 and CDC42EP3 (Cairns et al, 2018; Higashi et al, 2010; Maurus et al, 2017; Nagashima et al, 2015; Sinkala et al, 2021). The activity scores of MAPK1 in sarcoma lines also strongly correlated with drug sensitivity toward Cobimetinib, implying that resistant lines have low levels of intrinsic MAPK signalling activity (Fig. EV5A). Fewer potential resistance markers were identified compared to sensitive ones, including phosphorylated S669 of the

E3 ubiquitin ligase CBL. Interestingly, CBL protein levels did not correlate with response to Cobimetinib (Fig. EV5B). As CBL can terminate many tyrosine kinase receptor signalling pathways (Rubin et al, 2005; Tang et al, 2022), higher levels of CBL S669 phosphorylation may lead to activation of the ligase activity and, in turn to the reduction of MAPK signalling and rendering the cells less sensitive to MAP2K inhibitors. This hypothesis is supported by the strong negative correlation between the phosphorylation levels of MAPK1 active loop Y187 and CBL S669 (Fig. EV5C).

Another interesting example is presented by Infigratinib, an FGFR inhibitor approved for the treatment of advanced or metastatic cholangiocarcinoma patients carrying an FGFR2 fusion (Kang, 2021). As expected, several proteins and p-sites were identified as markers of sensitivity to Infigratinib (Fig. 5B). Notably, these included levels of several p-sites (but not protein levels) of FRS2, a key adaptor protein mediating FGFR signalling (Figs. 5B, left and EV5D). In contrast, SPRY2 is an antagonist of the FGFR pathway and phosphorylation at S115 and S139 were identified as resistance markers of Infigratinib because the activation of this protein results in a low level of FGFR signalling (Fig. EV5E). Among the less expected sensitivity markers were phosphorylation of ERBB2/HER2 at S1054, MAST2 (a serine/threonine-protein kinase) at S191, ULK2 (Serine/threonine-protein kinase) at S430 and TOP2B (DNA Topoisomerase) at S1552 (Figs. 5B,C and EV5F). The function of many of these p-sites is not known. However, the correlation between phosphorylation abundance and drug response implies some involvement in FGFR signalling in sarcoma cell lines. In fact, ULK2 has previously been shown to decrease the phosphorylation of FRS2 via binding to FRS2 and thus negatively regulates FGFR1 signalling (Avery et al, 2007). Therefore, it may be speculated that phosphorylation of S430 of ULK2 may interfere with its binding to FRS2, increasing the amount of free FRS2 and promoting FGFR signalling.

## FGFR inhibition by Infigratinib downregulates phosphorylation of ERBB2 and TOP2B

While the correlation and regression methods used above can be powerful, they do not necessarily imply causal relationships. To follow up on some of the observations made in the previous section, we measured the phosphoproteome of the FGFR2-overexpressing and FGFR2 inhibitor-sensitive rhabdoid sarcoma line G401 (Fig. 3B,C) in response to Infigratinib in a time-dependent fashion. More specifically, G401 cells were treated with 100 nM Infigratinib and samples were taken at nine time points ranging from 5 min to 24 h. DMSO controls were included at every time point to account for any (phospho)proteome changes due to cell growth during the experiment (Fig. EV6A). About 7900 proteins and 13,000 p-sites were quantified along the time course (Dataset EV9; Fig. EV6B). While regulation of protein abundance was not very pronounced after Infigratinib inhibition, substantial changes in phosphorylation levels were already observed after 10 min. Among these were FRS2 S428 and GAB1 T503 clearly indicating the down-regulation of FGFR signalling by the drug (Figs. 5D and EV6C). When looking up the candidate p-sites suggested by the elastic net regression above, ERBB2/HER2 S1054 and TOP2B S1552 were decreased after 10 min of Infigratinib treatment and remained down over the entire time course (Figs. 5E and EV6D). These results clearly show that

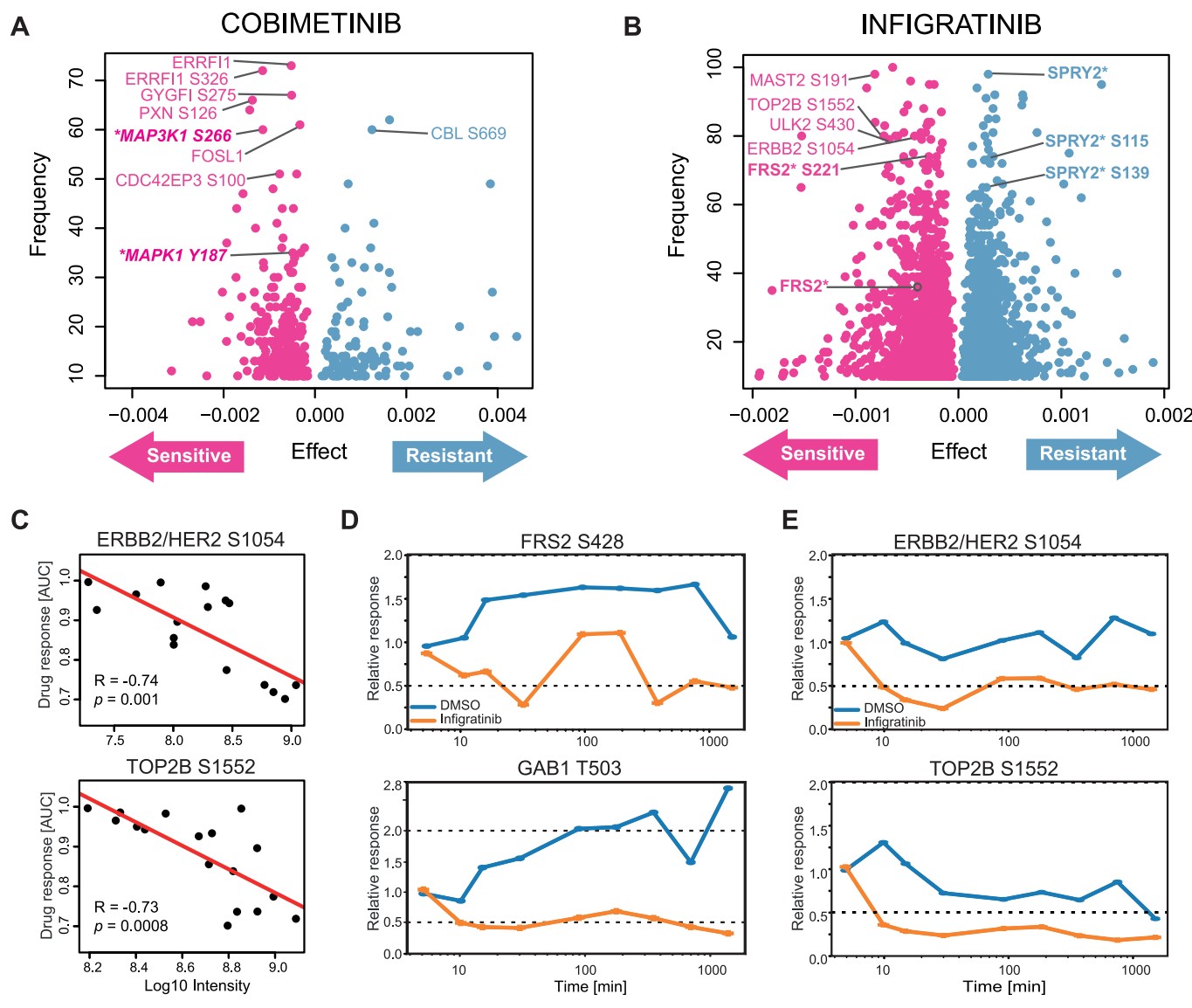

**Figure 5. Correlating phenotypes and proteotypes identifies markers of drug response.**

(A) Result of elastic net regression for the MAP2K inhibitor Cobimetinib highlighting proteins and phosphorylation sites indicative of drug resistance or sensitivity. (B) Same as panel (A) but for the FGFR inhibitor Infigratinib. (C) Correlation analysis between the level of phenotypic response of cell lines towards Infigratinib and the abundance of ERBB2 S1054 (top panel) and the abundance of TOP2B S1552 (bottom panel). (D) Time-dependent analysis of FRS2 S428 (top panel) and GAB1 T503 (bottom panel) in response to Infigratinib in G401 cells. Both phosphorylation sites are known members of FGFR signalling. (E) Same as panel (D) but for two novel phosphorylation sites ERBB2/HER2 S1054 (top panel) and TOP2B S1552 (bottom panel). Source data are available online for this figure.

the hypothesis created by elastic net regression can be experimentally confirmed.

The kinase responsible for the phosphorylation of TOP2B at S1552 is not known. Phosphorylation levels of this site not only correlated with Infigratinib response but also with other FGFR inhibitors (Erdafitinib and Rogaratinib), the multi-CDK inhibitor Dinaciclib and the DNA intercalator Dactinomycin (Fig. EV6E). This implies that the effects of all these drugs converge at the level of mitosis, a process in which TOP2B is intimately involved. Unlike S1130 and S1134 phosphorylation by CK1 on TOP2B, which promotes its degradation (Shu et al, 2020), TOP2B abundance did

not change in response to Infigratinib (Fig. EV6F). We again resorted to the aforementioned large-scale kinase-substrate relationship study to as which kinase may be responsible for TOP2B S1552 phosphorylation (Johnson et al, 2023). This analysis returned CDC7 and CK2 (both CK2A1 and CK2A2) as the strongest candidates but none of these showed a statistically significant correlation with computed kinase activity and TOP2B S1552 phosphorylation abundance. While this does not rule them out, the current data also does not provide supporting evidence.

Although crosstalk between different signalling cascades are not uncommon in cancer cells, a connection between ERBB2/HER2

and FGFR signalling in sarcoma cells has not yet been established and its functional consequences are not yet clear. The upstream kinase of ERBB2/HER2 S1054 is unknown, but the site is annotated to respond to EGF (PhosphositePlus). That said, it is unlikely that HER2 signalling is directly involved in tumour progression in G401 as none of the HER2 inhibitors in our screen inhibited the growth of G401 cells (Fig. EV6G). However, it has been shown that Infigratinib can sensitise EGFR mutant and drug-tolerant lung cancer cells, and that dual EGFR and FGFR inhibition may prevent and overcome drug resistance (Raoof et al, 2019). Recently, it has also been found that acquired resistance toward Infigratinib in hepatocellular carcinoma correlates with higher phosphorylation levels of ERBB2/3 and increased EZH2 expression (Prawira et al, 2021). We also found decreased phosphorylation levels of EZH2 after Infigratinib treatment, including phosphorylation on T487 of EZH2, which is known to induce cell growth (Fig. EV6H). Together, these results imply a functional connection between ERBB2/HER2 and FGFR signalling, which might rationalise both the hypothesis raised from the elastic net regression and the follow-up experiments.

## Discussion

Identifying associations between phenotypic drug responses and genomic, transcriptomic and proteomic signatures has proven useful in understanding the modes of action of drugs and to identify markers of drug resistance or sensitivity. Several large-scale efforts have been established to provide such resources for the community e.g., CCLE, CTRP, DepMap etc. (Barretina et al, 2012; Frejno et al, 2020; Garnett et al, 2012; Rees et al, 2016; Reinhold et al, 2012; Tsherniak et al, 2017). However, the molecular complexity of cellular systems often renders the interpretation of associations difficult. As protein phosphorylation plays crucial roles in oncogenic signalling, we previously extended drug phenotype–proteotype association analysis to the level of the phosphoproteomes of 125 diverse cancer cell lines (Frejno et al, 2020). However, not many clinical KIs were included in these experiments and the focus on biologically diverse cell lines often limited or even confounded the results.

In this study, we characterised the proteomes and phosphoproteomes of a small set of sarcoma lines representing many histological subtypes and, in parallel, systematically created phenotypic responses to drugs focusing on clinical KIs. These datasets allowed us to explore associations in a more controlled fashion than taking all or partial data from the literature. The analysis showed that many associations (drug targets or downstream proteins/p-sites) reported in the literature could be recapitulated. More importantly, many phosphorylation sites were identified that directly or indirectly associated drugs with targeted pathways in a plausible fashion and one of these was confirmed experimentally as an example. We also note that many of these were only apparent at the level of the phosphoproteome.

However, it is clear that the associations observed in this study do not provide the full picture regarding the MoAs of the drugs investigated. Correlations do not always translate into causality and the number of cell lines in this study was relatively small, limiting statistical power in many cases. Therefore, the many new

hypotheses that can be generated by mining the information deposited in ProteomicsDB will need experimental validation.

On a more technical level, our workflow isolated the complete phosphoproteome which typically leads to the identification of relatively few tyrosine phosphorylated peptides (713 pY-sites of a total of 53,087 p-sites identified across all cell lines; about 230 pY-sites per cell line on average) and thus potentially missing important information. For instance, a previous study (Bai et al, 2012) reported ~1900 pY-peptides across several sarcoma cell lines of which 39 were found on tyrosine kinases. From this data, the authors identified potential driver tyrosine kinases in A204 and RD-ES cells (both also used in the current study). These authors found a dependency of A204 cells on PDGFR and FGFR signalling, which was confirmed by another study (Wong et al, 2016) showing that dual inhibition of PDGFR and FGFR signalling had a synergistic effect. Our data also identified the FGFR association because the overall phosphoproteomic signature captured by our activity score was strong enough. However, in the same study from Bai et al, RD-ES cells were found to be dependent on IGF1R/INSR signalling. We could not confirm this from our results, as the activity scores of these two kinases (and many further kinases) were relatively low (Fig. 3A; Dataset EV6) and our compound deck did not include IGF1R/INSR inhibitors. A prevailing issue is the fact that most phosphorylation sites are still not functionally annotated. This remains a major obstacle for the functional interpretation of the proteotype–phenotype drug profiles. While the associations made in this study can aid in the functionalisation of the phosphoproteome, it can be anticipated that stronger information will come from phosphoproteomic analysis that include direct drug, genetic or other perturbations as well as further efforts to define kinase-substrate relationships in a cellular context (Johnson et al, 2023; Mitchell et al, 2023; Zecha et al, 2023).

Despite these limitations, the current study expands the repertoire of cell lines for which parallel high-quality phenotypic drug response and (phospho)proteomic data exist. Such datasets serve several important purposes. Broadly speaking, they provide a straightforward means to better understand the often puzzlingly heterogeneous drug effects in the context of molecularly hetero-geneous cancer model systems. More specifically, integration of such data may identify molecular markers of drug response which may be used (i) to better understand drug effects in vitro, (ii) as pharmacodynamic biomarkers in preclinical 3D culture, organoid or animal models or (iii) to measure drug response in patients. It is worth noting that the activity scores computed from the experimental data may be used more broadly than portrayed in this study. As mentioned above, high-scoring kinases in a particular cell line may not necessarily be relevant for the phenotype assayed for (here viability). However, the same scores may be used for finding molecular associations for any other phenotypic readout, which, in the long run, will provide further functional context.

An important future potential use, and a major motivation to undertake the current study in the first place, is in precision cancer medicine. It has become apparent that even cancers of similar origin or pathological classification can have vastly different genotypes and proteotypes. As the amount of proteomic patient profiling data is increasing rapidly, it becomes more and more important to be able to derive potential treatment options from the proteomic profiles of individual patients. Even though cancer cells are imperfect models for human cancer in vivo, cellular systems are

amenable for systematic drug response screening, while patient-derived material is often not. Therefore, strong associations made in vitro between molecular signatures at the (phospho)proteome level and the response to certain cancer drugs offer the opportunity to make novel treatment recommendations in molecular tumour boards or provide new ideas for clinical trials. The current work is placed within this context and the authors anticipate that making the data available in ProteomicsDB will engage the scientific community to mine the data in this or other directions.

# Methods

## Cell lines

Cell lines used in this study were obtained from ATCC, CLS (CLS Cell Lines Service GmbH, Germany), and DSMZ (German Collection of Microorganisms and Cell Cultures GmbH, Germany). SYO-1 is not commercially available and was a kind gift from Dr. Stefan Fröhling (NCT/DKFZ Heidelberg, Germany). The detailed information can be found in Dataset EV1. Cell lines were not authenticated for this study but came with authentication certificates from purchased sites.

## Drug library

The compounds were purchased from Selleckchem (Absource Diagnostics GmbH, Germany) and MedChemExpress (Hölzel Diagnostika Handels GmbH, Germany). As manufacturers noted, the compounds were dissolved in DMSO except for three compounds that were not soluble in DMSO. The stock concentration was 10 mM, whereas four compounds were made of the maximal concentration according to their solubility. Two drugs, Eribulin and Trabectedin, were donated from the hospital Klinikum Rechts der Isar (Munich, Germany) after regular chemotherapy and in a lower concentration of 0.9% NaCl. Detailed information on compounds can be found in Dataset EV2.

## Selection of cell lines and drugs

The selection of cell lines started from a published dataset including 63 human sarcoma lines that were screened against 100 FDA-approved oncology agents and 345 investigational agents (Teicher et al, 2015). Only 28 of these cell lines were commercially or public available (January 2019; Appendix Table S1). To maximise diversity in drug response, we correlated the drug responses of these 28 cell lines toward 52 KIs using the R package "pheatmap" (Appendix Fig. S6). Next, both diversity in sarcoma subtypes and phenotypic diversity (whether they were clustered closely) were used for selection. Four subtypes, including Ewing sarcoma, fibrosarcoma, osteosarcoma and rhabdomyosarcoma, are represented by more than one cell line because the above correlations between these cell lines were poor.

The choice of compounds (total of 150, Dataset EV2) focussed on KIs because of the focus of the current study on the phosphoproteome. The screening deck contained all 71 approved KIs (at the time of writing) and 52 phase III drugs. These 123 KIs are particularly valuable as they may be repurposed and/or recommended for treatment in a compassionate use setting.

Further, 16 phase I, II, or II/III KIs were included to cover complementary target proteins (Klaeger et al, 2017). In addition, four chemo drugs and one epigenetic drug already used for treating sarcomas today were included, as well as six non-KIs used for treating other types of cancer.

## Cell culture

Cell lines were grown in the conditions suggested by manufacturers or literature (Teicher et al, 2015). Upon receiving, the cells were grown and frozen in six vials. One of the vials was used to generate another six stock vials. Cells from the same batch of frozen stock (at the same passage) were thawed for the drug screen and proteomics analysis separately. In both conditions, the cells were cultured in less than two months. The additional supplemented with Pen Strep (100 U/mL penicillin and 100 ug/mL streptomycin, Gibco #15140-122) was added to the cell culture for the high-throughput drug screen. Detailed information on culture conditions can be found in Dataset EV1. All cell lines were grown at 37 °C and 5% $CO_2$.

## High-throughput drug screen

Prior to the drug screen, the density test was performed. Each cell line was cultivated in 384-well plates (CulturPlate™, Perkin Elmer, MA, US) with different densities ranging from 750–7500 cells per well in 50 µl using a Multidrop™ Combi Reagent Dispenser (Thermo Fisher Scientific, Waltham, MA, USA). After 96-h incubation, the density reached 80–90% of confluence was chosen. The cell number used for each cell line can be found in Dataset EV1. For the drug screen, cells were seeded in 384-well plates 24 h prior to drug treatment and cultured in standard conditions. Five compound plates were prepared with ten dilution series from 10 mM to 0.0002 mM in one to three steps. Six compounds with lower stock concentrations were diluted similarly, resulting in different final concentrations. Each experiment was performed in triplicate by adding 2.5 µl of dilution series and incubated for 72 h. Five micro molars of Staurosporine (Selleckchem #S1421) served as a positive control in each plate, whereas 0.1% DMSO as a negative control (Appendix Table S2). The viability was measured using the ATPLite assay following the manufacturer's instructions (Perkin Elmer, MA, USA) and calculated by the percent of cell growth. The Z-Prime of all assays ranges from 0.76 to 0.96.

## Cell lysis

For proteomics analysis, adherent cells grown in the 15-cm Petri dish (Greiner Bio-One GmbH, Germany) to a confluence of 80–90%. The adherent cells were washed twice with cold PBS and lysed in lysis buffer (40 mM Tris-HCl, pH 7.6, 2% of SDS). The suspension cells were harvested by centrifugation in 50-mL falcon at 1000×g for 5 min at 4 °C. After two washes with PBS and centrifugation, the cell pellet was suspended by lysis buffer. The cell lysates were frozen at −80 °C freezer until further use. For the time-dependent experiment, the cells were grown in the 10-cm Petri dish and harvested at the indicated time point with PBS wash, followed by adding lysis buffer. Cell lysates were boiled at 95 °C for 5 min, and trifluoroacetic acid (TFA) was added to a final concentration of 1% (Dagley et al, 2019) to hydrolyse DNA and reduce viscosity. Three molar Tris (pH 10) was added to reach a final concentration

of 195 mM (pH 7.8) to quench the samples. Protein concentration was determined using the Pierce BCA Protein Assay Kit (Thermo Scientific, MA, USA).

## Time-dependent treatment

The cell line G401 was seeded to the 10-cm cell petri dish (Greiner Bio-One GmbH, Germany), with $1.7 \times 10^6$ cells of each dish. After incubation for two days, 0.1 μM of Infigratinib or DMSO was added to the cells. The remaining untreated dishes were lysed immediately as time point zero. The other dishes were lysed at 5 min, 10 min, 15 min, 30 min, 90 min, 3 h, 6 h, 12 h and 24 h after treatment. The lysis procedure was described in the previous section. The cell lysate was stored at −80 °C after boiling at 95 °C for 5 min.

## Lysate clean-up, protein digestion and peptide de-salting

The lysate was cleaned up using the SP3 method on an automated Bravo liquid handling system (Agilent Technologies, CA, US) as previously described (Hughes et al, 2019) with minor modifications (Zecha et al, 2020). In short, 20 μl of carboxylate beads mix (50 μg/μl in $H_2O$, Sera-Mag Speed beads, cat# 45152105050250 and 65152105050250, GE Healthcare, IL, USA) were added to a 96-well plate (cat#951020401, Eppendorf, Germany). Following by one precipitation step with 60% ACN, two wash steps with 80% ethanol, and one wash step with 100% CAN, the beads were incubated with 100 μl of digestion buffer (50 mM HEPES, pH 8.5; 10 mM Tris(2-carboxyethyl)phosphine (TCEP); 50 mM chloroacetamide (CAA)) for 1 h at 1200 rpm and 37 °C. After reduction and alkylation, samples were digested overnight at 37 °C and 1200 rpm using a 1:50 trypsin-to-protein ratio. The supernatant containing peptides was transferred to a new 96-well plate and acidified by formic acid (FA) to reach a final concentration of 1%. Peptides were desalted using RP-S cartridges (5 μL bed volume, Agilent) and the standard peptide cleanup v2.0 protocol on the AssayMAP Bravo Platform (Agilent). Briefly, RP-S cartridges were primed with 100 μL of 50% ACN/0.1% FA and equilibrated with 50 μL of 0.1% FA at a flow rate of 10 μL/min. The samples were loaded at 5 μL/min, followed by an internal cartridge wash with 0.1% FA at a flow rate of 10 μL/min. Peptides were eluted with 80 μL of 70% CAN/0.1% FA at a flow rate of 5 μL/min. Samples were dried down and stored in −80 °C until further use.

## TMT labelling and multiplexing

The desalted peptides were labelled with TMT6 or TMT11 plex as previously described (Zecha et al, 2019). In short, one hundred μg of TMT reagent in 5 μl of anhydrous ACN was used to label 100 μg of peptides in 20 μl of 50 mM HEPES buffer (pH 8.5) and the labelling reaction was stopped by adding 3 μl of 5% hydroxylamine. After drying to remove excessive ACN, the (pulled) peptides were dissolved in 0.1% FA and desalted by 250 mg Sep-Pak C18-Cartridges (Waters, MA, USA) after vacuum drying. For reasons of compatibility with another ongoing project in the laboratory, peptides for baseline (phospho)proteomic analysis were labelled with TMT (6-plex) but samples were not (!) multiplexed. Instead, each cell line sample was analysed separately and the MS1 peptide intensity data was used for quantification purposes. For time-dependent (phospho)proteomics analysis, the TMT-labelled peptides were multiplexed in two batches of TMT11 (Lot number: TMT10plex, UK291565; TMT11, UH283151). Each batch consisted of nine time points (DMSO or Infigratinib-treated) with two identical zero time point samples to enable bridging data between batches (bridge channel).

## HPLC fractionation

TMT-labelled peptides were fractionated by off-line basic reversed-phase (bRP) fractionation as previously described (Zecha et al, 2020) with slight modifications for phosphoproteomics analysis. In brief, Dionex Ultra 3000 HPLC system equipped with a XBridge BEH130 C18 column (3.5 μm 4.6 × 250 mm) (Waters, MA, USA) was operated at a flow rate of 1 mL/min with a constant 10% of 25 mM ammonium bicarbonate (pH 8.0) in the running solvents. A 57-min linear gradient from 7 to 45% ACN followed by a 6 min linear gradient up to 80% ACN was performed. Ninety-six fractions were collected every half minute from min 7 to 55 and pooled into 48 fractions (fraction 1 + 49, fraction 2 + 50, etc.). To acidify the samples, 50% (v/v) FA in water were added to a final concentration of 1% (v/v) FA. For proteomics analysis, 75 μl of total volume (~15% of total peptide amount) was transferred to a new 96-well plate. Both plates were dried down and stored at −80 °C.

## Phosphorylation enrichment

The dried fractions remaining from bRP HPLC fractionation were dissolved in IMAC Equilibrium buffer (80% ACN/19.9% ddH2O/0.1% TFA) and pooled into 12 fractions for phosphoproteomics analysis. Twelve fractions of phosphorylated peptides were enriched in 12 Fe(III)-NTA cartridges (5-μL bed volume, Agilent) with Phosphopeptide Enrichment v2.0 protocol on the AssayMAP Bravo Platform (Agilent). Briefly, the cartridges were primed with 150 μL of IMAC Priming buffer (99.9% CAN/0.1% TFA) and equilibrated with 150 μL of IMAC Equilibrium buffer at a flow rate of 10 μL/min. Next, the samples were loaded at five μL/min, followed by three internal cartridge washes with IMAC Equilibrium buffer at 50 μL/min. Finally, the phosphorylated peptides were eluted with 50 μL of 1% ammonia at a flow rate of 5 μL/min into 50 μL of 10% formic acid. Samples were dried down and stored at −80 °C until subjected to LC–MS/MS.

## LC–MS/MS analysis

### Full proteomes

For proteomics profiling, a Dionex UltiMate 3000 RSLCnano System equipped with a Vanquish pump module and coupled to a Fusion Lumos Tribrid mass spectrometer (Thermo Fisher Scientific) was operated under micro-flow conditions as we described recently (Bian et al, 2021). Peptide fractions were dissolved in 1% FA/2%ACN and the fraction corresponding to 3–4 μg peptides was injected directly to Acclaim PepMap 100 C18 column (2-μm particle size, 1 mm ID × 150 mm; Thermo Fisher Scientific). Peptides were separated at a flow rate of 50 μl/min using a 25-min linear gradient of 4–32% micro-flow solvent B (0.1% FA and 3% DMSO in ACN) in micro-flow solvent A (0.1% FA and 3% DMSO in water). The data-dependent acquisition (DDA) with an H-ESI source were used to measure all the baseline proteomes and time-dependent proteomes with a high-resolution orbitrap (OT)

method. In brief, full scan MS1 spectra were recorded in the OT from 360 to 1600 $m/z$ at 60k resolution using an automatic gain control (AGC) target value of 4e5 charges and a maximum injection time (maxIT) of 50 ms. For baseline proteomes, MS2 spectra were acquired in the OT at 50k resolution after higher energy collisional dissociation (HCD, 34% NCE) and using an AGC target value of 1e5 charges, a maxIT of 86 ms, an isolation window of 0.7 $m/z$, and an intensity threshold of 5e4. The cycle time was set to 1.2 s and the dynamic exclusion lasted for 40 s. For time-dependent proteomes, MS2 spectra were acquired in the IT (ion trap) after HCD (32% NCE) and using an AGC target value of 1.2 e4 charges, a maxIT of 40 ms, an isolation window of 0.6 $m/z$, and an intensity threshold of 1e4. The quantitative information on TMT reporter ions was obtained by synchronous precursor selection (SPS) of up to eight most intense peptide fragments (McAlister et al, 2014) and further fragmentation via HCD using a NCE of 55%. The MS3 scan was recorded in the OT at 50k resolution (scan range 100–1000 $m/z$, isolation window of 1.2 $m/z$, AGC of 1e5 charges, maxIT of 86 ms). The cycle time was 1.2 s and the dynamic exclusion lasted for 50 s.

### Phosphoproteomes

For phosphoproteomics profiling, peptide fractions were dissolved in 0.1% FA and a half (time-dependent) or one-third (baseline) of the fraction was loaded to a trap column (75 μm × 2 cm, packed in-house with 5-μm C18 resin; Reprosil PUR AQ, Dr. Maisch). After washing with 0.1% FA at a flow rate of 5 μL/min for 10 min, peptides were transferred to an analytical column (75 μm × 45 cm, packed in-house with 3-μm C18 resin; Reprosil PUR AQ, Dr. Maisch). Peptides were separated at a flow rate of 300 nL/min using an 80-min linear gradient of 4–32% of solvent B (0.1% FA and 5% DMSO in ACN) in solvent A (0.1% FA and 5% DMSO in water). The data-dependent acquisition (DDA) was used with a high-resolution orbitrap (OT) method. In brief, full scan MS1 spectra were recorded in the OT from 360 to 1500 $m/z$ at 60k resolution using an automatic gain control (AGC) target value of 4e5 charges and a maximum injection time (maxIT) of 50 ms. For baseline phosphoproteomes, MS2 spectra were acquired in the OT at 15k resolution after higher energy collisional dissociation (HCD, 33% NCE) and using an AGC target value of 2e5 charges, a maxIT of 55 ms, an isolation window of 1.2 $m/z$, and an intensity threshold of 2.5e4. The numbers of the dependent scan was set to 25 and the dynamic exclusion lasted for 90 s. For time-dependent phosphoproteomes, MS2 spectra were acquired in the OT at 30k after collision-induced dissociation (CID, 35% CE) and using an isolation window of 0.7 $m/z$. Neutral loss mass was set to 97.9763. The quantitative information on TMT reporter ions was obtained by synchronous precursor selection (SPS) of up to 10 most intense peptide fragments in the OT and further fragmentation via HCD using a NCE of 55%. The MS3 scan was recorded in the OT at 50k resolution (scan range 100–1000 $m/z$, isolation window of 1.2 $m/z$, AGC of 1.2e5 charges, maxIT of 120 ms). The cycle time was 3 s and the dynamic exclusion lasted for 90 s.

## Data analysis

### Raw data processing

The raw files were searched in MaxQuant 1.6.2.10 (Tyanova et al, 2016) against a human database provided by UniProt (downloaded

Aug 11, 2018) and common contaminants. For baseline proteomes, the experiment type was left in default settings with a false discovery rate (FDR) cutoff of 1%. Searches were restricted with a precursor ion tolerance of 20 ppm and a fragment ion tolerance of 0.4 $m/z$. TMT-labelling N-terminus and lysine (229.1629) modification were considered as fixed modification, while cysteine carbamidomethylation, methionine oxidation and N-terminus acetylation were allowed as variable modifications. Trypsin/P was specified as the proteolytic enzyme with up to two missed cleavage sites allowed, and absolute quantification by iBAQ was enabled. Matching was enabled between fractions of samples (20 min alignment window, 0.2-min matching window). Default score cutoffs required a minimal Andromeda score of 40 and a delta score of 6 for modified peptides. For time-dependent proteomes, all the settings mentioned above were used with a few modifications. The 11plex TMT was specified as isobaric label within a reporter ion MS3 experiment type. Cysteine carbamidomethylation was considered as a fixed modification, while methionine oxidation and N-terminus acetylation were allowed as variable modifications. For baseline and time-dependent phosphoproteomes, the settings mentioned above were used with a few modifications. STY phosphorylation was added to the variable modifications, and matching was enabled between fractions of samples in 20-min alignment window and 0.7-min matching window. Protein quantification was obtained from the summed area under peptide elution profiles for baseline samples or from summed peptide reporter intensities for time-dependent treatment samples.

### Data post-processing

The files "proteinGroups.txt" and "Phospho (STY)Sites.txt" from MaxQuant were used for the proteome and phosphoproteome analyses. The abundance of proteins and phosphosites (p-sites) was quantified using intensity-based absolute quantification (iBAQ) (Schwanhausser et al, 2011) and MS1 precursor intensities for baseline (phospho)proteomes and corrected reporter intensity for time-dependent (phospho)proteomes. P-sites were not filtered a priori for localisation probability, yet the probabilities are provided for all p-sites in the online data matrix online when exploring elastic net regression and correlation data (https://www.proteomicsdb.org/sarquarium). Moreover, 85% of the identified phosphosites exhibit a localisation probability greater than 0.75, the conventional definition of class I phosphosites (Appendix Fig. S7). The post-processing were performed in RStudio (version 1.4.1717) using R (version 4.1.1) with packages "data.table" and "missMethods" (RCoreTeam, 2021; Rockel, 2020; RStudioTeam, 2020; Srinivasan, 2021). Hits to the reverse and contaminant database were removed. To correct for different loading amounts of the samples, quantification values were normalised by median-centring all samples to the overall median of the respective dataset (Appendix Fig. S8). Missing values in full proteome and phosphoproteome data were imputed using the protein-wise and p-site-wise (row-wise) half-lowest observed intensity method. The log10-transformed data of non-imputed and imputed intensities in baseline (phospho)proteomics profiling were provided in Dataset EV4 (full proteome) and Dataset EV5 (phosphoproteome). For time-dependent (phospho)proteomes, the reporter intensities were median-centric normalised without any imputation and reported in Dataset EV9. Graphic presentations were generated by BioRender, GraphPad Prism 5 (version 5.01), Adobe Illustrator CS6 (version

16.0.0), R packages "pheatmap" (Kolde, 2019), "RColorBrewer" (Neuwirth, 2014), "dbplyr" (Ruiz, 2021), "tidyr" (Wickham, 2021), "beeswarm" (Trimble, 2021), "reshape2" (Wickham, 2007), "janitor" (Firke, 2021), "basicTrendline" (Yu, 2020), "ggplot2" (Wickham, 2016).

### Cell viability AUC calculation

The dose–response data were normalised into a range between 1 (no response or full viability) and 0 (full response or no viability). Then, the classical symmetric four-parameter log-logistic model was fitted to each combination of drugs and cell lines:

$$f(x, (b, c, d, e)) = c + \frac{d - c}{1 + \exp(b(\log(x) - \log(e)))}$$

Where the four parameters correspond to: c="Lower Limit", d="Upper Limit", b="Slope" and e="ED50". Finally, the AUC was calculated as the standardised area under the dose–response curve in dose logspace using a modified computeAUC function from the drexplorer package (Zou and Hastie, 2005) v1.1.22 as described earlier (Frejno et al, 2017). These standardised AUC values also range between 1 (no response) and 0 (full response).

### Activity score calculation

The activity score of each kinase was calculated according to a previously proposed approach with a few modifications (Frejno et al, 2020). In brief, four matrices were generated. (a) kinase abundance (log10-transformed) from full proteome data were mapped to the kinase list downloaded from Uniprot (pkinfam, 08.11.2021). The z-scores were calculated by the R package "som" (Yan, 2016) across 17 cell lines. (b) kinase phosphorylation from phosphoproteome data were extracted according to the kinase list. The abundances of p-sites (non log10-transformed) from the same kinase were summed up followed by the log10 transformation and z-scores calculation. (c) kinase active loop phosphorylation from phosphoproteome data were mapped with active loop analysis from Phomics (Munk et al, 2016) followed by z-scores calculation. (d) kinase-substrate phosphorylation from phosphoproteome data were mapped to kinase-substrate list and the abundances of substrates (non log10-transformed) from the same kinase were summed up, followed by the log10 transformation and z-scores calculation. As a hard requirement, there has to be kinase abundance information from the full proteome data. Otherwise, no score is calculated for this kinase. We then add abundance of any of the three other layers (b, c, d) to the kinase (a) regardless of whether or not all three are available. To report the druggable activity landscapes, we curated a list from target profiling on ProteomicsDB (https://www.proteomicsdb.org/, downloaded on Feb 23, 2022). All the matrices mentioned above can be found in Dataset EV6.

### Calculation of proportions of cells in different cell cycle phases

Briefly, the pseudoperiodic protein clusters from Kelly et al (Kelly, 2022) was downloaded. In that study, the proteomics profiling of sixteen cell populations collected across annotated phases of the cell cycle showed five clusters containing a total of 119 "periodic" proteins. For each of the 17 cell lines in our study, the proportions of cells in different cell cycle phases from the bulk proteome data

were inferred as follows: (1) The identified proteins in our study were mapped to the list of pseudoperiodic proteins clusters of Kelly et al (all 119 proteins were found). (2) The intensities of all proteins in each Kelly cluster was summed up for each cell line. (3) Each Kelly cluster was rescaled from 0 to 1 among 17 cell lines to remove the bias from clusters that contain more proteins (higher summed intensity). (4) We calculated and normalised (0 to 1) the proportion of each Kelly cluster in each cell line (value of each cluster compared to the total sum of the five Kelly clusters in one cell line). These steps generated an approximation of the proportion of cells in a particular stage of the cell cycle (Dataset EV7 Tab3). Second, the Pearson correlation of each Kelly cluster proportion with drug response for each drug was computed. Only 87 drugs were chosen in our data that had a minimum AUC <0.8 to avoid obtaining correlations from drugs with very small effects. The correlations are provided in Dataset EV7 and plotted in pdf files (available via ProteomeXchange).

### Sparse multiblock partial least-square regression (SMBPLSR)

To identify drugs sharing correlated viability profiling and, at the same time, selecting biomarkers that predict the cell line sensitivity of the selected drugs, we used a method extended from partial least-square regression, i.e., sparse multiblock partial least-square regression (SMBPLSR). The method is described in detail in (Frejno et al, 2020; Karaman et al, 2015). Briefly, SMBPLS extended partial least square (PLS) in two ways. First, it accepts multiple matrices storing independent variables as the predictor. Then, SMBPLSR algorithm maximises the summed covariance between the components identified from each independent matrix and the components identified from the dependent matrix, e.g.,

$$argmax(\boldsymbol{u}_1, .., \boldsymbol{u}_k, ..., \boldsymbol{u}_K, \boldsymbol{v}) = \sum_{k=1}^{K} cov^2(\boldsymbol{X}_k \boldsymbol{u}_k, \boldsymbol{Y} \boldsymbol{v})$$

where $\boldsymbol{X}_1$ to $\boldsymbol{X}_K$ are $K$ independent matrices (predictors; protein and phosphorylation intensity matrix), and Y is the dependent matrix (drug sensitivity matrix of cell lines). $\boldsymbol{U}_k$ is a vector storing the linear combination coefficients of variables in the independent matrix $k$ (protein or phosphorylation site loadings). Vector v stores the linear combination coefficient of variables in the dependent matrix (drug loadings). Second, an L1 penalty is introduced on the drug and the proteins/phosphorylation site loadings. The sparsity on drug loading identifies a subset of drugs sharing correlated cell viability profiles. The sparsity on protein/phosphorylation site loading selects a subset of protein and phosphorylation markers that correlates well with the viability profiling of selected drugs. In our analysis, the two independent matrices are the iBAQ intensity of proteins and the intensity of phosphorylation sites. The dependent matrix is the cell line viability profiling of drugs, represented as the area under the curve (AUC). In total, 12 components are fitted. In each component, six drugs are selected, and 50 proteins and phosphorylation sites are identified as potential biomarkers.

### Elastic net regression

To discover proteins and p-sites that are associated with drug response, we applied Elastic net regression (Zou and Hastie, 2005)

to our data. Elastic net regression combines L1 (Lasso) and L2 (Ridge) regularisation to find a relatively sparse combination of protein and p-sites that can predict the response variable (drug response). The hyperparameter α ∈ [0, 1] is used to balance the L2 (α = 0) and L1-penalties (α = 1). A second hyperparameter λ controls the degree of regularisation. Elastic net regression models were fit using the R package glmnet v4.1-2. Elastic net regression was performed as described earlier with minor adaptations (Frejno et al, 2020). First, missing values were imputed by half of the lowest detected intensity for each protein and p-site. For each drug, we used bootstrapping to create multiple elastic net models. The hyperparameter λ was optimised using the cv.glmnet function while fixing α = 0.05. Using this optimal λ, we then selected the optimal α out of 0.01, 0.05 and 0.1, again using cross-validation. With these optimal hyperparameters, we then trained 100 elastic net models and used the average coefficient and selection frequency across all 100 models to create plots resembling classic volcano plots. Proteins and p-sites with a high average coefficient and high selection frequency can be considered candidate biomarkers for predicting drug response for that particular drug. It is useful to note that running Elastic net regression on the same data twice or swapping cell line replicates, does not necessarily lead to the same results but robust candidates will generally reproduce (see Appendix Fig. S9).

### Basic correlation

The correlation coefficients were calculated between the response of a given drug and the abundance of a given protein or p-site across all cell lines. The Pearson correlations with and without the imputation of missing values can be explored on ProteomicsDB (https://www.proteomicsdb.org/sarquarium). For the follow-up analysis, only drug-protein/p-site pairs with more than eight pairwise complete observations were considered to avoid artificially high/low correlations due to imputations.

### Time-dependent (phospho)proteome analysis

The MQ evidence file was used as the primary input for a custom data analysis pipeline (https://github.com/kusterlab/decryptM) (Zecha et al, 2023). This pipeline relied on public libraries such as numpy (version 1.20.2), pandas (version 1.2.4), scipy (version 1.6.3), statsmodels (version 0.12.2), matplotlib (version 3.4.2) and seaborn (version 0.11.1) and was written in python (version 3.9.4). Experimental information such as correspondence between time points and TMT channel ([1:0 min, 3:5 min, 4:10 min, 5:15 min, 6:30 min, 7:90 min, 8:180 min, 9:360 min, 10:720, 11:1440 min]), were stored in a toml-file, which can also be downloaded via ProteomeXchange. First, all impurity-corrected TMT channels were median-based normalised. Methionine oxidation was removed from the modified sequence, and all duplicated modified sequences were summed up. Next, ratios were calculated against the time point zero. Finally, the time-dependent data points were linearly interpolated and sorted by the absolute maximal response. The numbers of up- or downregulated proteins/p-sites were reported by filtering the relative responses with more than twofold (relative ratio <0.5 or >2) at least at one time point. All curves are provided in Dataset EV9 and plotted in pdf files (available via ProteomeXchange). The protein groups file from the full proteome time courses was transformed into the same format as the evidence file and processed as described above. The only difference was that one curve represented a protein group rather than a modified peptide.

## Data availability

The mass spectrometry proteomics data have been deposited to the ProteomeXchange Consortium (http://proteomecentral. proteomexchange.org) (Deutsch et al, 2023) via the MassIVE partner repository with the dataset identifier PXD039363 and PXD046959.

## Peer review information

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

## Acknowledgements

We are grateful to all members of the Kuster lab for fruitful discussions and for providing reagents and tools used in this study. We thank Prof. Stefan Fröhling and Dr. Priya Chudasama (NCT/DKFZ Heidelberg, Germany) for providing cell line SYO-1 and Chemical Biology Core Facility at EMBL Heidelberg for providing the expertise and support for drug screen experiments. This work was partly funded by the Federal Ministry of Education and Research (CLINSPECT-M, FKZ161L0214A; YIG-SysNS, FKZ161L0215; DIAS, FKZ 031L0168), the European Research Council (ERC AdG grant no. 833710 and no. 834154). CYL acknowledges the Postdoctoral Research Abroad Program from the Ministry of Science and Technology of Taiwan (grant no. 108-2917-I-564-034) and TUM University Foundation Fellowship for providing funding. All schematic figures were created with BioRender.com.

## Author contributions

**Chien-Yun Lee**: Conceptualisation; Data curation; Formal analysis; Supervision; Validation; Investigation; Visualisation; Writing—original draft; Project administration; Writing—review and editing. **Matthew The**: Resources; Data curation; Software; Formal analysis; Methodology; Writing—review and editing. **Chen Meng**: Formal analysis; Methodology. **Florian P Bayer**: Software; Formal analysis; Methodology. **Kerstin Putzker**: Formal analysis; Methodology. **Julian Müller**: Data curation; Software; Visualisation; Methodology. **Johanna Streubel**: Formal analysis; Validation. **Julia Woortman**: Formal analysis; Methodology. **Amirhossein Sakhteman**: Data curation; Visualisation; Methodology. **Moritz Resch**: Formal analysis; Validation. **Annika Schneider**: Formal analysis; Validation; Writing—review and editing. **Stephanie Wilhelm**: Conceptualisation; Project administration. **Bernhard Küster**: Conceptualisation; Supervision; Funding acquisition; Investigation; Writing—original draft; Project administration; Writing—review and editing.

## Funding

## Disclosure and competing interests statement

BK is a co-founder and shareholder of OmicScouts and MSAID. He has no operational role in either company. The remaining authors declare no competing interests.

# Expanded View Figures

**Figure EV1.  Summary of data quality.**

(**A**) Distributions of coefficient of variation (CV) of AUC (area under the curve) values determined from cell viability assays ($n = 3$) in response to 150 cancer drugs (each circle represents one drug). The central band represents the median, while the hinges denote the first and third quartiles with whiskers extending up to 1.5 times the interquartile range (IQR). (**B**) The number of quantified protein groups/peptides (left) and p-sites/phosphorylated peptides (right). (**C**) Unsupervised hierarchical clustering of biological triplicates of four cell lines on protein (left) and phosphoprotein (right) level. Pearson correlation was shown. R2, R3: biological replicates.

**A**

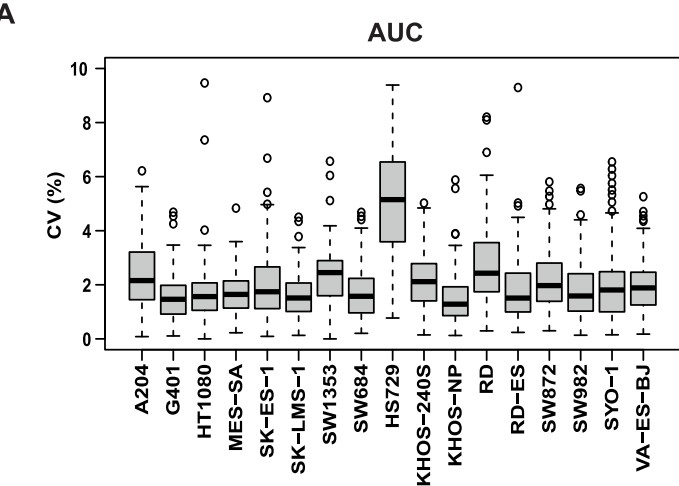

**B**

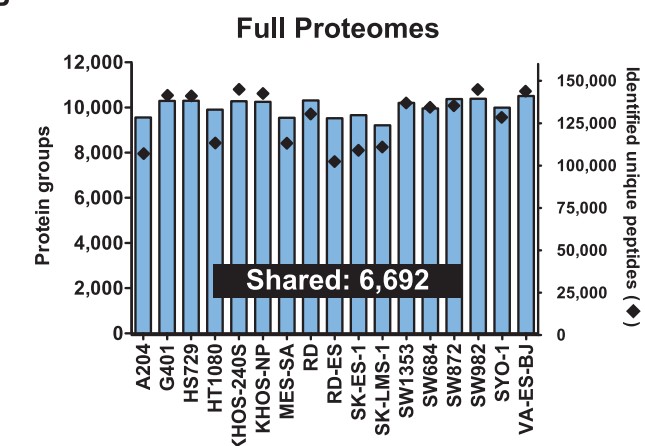

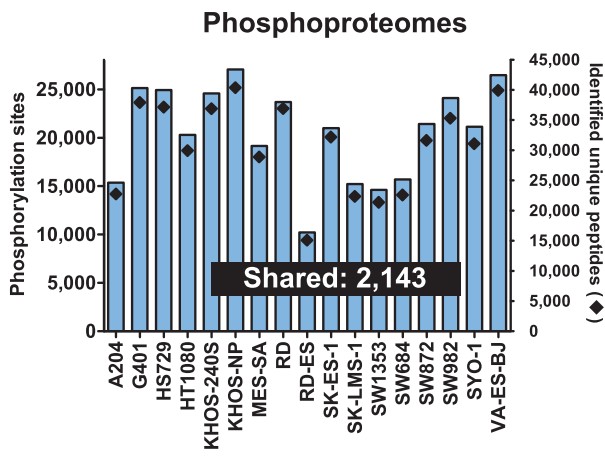

**C**

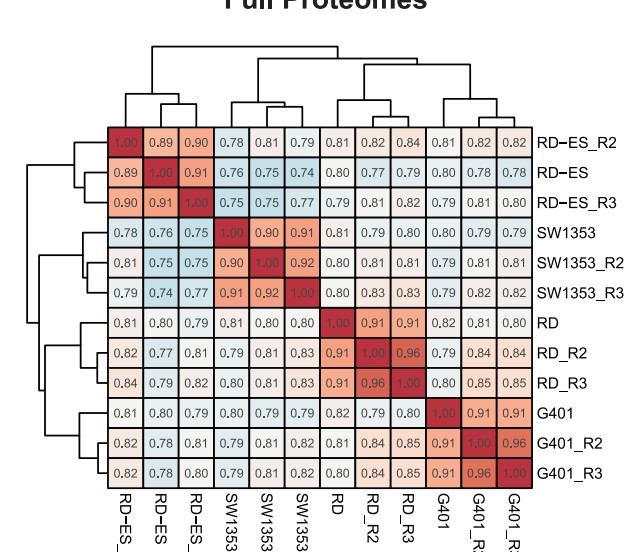

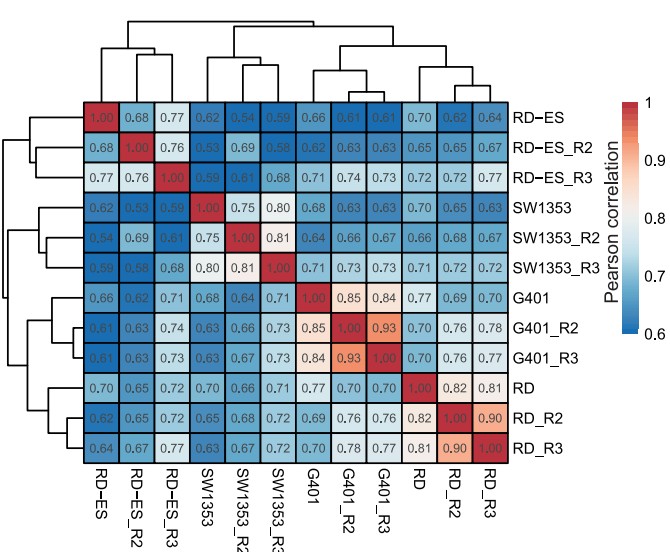

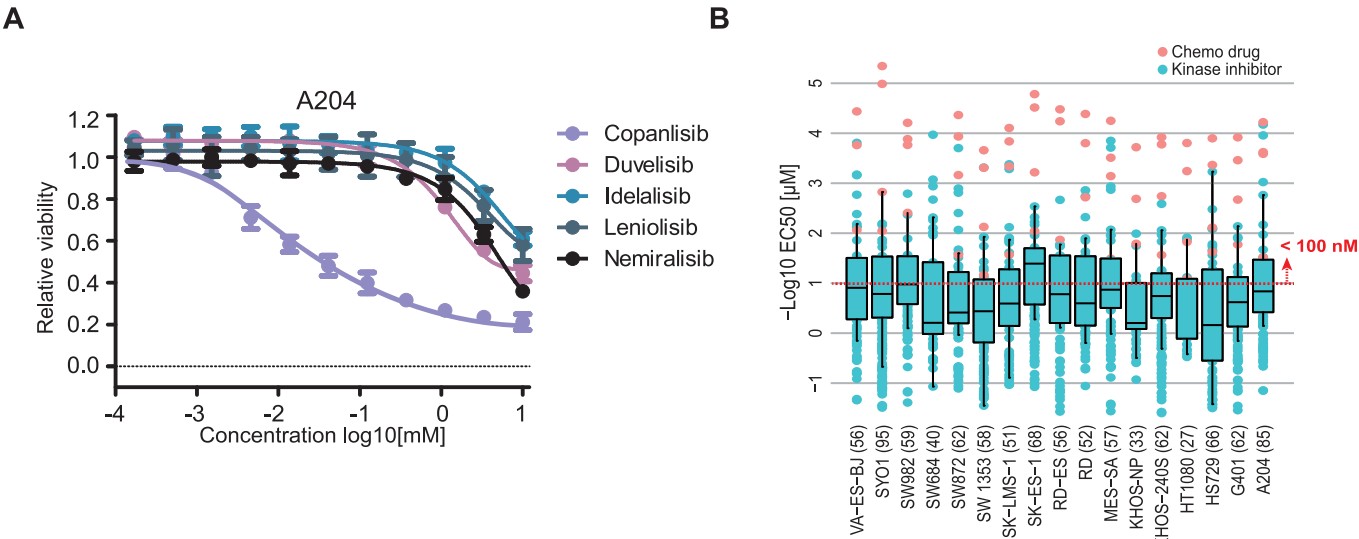

**Figure EV2.  Summary of the phenotypic dose–response characteristics of all cell lines and all drugs.**

(**A**) Viability curves of five selected PI3K inhibitors in A204 cells. Measurements were in technical triplicates and the error bars were shown in SD. (**B**) Distribution of calculated EC50 values for all cell lines and all drugs. Numbers in brackets after each cell line name indicate the number of drugs that show effects larger than AUC>0.9 and have relative effect sizes of >50%. The red dotted line marks an EC50 value of 100 nM. The central band represents the median, while the hinges denote the first and third quartiles with whiskers extending up to 1.5 times the IQR.

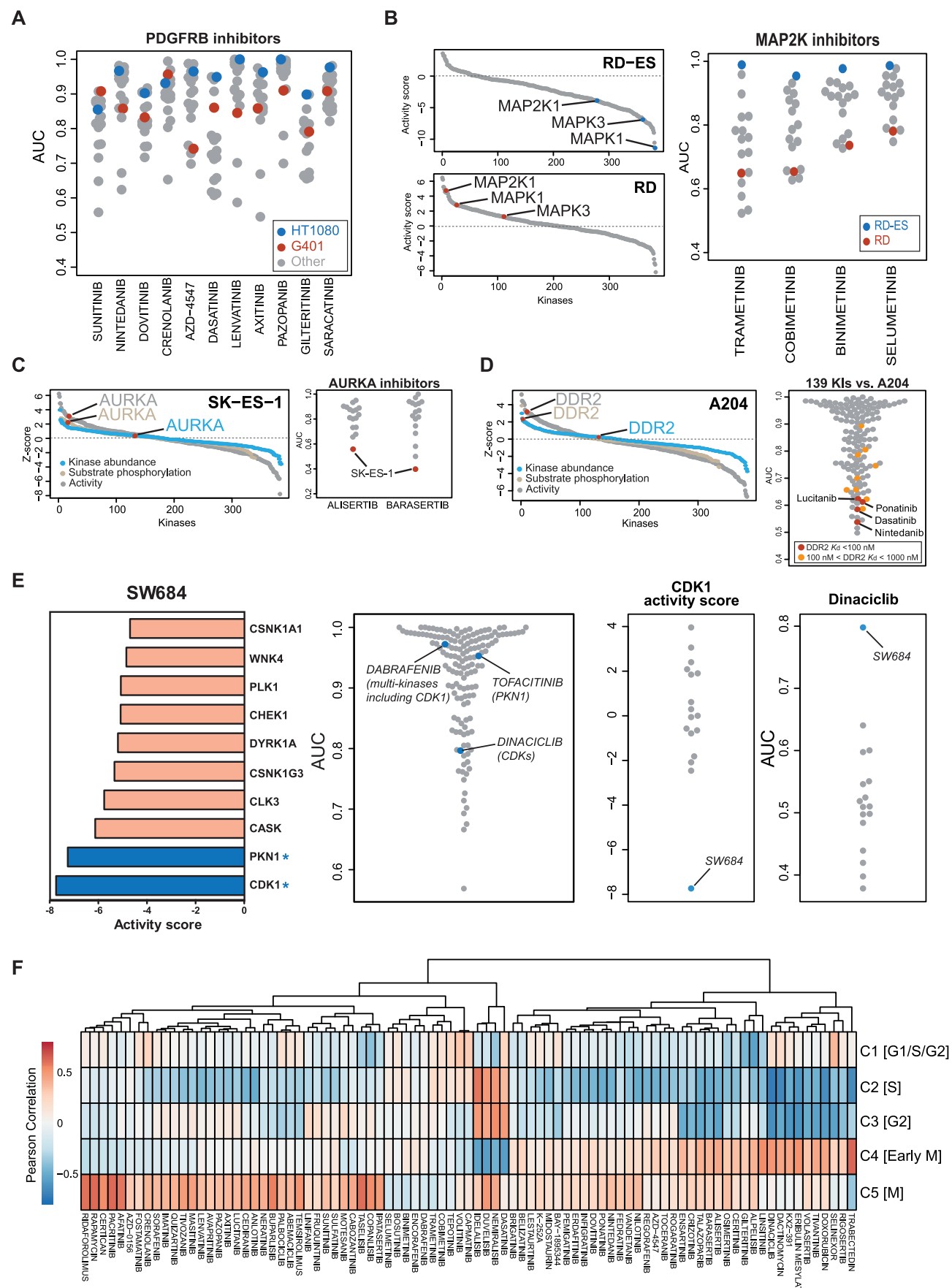

**Figure EV3.   Integration of phenotypic drug response and proteomic activity scores.**

(A) Phenotypic drug response of two sarcoma lines towards 11 PDGFRB inhibitors. (B) Distribution of activity scores of proteins in RD-ES (top panel) and RD cells (bottom panel) showing that MAP2K1 and its downstream substrates MAPK1 and MAPK3 have different activities in the two cell lines. This translates into differences in phenotypic response of the two cell lines to MAP2K inhibitors (right panel). (C) Ranked list of kinases either considering kinase abundance, substrate phosphorylation abundance or computed activity in SK-ES-1 cells (left panel) showing that AURKA phosphorylation and activity is substantially higher than protein abundance. This translates into a strong response of the cell lines to AURKA inhibitors (right panel). (D) Same as panel (C) but for DDR2 in A204 cells. (E) Left panel: list of kinases ranked by activity score in SW684 cells. The asterisk denotes a kinase that is a target of the kinase inhibitor. Middle panel: relative sensitivity of SW684 cells to all drugs in the screen. Right 2 panels: CDK1 activity score and drug responses toward Dinaciclib among 17 sarcoma lines. (F) The heatmap summarises the Pearson correlations of proportions of cell cycle clusters (from Kelly et al) and drug responses from 87 drugs (AUC <0.8).

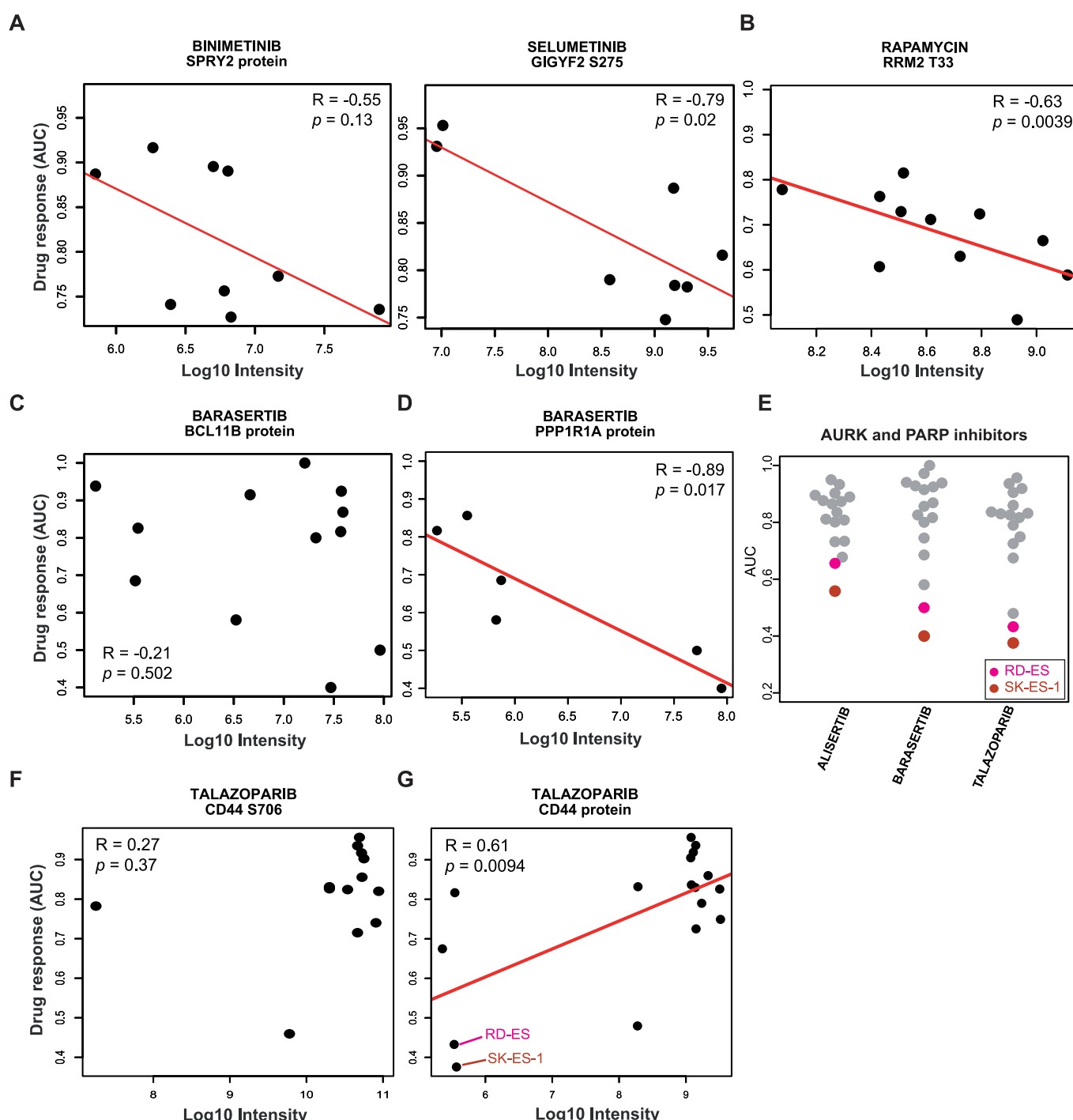

**Figure EV4. Examples of correlation analysis of phenotypic drug sensitivity and protein or p-site abundance for drugs forming clusters in sparse multiblock partial least-square regression (SMBPLSR) analysis.**

(A) SPRY2 protein levels as a marker for sensitivity towards Binimetinib (left panel) and GIGYF2 S275 phosphorylation abundance as a sensitivity marker for Selumetinib (right panel; both drugs are MAP2K inhibitors). (B–D) Same as panel (A) but for Rapamycin (mTOR inhibitor) and Barasertib (PARP inhibitor). (E) The response of two Ewing sarcoma lines (in red and pink) towards AURK and PARP inhibitors is more pronounced than for all other cell lines. (F, G) Same as panel A but for Talazoparib (PARP inhibitor). Again, the two Ewing sarcoma lines (in red and pink) are the most sensitive to this drug.

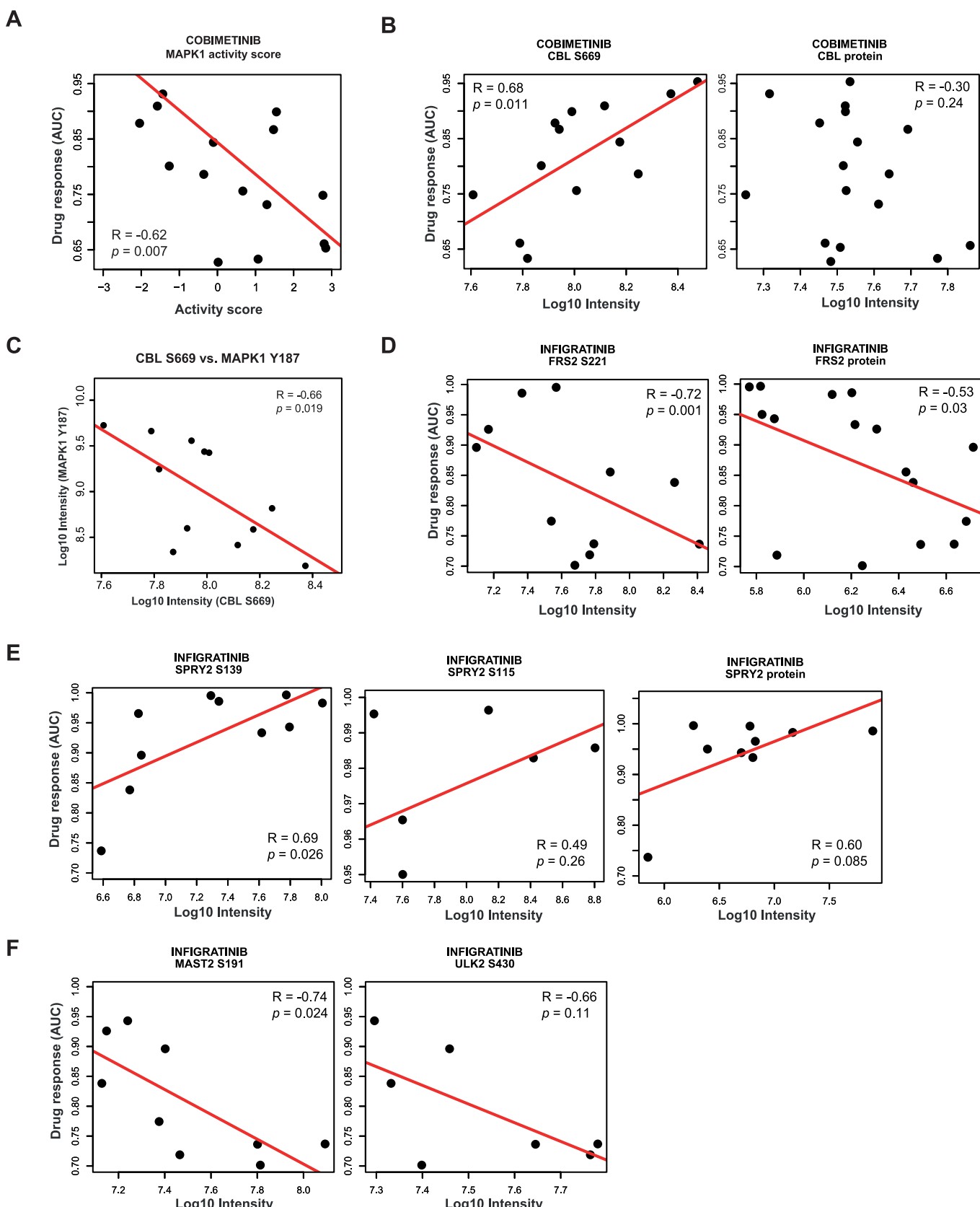

◀ **Figure EV5. Examples for correlation analysis of phenotypic drug sensitivity and protein or p-site abundance from candidates identified by elastic net regression.**

(**A**) Correlation of MAPK1 activity score and drug responses across sarcoma lines. (**B–F**) Correlation of protein and p-site abundance vs. drug responses across sarcoma lines (**B**, **D–F**). Correlation of the abundance of CBL S669 and MAPK1 Y187 across sarcoma lines (**C**).

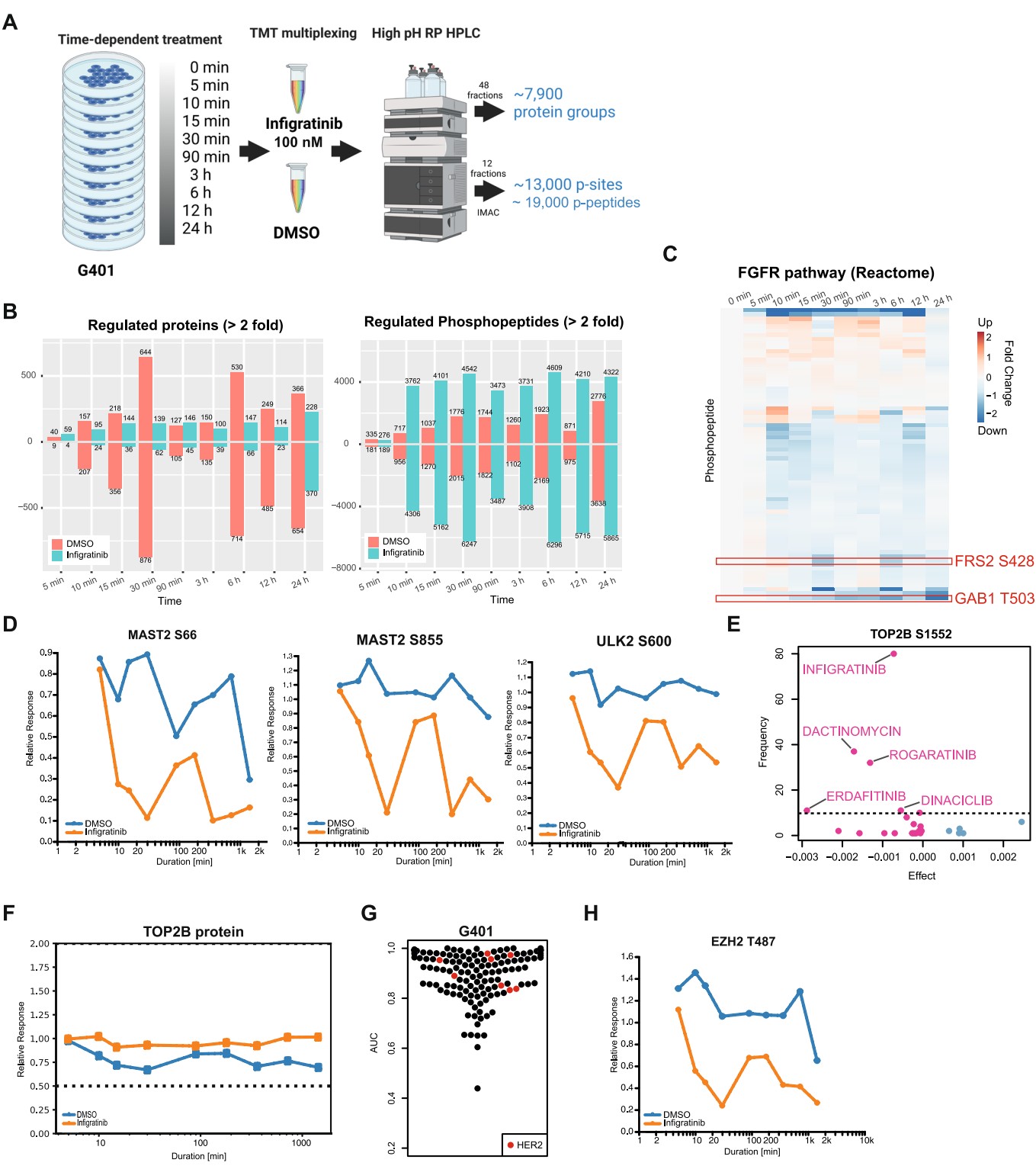

◀ **Figure EV6. Time-resolved (phospho)proteomic analysis of G401 cells in response to Infigratinib.**

(A) Experimental workflow and number of protein and phosphopeptide identifications/quantification. For each time point of drug treatment, a DMSO control was included to account for abundance changes that are not due to the drug treatment. (B) Number of drug-induced protein expression changes (left panel) and phosphopeptide abundance (at least twofold compared to zero min at different time points). (C) Heatmap of fold-changes of phosphopeptides from proteins involved in FGFR signalling (annotations from Reactome) compared to DMSO at different time points after Infigratinib treatment. (D) Time course of p-sites of MAST2 and ULK2 proteins following Infigratinib treatment. (E) Elastic net regression analysis using the phosphorylation site S1552 of TOP2B to identify drugs associated with this p-site. (F) TOP2B protein level at different time points after Infigratinib treatment. (G) Distribution of AUC values from the phenotypic drug screen in G401 cells (139 kinase inhibitors shown). Inhibitors targeting HER2 are highlighted in red. (H) Time course of EZH2 T487 following Infigratinib treatment.

