## [Peer Review File · Molecular Systems Biology]

Illuminating phenotypic drug responses of sarcoma cells to kinase inhibitors by phosphoproteomics

Chien-Yun Lee, Matthew The, Chen Meng, Florian Bayer, Kerstin Putzker, Julian Müller, Johanna Streubel, Julia Woortman, Amirhossein Sakhteman, Moritz Resch, Annika Schneider, Stephanie Wilhelm, and Bernhard Küster

DOI: [10.15252/msb.202211520](https://doi.org/10.15252/msb.202211520)

Corresponding author(s): Bernhard Küster (kuster@tum.de)

Review Timeline:

Submission Date:	23rd Dec 22
Editorial Decision:	8th Feb 23
Revision Received:	29th Mar 23
Editorial Decision:	16th May 23
Revision Received:	6th Nov 23
Accepted:	30th Nov 23

Editor: Maria Polychronidou

Transaction Report:

8th Feb 2023

Manuscript Number: MSB-2022-11520

Title: Illuminating phenotypic drug responses of sarcoma cells to kinase inhibitors by phosphoproteomics

Dear Bernhard,

Thank you again for submitting your work to Molecular Systems Biology. We have now heard back from the three reviewers who agreed to evaluate your study. As you will see below, the reviewers think that the study presents a relevant resource. However, they raise a series of concerns, which we would ask you to address in a revision.

The recommendations of the reviewers are quite clear and I therefore see no need to repeat any of them here. All issues raised by the referees would need to be satisfactorily addressed. Please let me know in case you would like to discuss in further detail any of the issues raised, I would be happy to schedule a call.

On a more editorial level, we would ask you to address the following points:

- Please provide a .doc version of the manuscript text (including legends for main figures and EV Figures) and individual production quality figure files for the main Figures and EV Figures (one file per figure).

- The Expanded View (EV format) includes a limited number of EV Figures (typically ~5). In this case (unless during revision the number of additional figures becomes > 6) all additional figures can be displayed as EV Figures. EV Figures should be provided as individual files and their legends should be included in the main text. For detailed instructions regarding expanded view please refer to our Author Guidelines:

If more EV figures are added during revision, we would ask you to combine additional figures in a PDF called Appendix. Appendix figures and Tables should be labeled and called out as: "Appendix Figure S1, Appendix Figure S2... Appendix Table S1..." etc. Each legend should be below the corresponding Figure/Table in the Appendix. Please include a Table of Contents in the beginning of the Appendix.

- Please provide a "standfirst text" summarizing the study in one or two sentences (approximately 250 characters), three to four "bullet points" highlighting the main findings and a "synopsis image" (550px width and max 400px height, jpeg format) to highlight the paper on our homepage.

- All Materials and Methods need to be described in the main text. We would encourage you to use 'Structured Methods', our new Materials and Methods format. According to this format, the Material and Methods section should include a Reagents and Tools Table (listing key reagents, experimental models, software and relevant equipment and including their sources and relevant identifiers) followed by a Methods and Protocols section in which we encourage the authors to describe their methods using a step-by-step protocol format with bullet points, to facilitate the adoption of the methodologies across labs. More information on how to adhere to this format as well as downloadable templates (.doc or .xls) for the Reagents and Tools Table can be found in our author guidelines: . An example of a Method paper with Structured Methods can be found here:

- Please include a Data availability section describing how the data and code have been made available. This section needs to be formatted according to the example below:

The datasets and computer code produced in this study are available in the following databases:

- Chip-Seq data: Gene Expression Omnibus GSE46748 (<https://www.ncbi.nlm.nih.gov/geo/query/acc.cgi?acc=GSE46748>)

- Modeling computer scripts: GitHub (<https://github.com/SysBioChalmers/GECKO/releases/tag/v1.0>)

- [data type]: [full name of the resource] [accession number/identifier] ([doi or URL or identifiers.org/DATABASE:ACCESSION])

- For data quantification: please specify the name of the statistical test used to generate error bars and P values, the number (n) of independent experiments (specify technical or biological replicates) underlying each data point and the test used to calculate p-values in each figure legend. The figure legends should contain a basic description of n, P and the test applied. Graphs must include a description of the bars and the error bars (s.d., s.e.m.).

- Molecular Systems Biology supports formal data citations in the Reference list, to cite previously published datasets. In addition to citing the original papers that reported the data, we encourage you to also cite the relevant datasets directly in the Reference list. In the text, references to datasets are included as "Data ref: Smith et al, 2001" or "Data ref: NCBI Sequence Read Archive PRJNA342805, 2017". In the Reference list, data citations are very similar to normal literature references but must be labeled with "[DATASET]" at the end of the reference. For detailed instructions please refer to our Author Guidelines .

- Please include a "Disclosure & Competing Interests Statement" in the main text.

- When you resubmit your manuscript, please download our CHECKLIST (<https://bit.ly/EMBOPressAuthorChecklist>) and include the completed form in your submission.

Please note that the Author Checklist will be published alongside the paper as part of the transparent process (<https://www.embopress.org/page/journal/17444292/authorguide#transparentprocess>).

If you feel you can satisfactorily deal with these points and those listed by the referees, you may wish to submit a revised version of your manuscript. Please attach a covering letter giving details of the way in which you have handled each of the points raised by the referees. A revised manuscript will be once again subject to review and you probably understand that we can give you no guarantee at this stage that the eventual outcome will be favorable.

Kind regards,

Maria

Maria Polychronidou, PhD
Senior Editor
Molecular Systems Biology

We realize that it is difficult to revise to a specific deadline. In the interest of protecting the conceptual advance provided by the work, we recommend a revision within 3 months (9th May 2023). Please discuss the revision progress ahead of this time with the editor if you require more time to complete the revisions. Use the link below to submit your revision:

IMPORTANT: When you send your revision, we will require the following items:

1. the manuscript text in LaTeX, RTF or MS Word format
 2. a letter with a detailed description of the changes made in response to the referees. Please specify clearly the exact places in the text (pages and paragraphs) where each change has been made in response to each specific comment given
 3. three to four 'bullet points' highlighting the main findings of your study
 4. a short 'blurb' text summarizing in two sentences the study (max. 250 characters)
 5. a 'thumbnail image' (550px width and max 400px height, Illustrator, PowerPoint or jpeg format), which can be used as 'visual title' for the synopsis section of your paper.
 6. Please include an author contributions statement after the Acknowledgements section (see <https://www.embopress.org/page/journal/17444292/authorguide>)
 7. Please complete the CHECKLIST available at (<https://bit.ly/EMBOPressAuthorChecklist>).
- Please note that the Author Checklist will be published alongside the paper as part of the transparent process (<https://www.embopress.org/page/journal/17444292/authorguide#transparentprocess>).
8. When assembling figures, please refer to our figure preparation guideline in order to ensure proper formatting and readability in print as well as on screen:

See also figure legend guidelines: <https://www.embopress.org/page/journal/17444292/authorguide#figureformat>

9. Please note that corresponding authors are required to supply an ORCID ID for their name upon submission of a revised manuscript (EMBO Press signed a joint statement to encourage ORCID adoption). (<https://www.embopress.org/page/journal/17444292/authorguide#editorialprocess>)
Currently, our records indicate that the ORCID for your account is 0000-0002-9094-1677.

Link Not Available

The system will prompt you to fill in your funding and payment information. This will allow Wiley to send you a quote for the article processing charge (APC) in case of acceptance. This quote takes into account any reduction or fee waivers that you may be eligible for. Authors do not need to pay any fees before their manuscript is accepted and transferred to the publisher.

EMBO Press participates in many Publish and Read agreements that allow authors to publish Open Access with reduced/no publication charges. Check your eligibility: <https://authorservices.wiley.com/author-resources/Journal-Authors/open-access/affiliation-policies-payments/index.html>

*** PLEASE NOTE *** As part of the EMBO Press transparent editorial process initiative (see our Editorial at <https://dx.doi.org/10.1038/msb.2010.72>), Molecular Systems Biology publishes online a Review Process File with each accepted manuscripts. This file will be published in conjunction with your paper and will include the anonymous referee reports, your point-by-point response and all pertinent correspondence relating to the manuscript. If you do NOT want this File to be published, please inform the editorial office at msb@embo.org within 14 days upon receipt of the present letter.

Reviewer #1:

Summary

In the present study by Lee et al, the authors apply phenotypic drug screening in 17 sarcoma cell lines in combination with state-of-the-art deep proteomics and phospho-proteomics of treatment naïve samples to investigate drug mechanisms of action and response markers. The study focuses on human sarcoma cell lines as this disease is largely understudied and potent inhibitors currently lacking. Throughout the manuscript, the authors describe several phenotype-proteotype associations emphasizing the quality of their approach and data to better understand cellular drug responses. One drug example (Infitagrini) is finally selected for a time course experiment following drug treatment to validate the elastic net regression predictions. The data is made publicly available, which further strengthens the work's resource character.

General remarks

The manuscript is well written and easy to follow. Figures are clearly presented. The authors describe several examples of their phenotype-proteotype associations, which, in most parts, convince the reader that the data is of high quality. Conceptually, the combination of drug response screening with global (phospho)proteome profiling is not novel and similar work was done before in different cell line models by the same group and others. The authors cite previous literature accordingly (Frejino et al, 2020; Frejino et al, 2017; Gholami et al, 2013; Lawrence et al, 2015). The clear focus on one understudied and highly heterogeneous cancer entity, namely sarcoma, is the main novelty of the study, but better integration with exciting datasets needed. After addressing the points described below, I believe the study will be relevant for the cancer community and more specifically the sarcoma research field.

Major points

1. To convince the reader that the presented phenotypic and molecular data is of highest quality, the authors should also compare their data to publicly available data for some of the 17 sarcoma cell lines. For example, do the drug response profiles agree with data from Teicher et al, 2015? A look into the DepMap database revealed that most of the 17 cell lines are included in this database. A comparison of these data to the data presented in Fig. 2 and/or Fig 3A of the manuscript would be very informative.
2. How robust or variable are the proteome and p-proteome read-outs? Do the proteome and p-proteome data represent three individual time points of cell collection or only replicates from the same cell culture dish? In other words, would the elastic net regression prediction result in different drug-proteotype associations when cells are harvested on different days?
3. Can the authors use their treatment naïve (phospho)proteome data to assess differences in cell cycle progression, which could help to explain some of the phenotypic drug responses? For example, does the response to the PLK1 inhibitor coincide with a higher mitotic activity? Maybe the kinase activation (INKA) scoring approach could be used in a similar fashion in a similar fashion to also infer cell cycle states?

Minor points

1. A summary of the known actionable genomic alterations for the 17 cell lines would be useful for the reader to select appropriate cell lines for future research. At a minimum, a summarizing table should be included or table EV1 expanded.
2. Similar to EV1D, CVs for proteome and p-proteome data would be important instead of showing single scatterplot examples in Figs. EV1C and EV1D.
3. EV6D, are error bars included here? If not, including them will be important to assess the variability of the measured phospho-sites. Are these changes significant? Same for Fig 5D and E.
4. The authors use the Phospho (STY)Sites.txt table from MaxQuant. Was the phospho-site data further filtered by localization probability? I do not see any information on this and clarification would be important.
5. Page 5, line 9, typo: 're' instead of 'are'
6. Availability of raw files (PRIDE) is still pending.

Reviewer #2:

Review of Lee et al., "Illuminating phenotypic responses of sarcoma cells to kinase inhibitors by phosphoproteomics"

In this manuscript the authors performed phosphoproteomic analysis on 17 sarcoma cell lines and evaluated their responses to 150 different cancer drugs. They then use this information, together with proteomic and phosphoproteomic analysis, to nominate protein phosphorylation sites, and associated kinase pathways, whose activity appears to correlate with sensitivity to certain classes of protein kinase inhibitors, and which might serve as biomarkers for sensitivity of the cell lines to specific drugs. As a last application, they then used an elastic regression approach to identify a correlation between FGFR inhibition and ERBB2 and Top2B phosphorylation, although the molecular basis for this pathway cross-talk is not further explored.

In general, the methods, analysis and conclusions are sound and, to the authors' credit, are interpreted with appropriate caution. The greater utility of the work, however, is probably the drug sensitivity data and the proteomic/phosphoproteomic datasets, which will undoubtedly be widely useful to the community of scientists interested in protein kinase signaling and cancer therapeutics. It seems likely that once a larger phosphoproteomic dataset of bona-fide human sarcoma tumor samples become available for analysis, the authors data will be even more useful.

I have a number of major and minor criticisms that when addressed, will substantially improve the paper.

Specific Comments:

1. Please show/label the staurosporine column in Figure 1A
2. Figure 2 caption - please clarify what % of viability loss was required for a drug to be considered as a relevant kinase inhibitor at concentrations < 100 nM
3. Page 6 - line 24 - I assume the authors mean that the heterogeneity of response was observed for MAP2K inhibitors in most of these cell lines. Please clarify and revise the text to indicate this.
4. I am a bit confused about some of the missing data in Table EV6. For example, the Kinase list includes MAPKAPK2, MAPKAPK3 and MAPKAPK5. However, the kinase abundance table only has MAPKAPK3 and MAPKAPK5, whereas the kinase phosphorylation data table only has MAPKAPK2 and MAPKAPK5, and the Kinase active loop table only has MAPKAPK5. Similar issues exist for other kinases like the Aurora kinase family members. The activity score calculation, according to the details on page 21, seems to involve a summation of kinase abundance, kinase phosphorylation and kinase activation loop phosphorylation. How is this dealt with in these cases of missing data? Is the data shown in Figures 3B, D, and E for all of the kinases, or only for those in which all of the elements of the activity score are present? More details of the activity score calculation would be very helpful.
5. Figure 3D - the authors should test MET inhibitors and c-KIT inhibitors in the SW684 cell line, in addition to the PI3K and MEK inhibitors to show the general utility of their method. Can they also test a few of the kinases that had very low activity scores in this same cell line as a control for specificity of their analysis method?
6. Similarly, the authors should test Volasertib, the Plk1 inhibitor, in their KHOS-NP cells.
7. Page 8 - lines 26-29: Can the authors comments on the sensitivity or specificity with which their kinase activity measurements predict kinase inhibition effects? I recognize that this may be challenging since not all kinases would be expected to impact viability, but perhaps it could be limited to those with clear growth-promoting or cell cycle regulatory effects?
8. Page 9 -lines 25-30. The connection between Lamin A phosphorylation and mTor inhibitor resistance is entirely speculative and not supported by any experimental data. In fact, the T394 site has not been reported to be phosphorylated in interphase cells (Torvaldsen et al., Nucleus. 2015; 6(3): 166-171) and is most likely a mitotic spindle assembly checkpoint site. The authors should see whether phosphorylation of this site correlates with sensitivity to anti-microtubule drugs like Eribulin, and/or should comment on whether this site correlates with other indicators of mitotic kinase or spindle checkpoint activation. An increase in cells in G2/M likely explains the lack of sensitivity to mTor inhibitors. At the least the text of this paragraph needs to be revised.
9. Page 10 - lines 11-14 - can the authors demonstrate a correlation on a per-cell basis between Cbl phosphorylation at S669 and the activity of MAPK (i.e. T185 and Y187 phosphorylation, or with any known MAPK substrate phosphorylation sites to support the claim that this site on Cbl is functionally indicative of inhibition of the MAPK signaling pathway?
10. Page 12 - lines 17-25. The TOP2B S1552 site is a pretty good match for CK2. The authors could consider seeing if any of the inhibitors that tried with CK2 inhibition activity affected the phosphorylation at this site.
11. A related study was published by Eric Haura's group in 2012 (Bai et al. Cancer Research 2012, already cited by the authors), albeit focused on tyrosine kinases. That work, which includes several of the same cell lines as the ones profiled in the current work (A204, HT1080, RD-ES) needs to be more thoroughly discussed in the current manuscript, since that paper also identified Met signaling as being important in a subset of the tumors, as well as a dependence on INSR/IGF1R in RD-ES cells and Src signaling in Saos2 cells. The authors need to significantly expand their discussion taking this prior data into account and compare their results with what was published in the Bai et al paper.

Reviewer #3:

Lee et al.: In this work, the authors report on proteomic, phosphoproteomic, and phenotypic drug profiling to systematically investigate how the cellular proteotypes of sarcoma cells shape their response to specific drugs, particularly Kinase inhibitors (KIs). The results of the proteomes and phosphoproteomes of 17 sarcoma cell lines are screened against 150 cancer drugs to reveal distinct drug responses and the cellular activity landscapes. The positive aspect of this paper is that, in principle, two different layers of proteomic information - full proteome and phosphoproteome - were integrated with phenotypic drug profiles, creating a potentially exciting database for the community. The bioinformatics tools employed for analyzing the omics data resulting in the establishment of correlations of, e.g., kinase abundance, kinase phosphorylation, active loop status, and kinase substrate phosphorylation, represent a major advancement. However, drawbacks, especially in offering this dataset set as a resource, lie in the design and execution of the proteomics analyses. Furthermore, to demonstrate the clinical relevance of the approach, it would be important to validate key observations in primary cells since it remains debatable how closely cell lines that have been propagated for years in tissue culture still represent the primary tumor. Overall, none of the biological observations presented were convincingly confirmed, yet it is claimed that the integration of such data may identify molecular markers of drug response.

Major Concerns:

1. The work is based on screening commercially available sarcoma cell lines, but the rationale for the inclusion criteria of those particular 17 cell lines needs to be better described (Results, page 5). The authors state that the selection of a given cell line is based on the literature and prioritized by "clear differences in drug response" (results page 5, line 7) towards 52 KIs. It is unclear, however, what a "clear difference in drug response" concretely means. Moreover, the authors introduce the 17 different sarcoma cell lines corresponding to 12 diverse histological subtypes. The evident discrepancy is caused by the fact that four histological subtypes are represented by two independent cell lines, which diverge in their response to KIs; no further information is provided. Because cancer is a multifactorial disease, with genetics being a significant contributing factor, presenting the genomic background of those cell lines would facilitate the later interpretation of the proteomic and phosphoproteomic profiles and perhaps even strengthen the importance of considering the protein level in drug screenings for cancer therapy.

2. Similarly, more in-depth information on cell lines would help to understand the inefficiency of a subset of the tested drugs. The authors, for instance, speculate that EGFR or HER2 (page 5, line 32) may not be oncogenic drivers in some of the cell lines but fail to provide any evidence.

3. The drug selection needs further clarification. For example, what is the rationality behind including chemo, epigenetic and non-KI drugs in the study? In addition, what is the overlap between the final selection of drugs/compounds and the drugs/compounds from the previously mentioned Teicher et al., 2015 study?

4. It needs to be clarified why all OMICs data is presented as single-data points ($n=1$). Likewise, discrepancies, especially in the phosphoproteomics workflow, must be addressed better. Additionally, all proteomic samples have been prepared using tandem mass tags (TMT), and MS data have been acquired for MS3-based quantification. It is agreeable that MS1/iBAQ-based quantification, as performed, should be feasible, yet it seems quite an odd choice for this experimental setup.

5. Figure EV1A and Table EV5 show the results from untreated/baseline phospho- proteomes acquired for 17 cell lines. The authors report quantifying 10,200 - 27,200 phosphorylation sites (p-sites) per cell line. Considering that all cell lines are different histological types of sarcomas from a biological standpoint, what could explain such discrepancy in p-sites quantified? Noteworthy, it is essential to consider the technical aspects of phosphoproteomics profiling. The authors report in the materials and methods section (page 19, line 31 following) that peptide fractions were dissolved in 0.1% FA and a half to one-third of the fraction was loaded into an LC-MS system. Different amounts of impute material are critical, mainly when quantification is performed on MS1/iBAQ level. Therefore, the reviewer is concerned that from the start, variability is introduced, making it difficult for further downstream normalization and processing.

6. To better understand the drug responses, the authors integrated phenotypic drug profiles with the proteomes and phosphoproteome of the cell lines at baseline. Thus, drugs targeting specific kinases would be more effective in cell lines driven by the elevated activity of those kinases; activity scores for more than 300 kinases were generated based on this data. However, even though each experiment of the phenotypic drug profiling was performed in triplicates (page 16, line 40), all proteomic/phosphoproteomic data (including the later time-course experiment) are presented as single-data points. In order to strengthen the confidence of the reader in the presented data and draw conclusions, replicates are highly recommended. This is particularly important when depicting substantial variations like for cell line RD-ES in Figure 3B or providing these data as a community resource and encouraging follow-up experiments (page 14, line 32).

7. Using kinase activity scores as a rational basis for choosing an effective drug/compound is intriguing (page 8, line 12 following). However, the presented examples and rationales are not straightforward. For example (page 8, line 20 following),

what led the authors to choose inhibitors for KHOS-NP cells based on the top 10 list of most active kinases? Would an AUC cut-off not present a far more rational criterion? Further, saying that the two chosen inhibitors (CYC-116 and PF-3748309) are among the most potent inhibitors for this cell line is correct, yet slightly covering up the fact that still very high concentrations (1.5 and 4.5 μM , respectively) of these drugs are needed.

Point to point response

The authors are grateful to the reviewers for their insightful comments, which have made the manuscript much stronger, and the point-to-point response below details how we addressed them. The most substantial changes are as follows:

- New experimental data: The reproducibility of the proteomic and phosphoproteomic data was addressed by adding further baseline TMT data sets.
- New data analysis: The consistency of our proteomic data was evaluated by comparison to publically available data from the cancer cell line encyclopaedia (CCLE) consortium.
- New data analysis: The consistency of the phenotypic screening data was evaluated by comparison to a published data set with overlapping sarcoma cell lines.
- Additional clarification: The rationale for selecting cell lines and drugs is explained in more detail in the main text and method section.
- Major re-write of the discussion section including comparison to a published study on the tyrosine phosphorylated proteome of overlapping sarcoma cell lines and a discussion on the limitation of the kinase activity scoring method for the interpretation of particular drug phenotypes.
- Data availability: all data is publicly available at the ProteomeXchange partner site MassIVE (identifier PXD039363).

We hope all concerns have been adequately addressed and that the manuscript is now fit for publication.

Referees' comments:

Reviewer #1:

Summary

In the present study by Lee et al, the authors apply phenotypic drug screening in 17 sarcoma cell lines in combination with state-of-the-art deep proteomics and phospho-proteomics of treatment naïve samples to investigate drug mechanisms of action and response markers. The study focuses on human sarcoma cell lines as this disease is largely understudied and potent inhibitors currently lacking. Throughout the manuscript, the authors describe several phenotype-proteotype associations emphasising the quality of their approach and data to better understand cellular drug responses. One drug example (Infitagrinib) is finally selected for a time course experiment following drug treatment to validate the elastic net regression predictions. The data is made publicly available, which further strengthens the work's resource character.

General remarks

The manuscript is well written and easy to follow. Figures are clearly presented. The authors

describe several examples of their phenotype-proteotype associations, which, in most parts, convince the reader that the data is of high quality. Conceptually, the combination of drug response screening with global (phospho)proteome profiling is not novel and similar work was done before in different cell line models by the same group and others. The authors cite previous literature accordingly (Frejno et al, 2020; Frejno et al, 2017; Gholami et al, 2013; Lawrence et al, 2015). The clear focus on one understudied and highly heterogenous cancer entity, namely sarcoma, is the main novelty of the study, but better integration with exciting data sets needed. After addressing the points described below, I believe the study will be relevant for the cancer community and more specifically the sarcoma research field.

The authors are happy to read that this reviewer thinks that the work is relevant for the community.

Major points

1. To convince the reader that the presented phenotypic and molecular data is of highest quality, the authors should also compare their data to publicly available data for some of the 17 sarcoma cell lines. For example, do the drug response profiles agree with data from Teicher et al, 2015? A look into the DepMap database revealed that most of the 17 cell lines are included in this database. A comparison of these data to the data presented in Fig. 2 and/or Fig 3A of the manuscript would be very informative.

As suggested, we have compared our drug screen results with those of Teicher et al. (2015). The overlap between the two studies is 16 cell lines and 55 compounds. We found that EC50 values were quite well correlated (Appendix Figure S1; one example shown below) given the many differences in experimental detail between the two studies. In our experience, differences in e. g. drug dose range and curve fitting parameters can easily result in differences of EC50 values of a factor 2-3. This is why we focussed our analysis and all examples presented in the manuscript on cases with strong effects only.

In addition, we compared the five overlapping cell lines from the CCLE proteomics study (part of DepMap) (Nusinow et al, 2020) to our work. Again, although parameters such as cell passage number, sample preparation and quantification methods (TMT reporter ion intensity for the CCLE study vs. MS1 intensity in our work) are quite distinct, the quantities of the proteins identified in both studies are well correlated (Appendix Figure S2; two example cell lines shown below). We added text to page 5 line 25-32 of the manuscript to point readers to these comparisons.

2. How robust or variable are the proteome and p-proteome readouts? Do the proteome and p-proteome data represent three individual time points of cell collection or only replicates from the same cell culture dish? In other words, would the elastic net regression prediction result in different drug-proteotype associations when cells are harvested on different days?

This point was addressed in two ways. First, we performed proteome and phosphoproteome analysis of two separate (physical and in time) cell culture replicates for two cell lines (RD-ES and SW1353) and using the bottom up proteomic approach outlined in main Figure 1. The data shown in (Appendix Figure S3; and below) demonstrate that the two replicates correlate very well. As is common experience, the full proteomes correlate better than the phosphoproteomes. The elastic net analysis uses data from 17 cell lines as input. Swapping out replicates of two cell lines would not be expected to lead to very different results.

A Full Proteomes

B Phosphoproteomes

Second, as part of the revision, we compared MS1 to TMT-based quantification for proteomes and phosphoproteomes of 16 cell lines (using the same cell culture material; Appendix Figure S4 & S5 respectively). Again, the results show these two datasets agree well with each other (example for one cell line shown below)

A Full Proteomes

From Appendix Figure S4

B Phosphoproteomes

From appendix Figure S5

3. Can the authors use their treatment naïve (phospho)proteome data to assess differences in cell cycle progression, which could help to explain some of the phenotypic drug responses? For example, does the response to the PLK1 inhibitor coincide with a higher mitotic activity? Maybe the kinase activation (INKA) scoring approach could be used in a similar fashion to also infer cell cycle states?

To address the point specifically, we measured the doubling time of the cells in our screen but found no correlation with the phenotypic drug response to the PLK1 inhibitors Rigosertib and Volasertib (see below Panel A and B). There was also no correlation between the phenotypic response and the computed PLK1 activity score (conceptually similar to INKA) (see below Panel C and D). Prior work by the authors have shown that the two drugs also have other targets (Klaeger et al, 2017) which may influence drug response. For the purpose of the current study, we did not synchronise the cells, which may have been better suited to investigate this particular point. In the future, a better way to address this question might be to measure the response of the (phospho)proteome to PLK1 inhibitors directly and in a time-dependent fashion.

Minor points

1. A summary of the known actionable genomic alterations for the 17 cell lines would be useful for the reader to select appropriate cell lines for future research. At a minimum, a summarising table should be included or table EV1 expanded.

We have included the requested information for cell lines also contained in the CCLE study (Barretina et al, 2012) and Project Drive (McDonald et al, 2017) in Table EV1 (expanded sheets). We also highlight kinases on the list.

2. Similar to EV1D, CVs for proteome and p-proteome data would be important instead of showing single scatterplot examples in Figs. EV1C and EV1D.

We have done as requested and the result is shown below (Appendix Figures S3C). There is indeed substantial variation between the replicates. However, we note again that we only highlight protein or p-peptides in this study that have differences between cell lines which are far larger (typically 10x) than the variation between replicates of the same cell line. We added text to page 6 line 3-7 of the manuscript to point readers to these comparisons.

3. EV6D, are error bars included here? If not, including them will be important to assess the variability of the measured phospho-sites. Are these changes significant? Same for Fig 5D and E.

This time-dependent (phospho)proteomics experiment was not performed in replicates, so we cannot provide error bars to assess statistical significance of each time point. However, we note that the time-dimension was multiplexed in a single TMT experiment. Given the high quantitative precision of TMT measurements, we think the trends observed in response to drug along the time axis can be meaningfully measured and interpreted.

4. The authors use the Phospho (STY)Sites.txt table from MaxQuant. Was the phospho-site data further filtered by localisation probability? I do not see any information on this and clarification would be important.

We did not filter the phosphoproteomics data by localisation probability because we think that the detection of a phosphorylation event on a particular peptide is valuable per se in the context of the interpretation of drug correlations. Our approach generally is to filter for such information late (if at all) in the process. Instead, we prefer checking individual sites in case this site is of particular interest. We acknowledge that there are differences in opinion on this point in the field. For clarity, we now specifically mention this point in the method section (page 22, line 19-22). In addition, the localisation probability of each p-site is provided along with the elastic net regression and correlation plots in ProteomicsDB for transparency (<https://www.proteomicsdb.org/sarquarium>) (see example below, red arrow). We hope this implementation will be useful for users of ProteomicsDB.

5. Page 5, line 9, typo: 're' instead of 'are'

Apologies, this typo has been corrected (page 5, line 9).

6. Availability of raw files (PRIDE) is still pending.

The authors apologise for the delay in supplying the accession number. We are experiencing issues with data submissions to the PRIDE repository. Hence, we have made our data available on ProteomeXchange via the MassIVE partner site where it can be found using the "Data set Identifier: PXD039363". This accession number has been added to the revised manuscript ("Data availability" section, page 25, line 19).

Reviewer #2:

Review of Lee et al., “Illuminating phenotypic responses of sarcoma cells to kinase inhibitors by phosphoproteomics”

In this manuscript the authors performed phosphoproteomic analysis on 17 sarcoma cell lines and evaluated their responses to 150 different cancer drugs. They then use this information, together with proteomic and phosphoproteomic analysis, to nominate protein phosphorylation sites, and associated kinase pathways, whose activity appears to correlate with sensitivity to certain classes of protein kinase inhibitors, and which might serve as biomarkers for sensitivity of the cell lines to specific drugs. As a last application, they then used an elastic regression approach to identify a correlation between FGFR inhibition and ERBB2 and Top2B phosphorylation, although the molecular basis for this pathway cross-talk is not further explored.

In general, the methods, analysis and conclusions are sound and, to the authors' credit, are interpreted with appropriate caution. The greater utility of the work, however, is probably the drug sensitivity data and the proteomic/phosphoproteomic data sets, which will undoubtedly be widely useful to the community of scientists interested in protein kinase signaling and cancer therapeutics. It seems likely that once a larger phosphoproteomic data set of bona-fide human sarcoma tumor samples become available for analysis, the authors data will be even more useful.

We are grateful to read that the reviewer sees the value of this work. We note that, particularly the last point is precisely why we have undertaken the study in the first place. We are involved in a larger sarcoma cancer patient study for which we intend to use the data and tools developed in the current manuscript. We are pointing this out in the discussion of the revised manuscript (page 16, line 26- page 17, line 2).

I have a number of major and minor criticisms that when addressed, will substantially improve the paper.

Specific Comments:

1. Please show/label the staurosporine column in Figure 1A.

We have added staurosporine (used as positive control in the drug screen) to Figure 2A, the underlying viability data to the appendix (Appendix Table S2) and modified the methods section (page 18, line 22-23). Before choosing this concentration, we performed triplicate viability assays to ensure the drug concentration applied was sufficient to kill most cells.

2. Figure 2 caption - please clarify what % of viability loss was required for a drug to be considered as a relevant kinase inhibitor at concentrations < 100 nM.

To clarify, kinase inhibitors are only included in Figure 2D if they result in a loss of viability of >50% at ≤ 100 nM drug. We have added this information to the main text (*page 7, line 12*) and the Figure 2 legend (*page 26, line 20*).

3. Page 6 - line 24 - I assume the authors mean that the heterogeneity of response was observed for MAP2K inhibitors in most of these cell lines. Please clarify and revise the text to indicate this.

Correct. We have revised the text accordingly (page 7, line 11).

4. I am a bit confused about some of the missing data in Table EV6. For example, the Kinase list includes MAPKAPK2, MAPKAPK3 and MAPKAPK5. However, the kinase abundance table only has MAPKAPK3 and MAPKAPK5, whereas the kinase phosphorylation data table only has MAPKAPK2 and MAPKAPK5, and the Kinase active loop table only has MAPKAPK5. Similar issues exist for other kinases like the Aurora kinase family members. The activity score calculation, according to the details on page 21, seems to involve a summation of kinase abundance, kinase phosphorylation and kinase activation loop phosphorylation. How is this dealt with in these cases of missing data? Is the data shown in Figures 3B, D, and E for all of the kinases, or only for those in which all of the elements of the activity score are present? More details of the activity score calculation would be very helpful.

The authors apologise for not explaining the active score more clearly and we have expanded the methods section to improve clarity (page 23, line 21-24). To clarify:

a) Regarding the missing data in Table EV6: There are several reasons for why the four layers of information (abundance of the kinase, its phosphorylation, its activation loop and of its substrates) are not consistently available for all kinases. First, if a kinase of overall low abundance, it may not be detectable in the full proteome. Second, the same applies to the other three layers. Third, because we enrich for phosphorylated peptides, it is possible (and actually observed) that a protein is solely identified by a phosphorylated peptide and no unmodified peptide is in the data. Fourth, the protein identification software groups proteins together that are represented by the same peptides (and phosphopeptides; known as protein grouping). Therefore, kinases with high sequence similarity may give rise to tryptic peptides that have identical sequences. Last, one may or may not observe substrates of kinases in a phosphorylated or non-phosphorylated form.

b) score calculation: As a hard requirement, there has to be kinase abundance information from the full proteome data. Otherwise, no score is calculated for this kinase. We then add

abundance of any of the three other layers to the kinase regardless of whether or not all three are available.

c) regarding Figure 3B: related to the paragraph before, only those kinases are included for which at least kinase abundance and one of the other three layers of information is available. For panels D and E, we further filtered the list of kinases for being a target of at least one inhibitor in the literature (Klaeger et al., 2017) (page 8, line 34- page 9, line1).

5. Figure 3D - the authors should test MET inhibitors and c-KIT inhibitors in the SW684 cell line, in addition to the PI3K and MEK inhibitors to show the general utility of their method. Can they also test a few of the kinases that had very low activity scores in this same cell line as a control for specificity of their analysis method?

We are grateful for this suggestion as further analysis highlighted an important point of discussion. Our screening deck actually contained 6 MET and 7 KIT inhibitors (either as designated target or off-target (see table below)). Interestingly, the MET inhibitors were not very active on SW684 cells even though the activity score indicated high MET activity in these cells (new Figure 3D and figure below). This likely means that MET activity is not particularly relevant for the monitored cellular phenotype (here cell viability). A similar argument applies to KIT inhibitors. The two most potent KIT inhibitors in the figure below (Dasatinib and Midostaurin) are very unselective kinase inhibitors. Hence, their phenotypic effect cannot be attributed to the inhibition of KIT. This is confirmed by the fact that Imatinib (an approved drug for the treatment of KIT-positive gastrointestinal stroma tumors) is inactive in the assay. This interpretation is also in line with data shown in Figure 2A from which it is apparent that most kinase inhibitors are inactive in most cell lines (even though many cell lines express the drug target). We have added this important point to the revised manuscript (page 9, line 26-32).

MET inhibitors	KIT inhibitors
Capmatinib	Avapritinib
Crizotinib	Dasatinib
Linsitinib	Dovitinib
Tepotinib	Imatinib
Tivantinib	Masitinib
Volitinib	Midostaurin
	Toceranib

As suggested, we also looked at proteins with low activity scores in the same cell line, notably CDK1 and PKN (Figure EV3E and below). Drugs targeting these proteins (Dabrafenib for which CDK1 is an off-target and Tofacitinib (PKN1)) were not potent in reducing the viability of SW684 as expected from their activity scores. Dinaciclib (a multi CDK inhibitor) only had a small phenotypic effect on SW684 cells, the computed CDK1 activity in SW684 cells was the lowest of all cell lines and SW684 cells were the least Dinaciclib-responsive of all 17 sarcoma lines (Figure EV3E, right panel). We have added this example to the manuscript (page 9, 6-11).

6. Similarly, the authors should test Volasertib, the Plk1 inhibitor, in their KHOS-NP cells.

Volasertib was also in our screening deck so this data was already in the submission, although not highlighted in the manuscript. The drug indeed potently and reproducibly inhibits the viability of KHOS-NP cells (EC_{50} of ~ 100 nM; AUC of 0.64; Table EV3; screenshot from ProteomicsDB below: (<https://www.proteomicsdb.org/sarquarium>)). To highlight this example, Volasertib was added to Figure 3E and to the corresponding text on page 9, line 19-20.

7. Page 8 - lines 26-29: Can the authors comments on the sensitivity or specificity with which their kinase activity measurements predict kinase inhibition effects? I recognise that this may be challenging since not all kinases would be expected to impact viability, but perhaps it could be limited to those with clear growth-promoting or cell cycle regulatory effects?

This is a great question but, unfortunately, it cannot be answered using the study design and data obtained here. This is because we chose to use kinase inhibitors that are either approved drugs or are in phase III clinical evaluation. These molecules only cover a fraction of all kinases expressed in cells. Perhaps as a result, and as mentioned above, most kinase inhibitors do nothing to most cell lines in our phenotypic screen (Figure 2A), because most kinases inhibited by these drugs are not relevant for the phenotypic effect we have measured here (cell viability). In addition, the cell lines in this study were intentionally chosen to be very different from each other to reflect the fact that human sarcomas are highly heterogeneous in terms of histological subtypes. As mentioned further above, such questions may be better addressed by directly measuring the response of the (phospho)proteome to particular drugs of interest. First steps are being taken in the field in this direction, exemplified by a recent study by the Gygi lab in which they recorded the proteomic response of some 800 drugs in a single cell line (Mitchell et al, 2023) and work for the authors laboratory that has gone only a few days ago (Zecha et al, 2023).

8. Page 9 -lines 25-30. The connection between Lamin A phosphorylation and mTor inhibitor resistance is entirely speculative and not supported by any experimental data. In fact, the T394 site has not been reported to be phosphorylated in interphase cells (Torvaldsen et al., Nucleus. 2015; 6(3): 166-171) and is most likely a mitotic spindle assembly checkpoint site. The authors should see whether phosphorylation of this site correlates with sensitivity to anti-microtubule drugs like Eribulin, and/or should comment on whether this site correlates with other indicators of mitotic kinase or spindle checkpoint activation. An increase in cells in G2/M likely explains the lack of sensitivity to mTor inhibitors. At the least the text of this paragraph needs to be revised.

The authors appreciate this insightful comment and have re-focused this paragraph. We performed the analysis as suggested (abundance of LMNA T394 vs sensitivity to Eribulin, KX-391 (a multi-kinases inhibitor also targeting tubulin polymerisation), and Rigosertib (a PLK1 inhibitor)). Unfortunately, the correlations did not achieve statistical significance despite some positive correlation (see figure below).

ERIBULIN MESYLATE: LMNA_T394

□ Impute missing values

Drug: KX2-391 Protein/Site: LMNA_T394

□ Impute missing values

Drug: RIGOSERTIB Protein/Site: LMNA_T394

□ Impute missing values

However, some G2/M marker proteins such as *AURKB*, *TOP2A/B*, *CDCA8* did show a statistically significant association with response to the mTOR inhibitor *Ridaforolimus* (see below and Appendix Figure S6).

And when mining a very recent large-scale *in vitro* kinase-substrate screen by the Yaffe and Cantley laboratories (Johnson et al, 2023), *BUB1*, a mitotic checkpoint serine/threonine-protein kinase, and four other kinases (*EEF2K*, *AAK1*, *MST1*, *PASK*) were identified as possible kinase that phosphorylate LMNA T394. But neither kinase abundance nor activity correlated with LMNA T394 phosphorylation abundance or sensitivity to mTOR inhibitors (see below). Hence, it remains unclear how exactly the observed lack of sensitivity to mTOR inhibitors (such as *Ridaforolimus* shown in Figure 4B) relates to LMNA T394

phosphorylation. We have revised the main text in light of this new data analysis (page 10, line 29- page 11, line 2).

9. Page 10 - lines 11-14 - can the authors demonstrate a correlation on a per-cell basis between Cbl phosphorylation at S669 and the activity of MAPK (i.e. T185 and Y187 phosphorylation, or with any known MAPK substrate phosphorylation sites to support the claim that this site on Cbl is functionally indicative of inhibition of the MAPK signaling pathway?

Thank you for the suggestion. The phosphorylation levels between MAPK Y187 and CBL S669 indeed negatively correlate with statistical significance, and we have added the plot to Figure EV5C and added text on page 12, line 21-22.

10. Page 12 - lines 17-25. The TOP2B S1552 site is a pretty good match for CK2. The authors could consider seeing if any of the inhibitors that tried with CK2 inhibition activity affected the phosphorylation at this site.

We checked the correlation of TOP2B S1552 abundance and the responses to Abemaciclib. It is a designated CDK4/6 inhibitor, but also has very strong off-target affinity (~4 nM) for CK2 (CSNK2A1 & CSNKA2; (Klaeger et al., 2017). As shown in the figure below, there is no statistically significant correlation that would make a strong case for this site to be a CK2 substrate in the cells we investigated. This may not be conclusive because if the cell lines in the analysis do not depend on FGFR signalling (the original observation that led us to TOP2B phosphorylation), one would not expect to see a correlation of CK2 abundance or activity or AUC. The time series data did not help to resolve this because if FGFR signalling is required to activate CK2, then suppressing that very signal using an FGFR inhibitor would a priori result in loss of TOP2B phosphorylation (akin to e. g. EGFR inhibition where the entire pathway goes down if the driving RTK is inhibited). An analysis of the aforementioned kinase-substrate screen (Johnson et al., 2023) also returned several other potential upstream kinase in addition to CK2 (see below). Again, none of them reached statistical significance. Hence, we do not know which kinase phosphorylates TOP2B S1552 and this is stated in the manuscript (page 13, line 22-34).

Site sequence
position: -5 -4 -3 -2 -1 0 1 2 3 4 5 6
sequence: R K A S G S/T E N E G D Y

Show 50 entries

Kinase	Kinase group	log ₂ (score)	site percentile	percentile rank
CDC7	Other	2.354	97.789 %	1
CK2A2	Other	4.653	96.241 %	2
CK2A1	Other	4.302	95.811 %	3
PRPK	Other	0.595	95.448 %	4
RSK4	AGC	2.800	94.636 %	5
PIM3	CAMK	2.316	94.330 %	6
PIM1	CAMK	2.476	93.468 %	7
MAPKAPK2	CAMK	2.482	93.448 %	8
GRIK7	AGC	2.603	92.910 %	9
P7056KB	AGC	2.054	92.623 %	10
NDR2	AGC	1.736	92.348 %	11
CAMK2B	CAMK	2.294	92.225 %	12
RSK2	AGC	2.178	91.750 %	13
AlphaK3	Alpha	-0.496	91.447 %	14
MOS	Other	1.381	91.160 %	15

11. A related study was published by Eric Haura's group in 2012 (Bai et al. Cancer Research 2012, already cited by the authors), albeit focused on tyrosine kinases. That work, which includes several of the same cell lines as the ones profiled in the current work (A204, HT1080, RD-ES) needs to be more thoroughly discussed in the current manuscript, since that paper also identified Met signaling as being important is a subset of the tumors, as well

as a dependence on INSR/IGF1R in RD-ES cells and Src signaling in Saos2 cells. The authors need to significantly expand their discussion taking this prior data into account and compare their results with what was published in the Bai et al paper.

Have done as suggested. Four cell lines in our panel were also investigated by Bai et al. (A204, RD-ES, SK-LMS-1, HT1080), of which A204 and RD-ES cells were identified to be “driven” by tyrosine kinases. Therefore, we compared these two cell lines and now significantly expand the discussion about the consistency and discrepancy of the data sets (page 15, line 27- page 16, line 12). Briefly, the two data sets agree for A204 cells in terms of PDGFR and FGFR signalling. For RD-ES cells, Bai et al state that INSR/IGF1R signalling is important in these cells but these proteins did not score high in our analysis. Unfortunately we had no INSR/IGF1R inhibitors in our assay panel to investigate this further.

Reviewer #3:

Lee et al.: In this work, the authors report on proteomic, phosphoproteomic, and phenotypic drug profiling to systematically investigate how the cellular proteotypes of sarcoma cells shape their response to specific drugs, particularly Kinase inhibitors (KIs). The results of the proteomes and phosphoproteomes of 17 sarcoma cell lines are screened against 150 cancer drugs to reveal distinct drug responses and the cellular activity landscapes. The positive aspect of this paper is that, in principle, two different layers of proteomic information - full proteome and phosphoproteome - were integrated with phenotypic drug profiles, creating a potentially exciting database for the community. The bioinformatics tools employed for analysing the omics data resulting in the establishment of correlations of, e.g., kinase abundance, kinase phosphorylation, active loop status, and kinase substrate phosphorylation, represent a major advancement. However, drawbacks, especially in offering this data set as a resource, lie in the design and execution of the proteomics analyses. Furthermore, to demonstrate the clinical relevance of the approach, it would be important to validate key observations in primary cells since it remains debatable how closely cell lines that have been propagated for years in tissue culture still represent the primary tumor. Overall, none of the biological observations presented were convincingly confirmed, yet it is claimed that the integration of such data may identify molecular markers of drug response.

The authors are happy to read that the data resource and the methods employed are considered useful. We acknowledge that the lack of follow up in translational models limits the scope of the current work and we have added text to the discussion to make that clear. No doubt, future work should include such models and we have begun collaborating with experts groups in the sarcoma field on this. It will still take considerable time before that work will be fit for sharing with the community so we think it is outside the scope of the current work to postpone publication of the current work.

To review briefly the motivation for the current study and why we think the work is already valuable in its current form: the current study is part of a larger research program in which we use proteomic and phosphoproteomic profiling of (hundreds) of sarcoma patients to aid therapeutic recommendation making in molecular tumor boards (unpublished). As sarcomas represent more than 70 different histological subtypes, the proteomes and phosphoproteomes of patients are actually very diverse and most phosphorylation events are functionally not understood. Because it is not feasible to collect systematic phenotypic drug response data from humans for correlating protein expression and phosphorylation levels of tumors, we reasoned that drug screening in a diverse set of 2D cell culture models would be a meaningful first step to learn more about which molecularly targeted drug(s) may

(eventually) be useful for the treatment of sarcoma patients and which phosphorylation markers one may want to monitor to evaluate response. This is also why we only included approved drugs or phase III clinical compounds that may either be used off-label or in a compassionate use setting. We have added this rationale to the revised manuscript in order to provide the correct 'framing' for the current work. In this regard, we are convinced that the current work is informative for future preclinical work aiming at re-purposing existing drugs for treating sarcoma patients and providing candidate biomarkers for evaluating molecular response.

Major Concerns:

1. The work is based on screening commercially available sarcoma cell lines, but the rationale for the inclusion criteria of those particular 17 cell lines needs to be better described (Results, page 5). The authors state that the selection of a given cell line is based on the literature and prioritised by "clear differences in drug response" (results page 5, line 7) towards 52 KIs. It is unclear, however, what a "clear difference in drug response" concretely means. Moreover, the authors introduce the 17 different sarcoma cell lines corresponding to 12 diverse histological subtypes. The evident discrepancy is caused by the fact that four histological subtypes are represented by two independent cell lines, which diverge in their response to KIs; no further information is provided. Because cancer is a multifactorial disease, with genetics being a significant contributing factor, presenting the genomic background of those cell lines would facilitate the later interpretation of the proteomic and phosphoproteomic profiles and perhaps even strengthen the importance of considering the protein level in drug screenings for cancer therapy.

Fair points and we have made the following additions to the manuscript to provide more detail. First, and as mentioned above, for compound selection, we focussed on approved drugs and phase III compounds. We made sure this is adequately covered in the revised manuscript. Second, we included the genetic information (such as driver genes and mutations) provided by (Barretina et al., 2012; McDonald et al., 2017) for the cell lines in Table EV1. Third, we included a more detailed description of the rationale and criteria for cell line selection. The initial list of 28 commercial cell lines is provided in Appendix Table S1. Because an in-house drug screen of all 28 cell lines was not feasible, we correlated drug response data from the literature for all 28 cell lines (now included as Appendix Figure S7) to select those cell lines that showed the strongest differences in drug response. The reasoning was that these cell lines would cover as much as possible of the phenotypic, and presumably molecular, diversity of all cells lines. The corresponding text is on page 5, line 2-20 and the extended description in the Method section on page 17, line 21-38.

2. Similarly, more in-depth information on cell lines would help to understand the inefficiency of a subset of the tested drugs. The authors, for instance, speculate that EGFR or HER2 (page 5, line 32) may not be oncogenic drivers in some of the cell lines but fail to provide any evidence.

See also our comments above. From the information available in Table EV1, none of the cell lines is annotated to be driven by EGFR/HER2 and the proteins are not mutated. We included this point on page 6, line 17-19. We also refer to a point made in the manuscript that pertains to the motivation of the study (page 4, line 8-15). Genomic profiling of sarcomas is often not very informative and this is the reason why adding a proteomic and phosphoproteomic angle to the molecular profiles is particularly attractive.

3. The drug selection needs further clarification. For example, what is the rationality behind including chemo, epigenetic and non-KI drugs in the study? In addition, what is the overlap between the final selection of drugs/compounds and the drugs/compounds from the previously mentioned Teicher et al., 2015 study?

Please also see our comments above regarding the choice of kinase inhibitors. The chemo and epigenetic drugs are included because they are approved therapies for sarcomas. The non-KI drugs were included opportunistically to cover a few further modalities (such as PARP inhibition or Exportin 1), all of which are approved for a number of entities. The overlap with the Teichmann study is 55 drugs (52 KIs, two chemo drugs, and one PARP inhibitor).

4. It needs to be clarified why all OMICs data is presented as single-data points (n=1). Likewise, discrepancies, especially in the phosphoproteomics workflow, must be addressed better. Additionally, all proteomic samples have been prepared using tandem mass tags (TMT), and MS data have been acquired for MS3-based quantification. It is agreeable that MS1/iBAQ-based quantification, as performed, should be feasible, yet it seems quite an odd choice for this experimental setup.

Please see our response to reviewer #1 regarding the n=1 point.

We acknowledge that this set-up seems odd at first glance. This goes back to the aforementioned project on profiling sarcoma patient proteomes and phosphoproteomes. In that clinical setting, we multiplex 8 patient samples and three of the cell lines featuring in the current study for QC purposes into one TMT-11 batch. As we plan to use the cell line work presented here for the annotation of patient (phospho)proteomes, we also used TMT labelling here to maximize (phospho)proteome overlap between cell lines and patients (as the peptides detected from TMT or unlabeled samples are not necessarily the same). For

quantifying the 17 cell line (phospho)proteomes, we did not multiplex the cell lines and did not use the TMT reporter ions but used the MS1 intensities instead because MS1 intensities afford more dynamic range. We have added a note on this to the revised manuscript (page 19, line 35-37).

The only time we performed an MS3-based quantification is in the time-dependent (phospho)proteome profiling experiment in response to FGFR inhibition shown in Figure 5 (and EV6). In this design, we can make sure that all peptides are covered at all time points.

5. Figure EV1A and Table EV5 show the results from untreated/baseline phospho-proteomes acquired for 17 cell lines. The authors report quantifying 10,200 - 27,200 phosphorylation sites (p-sites) per cell line. Considering that all cell lines are different histological types of sarcomas from a biological standpoint, what could explain such discrepancy in p-sites quantified? Noteworthy, it is essential to consider the technical aspects of phosphoproteomics profiling. The authors report in the materials and methods section (page 19, line 31 following) that peptide fractions were dissolved in 0.1% FA and a half to one-third of the fraction was loaded into an LC-MS system. Different amounts of impute material are critical, mainly when quantification is performed on MS1/iBAQ level. Therefore, the reviewer is concerned that from the start, variability is introduced, making it difficult for further downstream normalisation and processing.

To reduce variability, all cell lines were cultured in house under standard conditions and processed via the same sample preparation and analytical workflows. We apologize for the unclear wording regarding how much of the material was loaded onto the LC-MS system. To clarify: for the baseline datasets, one third of a sample was used; for the time-course experiment, half of the sample was used (page 21, line 11-12). Hence, there was no bias in sample loading. The normalization data shown below (added as Appendix Figure S8), shows that there was no major technical bias. Together with the consistent results of replicate analysis for the two cell lines that returned the lowest number of p-sites (see further above), we conclude that the wide range of p-sites detected for the different cell lines (and RD-ES in particular) is rooted in biology and not in the technical variability of the workflow. The kinase activity landscape shown in Figure 3A already indicated that RD-ES cells (the ones with the lowest phosphorylation levels of all cell lines) have the lowest overall kinase activity. They also have the lowest overall kinase abundance levels (see figure below). GO analysis did not provide clear insights into why RD-ES cells behave so different to the others.

A Full Proteomes

B Phosphoproteomes

6. To better understand the drug responses, the authors integrated phenotypic drug profiles with the proteomes and phosphoproteome of the cell lines at baseline. Thus, drugs targeting specific kinases would be more effective in cell lines driven by the elevated activity of those kinases; activity scores for more than 300 kinases were generated based on this data. However, even though each experiment of the phenotypic drug profiling was performed in

triplicates (page 16, line 40), all proteomic/phosphoproteomic data (including the later time-course experiment) are presented as single-data points. In order to strengthen the confidence of the reader in the presented data and draw conclusions, replicates are highly recommended. This is particularly important when depicting substantial variations like for cell line RD-ES in Figure 3B or providing these data as a community resource and encouraging follow-up experiments (page 14, line 32).

Please also see our comments above regarding replicates. On the specific case of RD-ES cells, we had already replicated the data because their phosphorylation levels were so much lower than for the others. The replicate experiment confirmed this to be true. In addition, throughout the manuscript, we only discuss drug-protein or drug-phosphopeptide associations that: a) show large variation in drug response (>0.3 AUC), b) show large differences in abundance (min. 10-fold) and c) reach statistical significance ($p < 0.01$). Hence, it is extremely unlikely that technical variations would lead to such associations. The online tool allows users to evaluate any association to enable them to form their own assessment regarding the risk and opportunity of initiating follow-up experiments. We have revisited the text in order to ensure that readers are not led to over-interpret the results (page 5, line 21- page 6, line 7).

7. Using kinase activity scores as a rational basis for choosing an effective drug/compound is intriguing (page 8, line 12 following). However, the presented examples and rationales are not straightforward. For example (page 8, line 20 following), what led the authors to choose inhibitors for KHOS-NP cells based on the top 10 list of most active kinases? Would an AUC cut-off not present a far more rational criterion? Further, saying that the two chosen inhibitors (CYC-116 and PF-3748309) are among the most potent inhibitors for this cell line is correct, yet slightly covering up the fact that still very high concentrations (1.5 and 4.5 μM , respectively) of these drugs are needed.

Please also see our comment above regarding the overall framing of the current study. As we show in other parts of the manuscript (Figure 4), correlating AUCs is indeed very rational in case drug sensitivity data is available. For patients, however, this would not work because drug AUC data cannot be created. Here, computing scores from the baseline (phospho)proteomes represents an attempt to discover candidate kinases (or signaling pathways) that may be driving the growth of a particular tumor. In turn, these candidates can then be tested experimentally should a drug for the protein in question and a suitable preclinical or PDX/PDO model exist. In a clinical setting, high scoring druggable candidates could be considered as therapeutic options in a molecular tumor board for a particular patient. We have added this aspect to the discussion (page 16, line 26 - page 17, line 2).

While among the best 10 drugs for inhibiting the viability of KHOS-NP cells, the authors acknowledge that CYC-116 and PF-3748309 are not particularly potent. As we discussed further above for the example shown in Figure 3D (MET; and in the revised manuscript along with revised figures), the computed activity scores for particular kinases may or may not be relevant for the phenotypic readout used in the present work (cell viability). Highly active kinases may serve other functions in this cell line. This may actually be the case for some of the high-scoring kinases in Figure 3E because inhibitors of the MAPK pathway are the most potent in this cell line. We have re-phrased this section to make sure readers are not misled to conclude that all candidates from this analysis are causally involved in reducing cell viability (page 9, line 12-32).

References

- Barretina J, Caponigro G, Stransky N, Venkatesan K, Margolin AA, Kim S, Wilson CJ, Lehar J, Kryukov GV, Sonkin D *et al* (2012) The Cancer Cell Line Encyclopedia enables predictive modelling of anticancer drug sensitivity. *Nature* 483: 603-607
- Johnson JL, Yaron TM, Huntsman EM, Kerelsky A, Song J, Regev A, Lin TY, Liberatore K, Cizin DM, Cohen BM *et al* (2023) An atlas of substrate specificities for the human serine/threonine kinome. *Nature* 613: 759-766
- Klaeger S, Heinzlmeir S, Wilhelm M, Polzer H, Vick B, Koenig PA, Reinecke M, Ruprecht B, Petzoldt S, Meng C *et al* (2017) The target landscape of clinical kinase drugs. *Science* 358
- McDonald ER, 3rd, de Weck A, Schlabach MR, Billy E, Mavrakis KJ, Hoffman GR, Belur D, Castelletti D, Frias E, Gampa K *et al* (2017) Project DRIVE: A Compendium of Cancer Dependencies and Synthetic Lethal Relationships Uncovered by Large-Scale, Deep RNAi Screening. *Cell* 170: 577-592 e510
- Mitchell DC, Kuljanin M, Li J, Van Vranken JG, Bulloch N, Schweppe DK, Huttlin EL, Gygi SP (2023) A proteome-wide atlas of drug mechanism of action. *Nat Biotechnol*
- Nusinow DP, Szpyt J, Ghandi M, Rose CM, McDonald ER, 3rd, Kalocsay M, Jane-Valbuena J, Gelfand E, Schweppe DK, Jedrychowski M *et al* (2020) Quantitative Proteomics of the Cancer Cell Line Encyclopedia. *Cell* 180: 387-402 e316
- Zecha J, Bayer FP, Wiechmann S, Woortman J, Berner N, Muller J, Schneider A, Kramer K, Abril-Gil M, Hopf T *et al* (2023) Decrypting drug actions and protein modifications by dose- and time-resolved proteomics. *Science*: eade3925

16th May 2023

Manuscript Number: MSB-2022-11520R

Title: Illuminating phenotypic drug responses of sarcoma cells to kinase inhibitors by phosphoproteomics

Dear Bernhard,

Thank you for sending us your revised manuscript. We have now heard back from the three reviewers who were asked to evaluate your revised study. As you will see below, the reviewers think that the study has improved as a result of the performed revisions. However, reviewers #1 and #3 still raise significant concerns related to the reproducibility of the results and that fact that most conclusions are derived from measurements that have not been replicated (n=1).

During our cross-commenting process, we specifically asked the reviewers to provide further feedback on these issues, as they seem especially important given the resource nature of the study. Their additional comments were the following:

Reviewer #1: "In principle, single point measurements for discovery-driven research could be fine if orthogonal validation experiments are performed. In the present case, the focus is clearly on the resource character of the manuscript, where validations understandably cannot be done for all the different drug-proteotype associations. In my opinion, the authors could have done a better job with more replicate proteome and p-proteome measurements for few (not all) selected cell lines to convince reviewer #3 (and myself) that the results for the predictions and proposed mechanisms of drug action are robust and not strongly influenced by any technical variations. The authors stated 'The elastic net analysis uses data from 17 cell lines as input. Swapping out replicates of two cell lines would not be expected to lead to very different result'. But this is not shown. I think the influence of the variability coming from different cell line collection time points and different days of sample preparation etc. on the elastic net regression outcome should be the key point to be addressed prior to publication."

Reviewer #2: "The issue about reproducibility and replicate experiments is complicated when dealing with many cell lines, due to time and cost constraints. The correlation coefficient of ~0.6 shows that the replicates are not perfect, but perhaps if the authors could do 3 replicates for maybe half a dozen of the cell lines with a single drug or two at a single dose, we would have some idea how to interpret their data. I suspect the data will be a bit noisy, but at least we would have some benchmark number to know how much confidence to put on the findings as a resource."

Considering the comments above, we have decided to give you a chance to address the remaining concerns in an exceptional second round of major revision. We think that it is important to include additional analyses (along the lines described in the comments above) demonstrating the robustness of the results.

On a more editorial level, we would also ask you to address a few minor editorial issues listed below.

- Our data editors have noticed some unclear or missing information in the figure legends, please see the attached .doc file. Please make all requested text changes using the attached file and *keeping the "track changes" mode* so that we can easily access the edits made.

- Please include callouts to Fig. EV1B, Fig. EV1C, Appendix Fig. S3A-C, S8A and S8B in the main text.

- Tables EV1-EV9 should be renamed to "Datasets EV1-EV9" (as they are rather complex). Please make sure that the callouts in the text are updated accordingly.

- The Figure Legends should be moved after the References.

If you feel you can satisfactorily deal with these points and those listed by the referees, you may wish to submit a revised version of your manuscript. Please attach a covering letter giving details of the way in which you have handled each of the points raised by the referees. A revised manuscript will be once again subject to review and you probably understand that we can give you no guarantee at this stage that the eventual outcome will be favorable.

Kind regards,

Maria

Maria Polychronidou, PhD
Senior Editor
Molecular Systems Biology

We realize that it is difficult to revise to a specific deadline. In the interest of protecting the conceptual advance provided by the work, we recommend a revision within 3 months (14th Aug 2023). Please discuss the revision progress ahead of this time with the editor if you require more time to complete the revisions. Use the link below to submit your revision:

IMPORTANT: Please note that corresponding authors are required to supply an ORCID ID for their name upon submission of a revised manuscript (EMBO Press signed a joint statement to encourage ORCID adoption).

(<https://www.embopress.org/page/journal/17444292/authorguide#editorialprocess>)

Currently, our records indicate that the ORCID for your account is 0000-0002-9094-1677.

Link Not Available

The system will prompt you to fill in your funding and payment information. This will allow Wiley to send you a quote for the article processing charge (APC) in case of acceptance. This quote takes into account any reduction or fee waivers that you may be eligible for. Authors do not need to pay any fees before their manuscript is accepted and transferred to the publisher.

EMBO Press participates in many Publish and Read agreements that allow authors to publish Open Access with reduced/no publication charges. Check your eligibility: <https://authorservices.wiley.com/author-resources/Journal-Authors/open-access/affiliation-policies-payments/index.html>

*** PLEASE NOTE *** As part of the EMBO Press transparent editorial process initiative (see our Editorial at <https://dx.doi.org/10.1038/msb.2010.72>), Molecular Systems Biology publishes online a Review Process File with each accepted manuscripts. This file will be published in conjunction with your paper and will include the anonymous referee reports, your point-by-point response and all pertinent correspondence relating to the manuscript. If you do NOT want this File to be published, please inform the editorial office at msb@embo.org within 14 days upon receipt of the present letter.

Reviewer #1:

The authors now include the results from additional analyses and provide more supplementary information, which certainly improved the manuscript and its usability as resource in future studies. I have a few remaining comments regarding these new analyses, which should be addressed prior publication.

Related to major points 1 and 2:

As requested, the authors now provide proteome correlations using publicly available CCLE data and additionally their own replicate proteome measurements for two cell lines. To show the specificity for these comparisons, the authors should include proteome correlations across cell lines (e.g. A204 vs. G401 for Appendix S2). The statement that the cell line proteome data is well correlated across studies (based on a relatively low Pearson R value of around 0.6) is a bit arbitrary. It is expected that proteome correlations should be much lower across different cell lines, is this actually true? Similarly, for Appendix Fig. S3, replicate correlations should also be compared to those obtained across cell lines to see how specific the quantitative read-out really is. For Appendix S2, one could calculate the mean correlations for all cell line matched proteome comparisons versus those obtained across cell lines. Does this reach high statistical significance?

Additionally, can the authors explain why the observed replicate correlation of the RD-ES cell line proteome is only 0.81, much lower than the rather expected correlation of > 0.9 obtained from the SW1353 cell line. For full proteome replicate measurements of untreated cells, 0.81 seems rather on the very low side for state-of-the-art quantitative MS based proteomics.

Related to major point 3:

The authors now provide additional data to correlate cell line doubling times to the drug response data, which does not show any correlation for the PLK1 inhibitors Rigosertib and Volasertib. This is not surprising as no data was used that reflects the relative proportion of cells in G2/M. Instead of using doubling times, point 3 rather related to the proportion of cells in different cell cycle phases as predictor of drug responses. As protein abundance and phosphorylation states of well-known cell cycle regulators and cell cycle regulated proteins (CDKs, DNA replication proteins, mitotic spindle assembly proteins etc.) are all quantitatively measured, the bulk (phospho)proteome data could be used to infer proportions of cells in different cell cycle phases. This would allow to distinguish cell cycle dependent and independent effects. For example, would cells with the highest

(relative) intensities of S-phase related proteins and phospho-peptides (e.g. ATR substrate motif) be most susceptible to Doxorubicin? As bulk proteomes and p-proteomes are exclusively used in this manuscript and no separate cell cycle experiments performed (e.g. flow cytometry), the integration of cell cycle deconvolution approaches would add an important information layer to interpret the results.

Please also see previous literature:

<https://doi.org/10.7554/eLife.27574>

10.1186/s13059-021-02581-y

<https://doi.org/10.1016/j.mcpro.2021.100169>

Reviewer #2:

The authors have done a very nice job addressing all of my comments with new data, analysis of existing data, and/or well thought out and scholarly revisions. This is a very nice piece of work and will be widely useful to the community of protein kinases aficionados and everyone whose work touches on phosphoproteomic analysis for cancer therapeutics.

Reviewer #3:

While I appreciate the efforts the authors made to provide additional information and improve transparency, it seems that the changes in the main text of the manuscript made in response to the reviewers' comments are minor and most of the additional information is hidden in tables in the EV section and Appendix.

For example, in response to the request to better introduce the selection of the 17 cell lines, the authors mention now in the point-by-point response 28 commercially available cell lines while in the revised manuscript it remains at 17 cell lines. The "more detailed description of the rationale and criteria for cell line selection" stated in the response letter cannot be found in the results section of the revised manuscript. Furthermore, the claimed included genetic information is only provided for about eight cell lines in Table EV1. The panels now included in Appendix Figure S7 do not allow to identify the selected cell lines and thus do not provide much additional information on the individual cell lines. It is claimed that the corresponding text addressing the reviewer's comments can be found on page 5 line 2-20 but the text in the revised manuscript is only slightly changed compared to the original manuscript and does not contain new information.

Importantly, major concerns remain regarding the reproducibility of the results, which is the basis for reliably judging drug responses based on proteomics determinations and an implementation in molecular tumor boards as mentioned in the response letter. In particular, as the presented experiments were performed with cell lines that can be cultivated under standardized conditions in unlimited amounts.

As stated previously, a main concern is the use of single data points ($n=1$) to represent the proteome and phosphoproteome data. To address this, the authors explained in the response letter that they performed proteome and phosphoproteome analysis of two separate cell culture replicates ($n=2$) for two cell lines (RD-ES and SW1353) using the workflow outlined in Figure 1. However, in the text it is only vaguely stated that "cells were replicated". Since cell lines are used, at a minimum three biological replicates for each cell line were expected to facilitate appropriate statistical analyses and to confirm reliability of the determinations. The data shown in Appendix Figure S3 demonstrate that the two replicates correlate with an R square of 0.81 (RD-ES) and 0.93 (SW1353) for full proteomes, but only with an R square of 0.60 (RD-ES) and 0.62 (SW1353) for phosphoproteomes. For both, full proteome and phosphoproteome, the authors claim that the data "correlates well". However, a correlation coefficient of 0.6 as obtained for the phosphoproteome data of both cell lines suggests a moderate correlation, which is not strong enough to support confident conclusions about the reproducibility of the data. A moderate correlation can be due to various factors, including random or systematic error, technical variability, or other confounding factors. It may not reflect the true biological variation in the samples. In the case of phosphoproteomics, the concern is particularly significant as the identification and quantification of phosphorylation sites are often challenging due to their low abundance and the dynamic nature of phosphorylation events. Additionally, phosphoproteomics data can be highly variable due to technical variability arising from sample preparation and data acquisition. Thus, to confirm reliability of the workflow for phosphoproteomics, which is e.g. essential to consider an implementation in molecular tumor boards, additional replicates are essential to improve the statistical power and demonstrate reproducibility of the data.

Along the same lines, another major concern was the wide range of phospho-sites detected (10,200 - 27,200 phosphorylation sites (p-sites) per cell line). The authors used kinase activity scores as a rationale basis for choosing an effective drug/compound, assuming that more from a phosphosite means that the respective kinase has been more active. However, without filtering by localization probability, there is a risk of including non-functional phosphorylation sites in the analysis, which could lead to incorrect interpretations of drug correlations. The authors concluded that "the wide range of p-sites detected for the different cell lines (and RD-ES in particular) is rooted in biology and not in the technical variability of the workflow" but the presented data is too weak to draw such conclusions.

Furthermore, in response to questions regarding TMT labeling the authors present a correlation between TMT and MS1 quantification as a measure of the accuracy and reliability of their measurements. An R square of 0.6 between TMT and MS1 quantification is moderate and a higher correlation coefficient would be expected to indicate a stronger linear relationship between the two methods.

Point to point response

The authors thank the reviewers for their additional efforts. In light of the comments and suggestions made, we have made some major and multiple minor changes to the manuscript. The major changes are as follows:

- a) We have generated more proteomic data. Specifically, we now report three independent biological replicate full- and phospho-proteome data sets for four sarcoma cell lines (G401, RD-ES, RD, SW1353). The cells for these replicates were collected at different times throughout the project with many months in between. This added 360 raw LC-MS/MS files (equivalent to 12 days of measurement time) to the data repository.*
- b) Based on the new data, we have assessed the technical reproducibility of the workflow using the replicated data, leading to the addition of new Figure EV1C; new Appendix Figure S3 & new Appendix Figure S4; updated Datasets 4&5).*
- c) We have assessed the robustness of the elastic net regression analysis leading to the addition of a new Appendix Figure 9.*
- d) We have added a new data analysis that associates the sensitivity of certain drugs to a particular phase of the cell cycle leading to the addition of new Figure EV3F and new Datasets EV7.*
- e) We have made multiple changes to the manuscript to reflect the additional data and analysis (Track Changes in docx. file to show the edits).*

With these improvements, the authors hope that the work is now fit for publication.

Referees' comments during the cross-commenting process:

Reviewer #1:

In principle, single point measurements for discovery-driven research could be fine if orthogonal validation experiments are performed. In the present case, the focus is clearly on the resource character of the manuscript, where validations understandably cannot be done for all the different drug-proteotype associations. In my opinion, the authors could have done a better job with more replicate proteome and p-proteome measurements for few (not all) selected cell lines to convince reviewer #3 (and myself) that the results for the predictions and proposed mechanisms of drug action are robust and not strongly influenced by any technical variations.

We have generated additional data as mentioned above (three biological replicates for four cell lines) to assess the reproducibility of the proteomic workflow. More specifically, we calculated the coefficient of variation of proteins and phosphopeptides for these replicates. As can be seen from the figures below, about 90% of all proteins showed variation of below 2-fold and all proteins showed variation below 3-fold. Expectedly, variation in the phosphoproteomic data was higher but still, 60-90% of all phosphopeptides showed CVs of below 2-fold and all below 3-fold.

New Appendix Figure S3B

New Appendix Figure S4B

As a further illustration of the data quality, the cluster analysis below of the 3x4 cell line proteomes (left) grouped the data by biological replicates and separated the 17 cell lines used in the study (right).

New Figure EV1C

New Appendix Figure S3A

Similar results were obtained for clustering the phosphoproteomes (below)

New Figure EV1C

New Appendix Figure S4A

Taken together, the above shows that variance between cell lines is greater than between replicates. In addition, as we discuss in the manuscript and further below, we only highlight proteins or phosphosites in the manuscript that show at least 10x differences in abundance between cell lines. This requirement was chosen to make sure that the examples highlighted in the manuscript are not artifacts owing to technical variation. We did not hard-code the 10x requirement in the online tool to enable users to fully explore the data. But we emphasize in the manuscript that users should be careful when abundance differences between cell lines are close to the determined technical reproducibility of the proteomic workflow.

The authors stated 'The elastic net analysis uses data from 17 cell lines as input. Swapping out replicates of two cell lines would not be expected to lead to very different result'. But this is not shown. I think the influence of the variability coming from different cell line collection time points and different days of sample preparation etc. on the elastic net regression outcome should be the key point to be addressed prior to publication.

We have added this analysis here and to the manuscript (page 26, line 39-41). Illustrated for two drugs, we first compared the results of the elastic net regression analysis in the original manuscript version to a re-run of the analysis using the exact same data of 17 cell line proteomes / phosphoproteomes. Plotting the outcomes (selection frequency) of the two analyses against each other revealed that they are highly correlated but not identical (below and Appendix Figure S9A). This is due to an inherent feature of the elastic net regression algorithm where the random seed, used to initialize the random number generator, affects the cross-validation fold assignments. As a result, different hyperparameters may be selected for the elastic net in each of the runs, impacting the selection frequency.

A Original vs. Rerun

New Appendix Figure S9A

Second, we swapped out the proteomes and phosphoproteomes of the four cell lines with replicates but kept the 13 proteomes and phosphoproteomes of cell lines without replicates. Here, the correlations are much worse but 69% and 58% of the proteins in the original analysis were recapitulated in the second for Cobimetinib and Infigratinib respectively. Moreover, all the examples presented in the manuscript (Figure 5 below and highlighted in

color in figures above and below) were identified in both EN analysis. Given that approximately 12,000 proteins, 53,000 phosphosites and 150 drugs went into the EN analysis, the results were almost surprisingly robust. Still, to make the utility of the EN analysis clearer to readers, the revised manuscript presents the EN analysis as a tool that can be used to prioritize the approx. 9.7 million drug response correlations in the data set. As above, we did not set ad hoc cut-offs drug response AUC, number of cell lines or minimum fold change of protein/phosphopeptide abundance in the online tool so that users can fully explore the data set.

B Original vs. Duplicate

New Appendix Figure S9B

New Appendix Figure S9C

Figure 5A

Figure 5B

As a further illustration for the purpose of this point to point response, the figure below shows two examples of phosphorylation sites comparing the drug response vs abundance correlation when using different replicate proteome/phosphoproteome data sets (i. e. swapping out data for four cell lines; highlighted in blue). It is apparent, that the two results are consistent with each other.

(Fig. 5C)

Replicates

Reviewer #2:

The issue about reproducibility and replicate experiments is complicated when dealing with many cell lines, due to time and cost constraints. The correlation coefficient of ~ 0.6 shows that the replicates are not perfect, but perhaps if the authors could do 3 replicates for maybe half a dozen of the cell lines with a single drug or two at a single dose, we would have some idea how to interpret their data. I suspect the data will be a bit noisy, but at least we would have some benchmark number to know how much confidence to put on the findings as a resource.

Please see our response above. To clarify, we did not measure (phospho) proteomes in response to drug treatment, but treatment-naïve cells.

Referees' comments to first revision

Reviewer #1:

The authors now include the results from additional analyses and provide more supplementary information, which certainly improved the manuscript and its usability as resource in future studies. I have a few remaining comments regarding these new analyses, which should be addressed prior publication.

Related to major points 1 and 2:

As requested, the authors now provide proteome correlations using publicly available CCLE data and additionally their own replicate proteome measurements for two cell lines. To show the specificity for these comparisons, the authors should include proteome correlations across cell lines (e.g. A204 vs. G401 for Appendix S2). The statement that the cell line proteome data is well correlated across studies (based on a relatively low Pearson R value of around 0.6) is a bit arbitrary. It is expected that proteome correlations should be much lower across different cell lines, is this actually true?

Yes, this is broadly the case. Five cell lines are common between our and the CCLE study. Because the CCLE data used TMT quantification, we first correlated MS1 and TMT intensities of the proteomes of these cell lines generated in our study (left panel below). The two quantification methods correlate well and better for the same cell line than across cell lines. We then correlated the TMT intensities from our data with those of the CCLE study (right panel below). The correlations were weaker but also mostly showed that intensities correlate better for the same cell line than across cell lines. An exception is the RD cell line data and the reasons are not clear at present. One might speculate that for certain cell lines grown in different laboratories under different conditions may sometimes lead to more substantial differences in proteomic content than for other cell lines.

New Appendix Figure S2B

Similarly, for Appendix Fig. S3, replicate correlations should also be compared to those obtained across cell lines to see how specific the quantitative read-out really is.

Please see our comments above.

For Appendix S2, one could calculate the mean correlations for all cell line matched proteome comparisons versus those obtained across cell lines. Does this reach high statistical significance?

Yes, it does. We performed a t-test on the correlations between matched cell lines (e.g., A204 in CCLE vs. A204 in our dataset) and unmatched cell lines (e.g., A204 in CCLE vs. G401 in our dataset). To ensure an equal group size for the t-test, we compared the 5 correlations of matched cell lines (A204, G401, HT1080, RD, SK-ES-1) to correlations of five unmatched cell line pairs, selected using a derangement approach, i.e. a particular type of permutation where no cell line can be matched to itself. Setting the random number seed to 1, resulted in the unmatched cell line pairs SK-ES-1+RD, A204+G401, RD+HT1080, G401+SK-ES-1, HT1080+A204 (first cell line in CCLE, second cell line in our dataset). The 5 correlations of these unmatched cell lines were compared to the matched ones using a paired t-test and produced a highly significant p-value of 0.00012. Derangements from different random number seeds resulted in comparably low p-values. Consequently, we conclude that the proteomes of unmatched cell lines are less similar to each other than matched cell lines.

Additionally, can the authors explain why the observed replicate correlation of the RD-ES cell line proteome is only 0.81, much lower than the rather expected correlation of > 0.9 obtained from the SW1353 cell line. For full proteome replicate measurements of untreated cells, 0.81 seems rather on the very low side for state-of-the-art quantitative MS based proteomics.

Please see our comments above. More specifically, using the additional datasets (triplicates of four cell lines), the correlations within the same cell lines are between 0.89-0.92. For the RD-ES cells, the correlations are equally high: 0.89, 0.90, and 0.91.

Related to major point 3:

The authors now provide additional data to correlate cell line doubling times to the drug response data, which does not show any correlation for the PLK1 inhibitors Rigosertib and Volasertib. This is not surprising as no data was used that reflects the relative proportion of cells in G2/M. Instead of using doubling times, point 3 rather related to the proportion of cells in different cell cycle phases as predictor of drug responses. As protein abundance and phosphorylation states of well-known cell cycle regulators and cell cycle regulated proteins (CDKs, DNA replication proteins, mitotic spindle assembly proteins etc.) are all quantitatively measured, the bulk (phospho)proteome data could be used to infer proportions of cells in different cell cycle phases. This would allow to distinguish cell cycle dependent and independent effects. For example, would cells with the highest (relative) intensities of S-phase related proteins and phospho-peptides (e.g. ATR substrate motif) be most susceptible to Doxorubicin? As bulk proteomes and p-proteomes are exclusively used in this manuscript and no separate cell cycle experiments performed (e.g. flow cytometry), the integration of cell cycle deconvolution approaches would add an important information layer to interpret the results.

Please also see previous literature:

<https://doi.org/10.7554/eLife.27574>

10.1186/s13059-021-02581-y

<https://doi.org/10.1016/j.mcpro.2021.100169>

We appreciate this interesting suggestion and added a new paragraph in results (page 10, line 5-21 (docx); line 8-24 (pdf)), Methods (page 25, line 26-42), and a new Figure EV3F, and a new Datasets EV7 to the manuscript that report these results. Briefly, we downloaded the pseudoperiodic protein clusters from Kelly et al. (Kelly et al, 2022). In that study, sixteen cell populations were collected across six annotated stages of the cell cycle. The original Figure 3B from this publication (below) shows five clusters containing a total of 119 “periodic” proteins. It is quite clear that there are no sharp boundaries between the clusters and that Cluster 1 is a mix of G1/S/G2 that partially overlaps with Clusters 2 and 3. This should be kept in mind for the following analysis.

(Figure 3B from Kelly et al. (Kelly et al., 2022))

For each of the 17 cell lines in our study, we inferred the “proportions of cells in different cell cycle phases” from the bulk proteome data as follows:

1. We mapped the identified proteins in our study to the list of pseudoperiodic proteins clusters of Kelly et al. (115 of these 119 proteins were found).
2. We summed up the intensities of all proteins in each Kelly cluster for each cell line.
3. We rescaled each Kelly cluster from 0 to 1 among 17 cell lines. This step is required to remove the bias from clusters that contain more proteins (higher summed intensity).
4. We calculated the proportion of each Kelly cluster in each cell line (value of each cluster compared to the total sum of the five Kelly clusters in one cell line). This would represent an approximation of the proportion of cells in a particular stage of the cell cycle.

Second, we computed the correlation of each Kelly cluster proportion with drug response for each drug. We only chose the 87 drugs in our data that had a minimum AUC < 0.8 to avoid obtaining correlations from drugs with very small effects. The heat map below (and new Figure EV3F) summarizes these correlations (negative correlation in blue; the higher the Kelly cluster proportion, the more sensitive a cell line is to the respective drug; positive correlation in red).

New Figure EV3F; New Datasets EV7

We then focused on negative correlations stronger than $R < -0.5$ (cells that are sensitive to drugs; see table below and Datasets EV7). Interestingly, for 10 out of 14 drugs, the expression of proteins in a cluster could be correlated to the phase of the cell cycle in which the respective drug has been reported to arrest the cells (see below and Datasets EV7). Examples include Dinaciclib, a CDK2,5,9 inhibitor that blocks cells in S phase and is most negatively correlated in Cluster 2. The same was observed for Doxorubicin, an inhibitor of DNA replication that happens in S phase (Cluster 2). The correlation for Dasatinib was strongest in Cluster 4 (early M phase) but given the many targets of this drug, it is difficult to attribute this effect to a particular kinase. In cluster 5, the large number of drugs with positive correlations (resistance) is noteworthy. Eight of these drugs are PI3K/mTOR inhibitors (marked with asterisk) indicating that cells in M-phase, are particularly resistant to inhibition of these pathways.

New Datasets EV7 Tab1 (rank by negative correlation)

Drug	C1 [G1/S/G2]	C2 [S]	C3 [G2]	C4 [Early M]	C5 [M]	Phase (literature)	Phase (our data)	Consistent?
DINACICLIB	0.17	-0.72	-0.63	0.49	0.04	S/G2	S/G2	YES
DACTINOMYCIN	0.20	-0.65	-0.51	0.43	-0.03	G1/S	S/G2	YES
TRABECTEDIN	0.04	-0.65	-0.32	0.63	-0.37	S/G2	S	YES
DOXORUBICIN	0.03	-0.64	-0.56	0.44	0.19	S/G2	S/G2	YES
ERIBULIN MESYLATE	0.04	-0.58	-0.61	0.51	0.03	G2/M	S/G2	YES
KX2-391	0.09	-0.48	-0.58	0.44	-0.01	G2/M	G2	YES
DUVELISIB	-0.19	0.48	0.42	-0.56	0.49	G1/S	Early M	NO

DASATINIB	0.24	0.35	0.46	-0.56	0.08	G1	Early M	NO
SELINEXOR	0.37	-0.40	-0.55	0.25	-0.08	G1/S	G2	NO
ALPELISIB	-0.54	-0.40	-0.04	0.39	0.30	G1	G1/S/G2	YES
TIVANTINIB	-0.02	-0.48	-0.54	0.42	0.10	S/G2	G2	YES
IDELALISIB	-0.19	0.59	0.48	-0.54	0.29	G1	Early M	NO
ALISERTIB	-0.12	-0.46	-0.53	0.41	0.21	G2	G2	YES
TALAZOPARIB	-0.12	-0.49	-0.52	0.39	0.27	G2	G2	YES

New Datasets EV7 Tab1 (rank by positive correlation)

Drug	C1 [G1/S/G2]	C2 [S]	C3 [G2]	C4 [Early M]	C5 [M]	Target(s)
TRABECTEDIN	0.04	-0.65	-0.32	0.63	-0.37	DNA
PACRITINIB	-0.05	-0.07	-0.16	-0.21	0.63	JAK2
RAPAMYCIN*	0.04	-0.14	-0.21	-0.21	0.62	mTOR
TASELISIB*	-0.32	-0.20	0.04	-0.06	0.61	PI3K
RIDAFOROLIMUS*	0.07	-0.06	-0.17	-0.27	0.61	mTOR
AFATINIB	-0.11	-0.06	0.02	-0.23	0.60	ErbB
IDELALISIB*	-0.19	0.59	0.48	-0.54	0.29	PI3K
BUPARLISIB	-0.12	-0.27	-0.26	0.01	0.56	PI3K
NEMIRALISIB*	-0.36	0.49	0.53	-0.47	0.42	PI3K
COPANLISIB*	-0.39	-0.19	0.10	0.00	0.53	PI3K
CERTICAN*	0.09	-0.13	-0.14	-0.22	0.53	PI3K
LINSITINIB	-0.32	-0.41	-0.26	0.52	-0.02	IGF-1R
ERIBULIN MESYLATE	0.04	-0.58	-0.61	0.51	0.03	microtubule
TEMSIROLIMUS*	0.11	-0.34	-0.33	-0.03	0.50	mTOR

We also attempted an analogous analysis for the phosphoproteome data, either using annotated substrate phosphorylation sites for cell cycle related kinases or the 115 early rising M-phase phosphorylation sites from Ly et al. (Ly et al, 2017) but neither of the two approaches led to conclusive results.

Reviewer #2:

The authors have done a very nice job addressing all of my comments with new data, analysis of existing data, and/or well thought out and scholarly revisions. This is a very nice piece of work and will be widely useful to the community of protein kinases aficionados and everyone whose work touches on phosphoproteomic analysis for cancer therapeutics.

Thank you.

Reviewer #3:

While I appreciate the efforts the authors made to provide additional information and improve transparency, it seems that the changes in the main text of the manuscript made in response to the reviewers' comments are minor and most of the additional information is hidden in tables in the EV section and Appendix.

Please see our comments above regarding additional data, data analysis and figures incorporated in the current revision. As a result, we also made more changes to the main text. Because the additional items mostly address technical issues, we placed most of the related text in the appendix to avoid distraction from the main story line of the paper. We hope that this reviewer finds this acceptable.

For example, in response to the request to better introduce the selection of the 17 cell lines, the authors mention now in the point-by-point response 28 commercially available cell lines while in the revised manuscript it remains at 17 cell lines. The "more detailed description of the rationale and criteria for cell line selection" stated in the response letter cannot be found in the results section of the revised manuscript.

The rationale for cell line selection is detailed in the Materials and Methods section (page 19, line 20-37), and we now refer to this explicitly in the results section (page 5 line 10-11).

Furthermore, the claimed included genetic information is only provided for about eight cell lines in Table EV1.

We have extended the search to three further databases, notably Cellosaurus (Bairoch, 2018), DepMap (from new CCLE (Ghandi et al, 2019)), and COSMIC (Tate et al, 2019) so that genetic information is now available for 15 cell lines in extended Datasets EV1 (previous Table EV1). We did not find omics data for KHOS-240S and KHOS-NP.

The panels now included in new Appendix Figure S7 do not allow to identify the selected cell lines and thus do not provide much additional information on the individual cell lines. It is claimed that the corresponding text addressing the reviewer's comments can be found on

page 5 line 2-20 but the text in the revised manuscript is only slightly changed compared to the original manuscript and does not contain new information.

We are surprised by this comment because the selected cell lines are labelled in yellow and they are covered in different clusters. We added one additional paragraph to the Materials and Methods section describing the choice of cell lines (page 19, line 20-37(docx); line 20-30(pdf)). As mentioned above, we now specifically point to this information in the results section (page 5 line 10-11).

Importantly, major concerns remain regarding the reproducibility of the results, which is the basis for reliably judging drug responses based on proteomics determinations and an implementation in molecular tumor boards as mentioned in the response letter. In particular, as the presented experiments were performed with cell lines that can be cultivated under standardized conditions in unlimited amounts. As stated previously, a main concern is the use of single data points (n=1) to represent the proteome and phosphoproteome data. To address this, the authors explained in the response letter that they performed proteome and phosphoproteome analysis of two separate cell culture replicates (n=2) for two cell lines (RD-ES and SW1353) using the workflow outlined in Figure 1. However, in the text it is only vaguely stated that "cells were replicated". Since cell lines are used, at a minimum three biological replicates for each cell line were expected to facilitate appropriate statistical analyses and to confirm reliability of the determinations. The data shown in new Appendix Figure S3 demonstrate that the two replicates correlate with an R square of 0.81 (RD-ES) and 0.93 (SW1353) for full proteomes, but only with an R square of 0.60 (RD-ES) and 0.62 (SW1353) for phosphoproteomes. For both, full proteome and phosphoproteome, the authors claim that the data "correlates well". However, a correlation coefficient of 0.6 as obtained for the phosphoproteome data of both cell lines suggests a moderate correlation, which is not strong enough to support confident conclusions about the reproducibility of the data. A moderate correlation can be due to various factors, including random or systematic error, technical variability, or other confounding factors. It may not reflect the true biological variation in the samples. In the case of phosphoproteomics, the concern is particularly significant as the identification and quantification of phosphorylation sites are often challenging due to their low abundance and the dynamic nature of phosphorylation events. Additionally, phosphoproteomics data can be highly variable due technical variability arising from sample preparation and data acquisition. Thus, to confirm reliability of the workflow for phosphoproteomics, which is e.g. essential to consider an implementation in molecular tumor boards, additional replicates are essential to improve the statistical power and demonstrate reproducibility of the data.

Please see our comments above. Briefly, we have generated additional data for four cell lines (biological triplicates) from which we determined the technical reproducibility of the data and the ability to distinguish technical from biological variance. We now specifically point out that any of the examples shown in the manuscript required abundance differences of at least 10-fold between cell lines (and a minimum of 8 cell lines), which is 5x higher than the median reproducibility of the proteomic data. This should ensure that specific statements made for certain drugs, proteins or phosphorylation sites are not the result of spurious correlations.

Also as mentioned above, we did not impose such cut-offs for the online exploration of the data in ProteomicsDB so that users can fully mine the available data.

Along the same lines, another major concern was the wide range of phospho-sites detected (10,200 - 27,200 phosphorylation sites (p-sites) per cell line). The authors used kinase activity scores as a rational basis for choosing an effective drug/compound, assuming that more from a phosphosite means that the respective kinase has been more active. However, without filtering by localization probability, there is a risk of including non-functional phosphorylation sites in the analysis, which could lead to incorrect interpretations of drug correlations. The authors concluded that "the wide range of p-sites detected for the different cell lines (and RD-ES in particular) is rooted in biology and not in the technical variability of the workflow" but the presented data is too weak to draw such conclusions.

Please also see our comments above regarding technical reproducibility. The same workflow was used for all cell lines and the determined technical reproducibility is too high to explain the wide range of p-sites per cell line. This was confirmed by the replicate analysis of four cell lines including the RD-ES cell line. In published work, we have characterized the phospho-proteomes of 125 cell lines (Frejno et al, 2020) and found such outlier cell lines before, indicating that the phosphoproteomes of cancer cell lines can vary very substantially.

Regarding phosphorylation site localization: the localization probabilities for all phosphosites are in MaxQuant output tables deposited in the massIVE repository. This information is also provided for every phosphosites via the interactive web interface of ProteomicsDB. We included the new Appendix Figure S7 (see below) to show that the vast majority of all phosphorylation sites has very high localization probabilities (median of 0.9954). And more specifically, all phosphorylation sites discussed in the manuscript have localization probabilities ranging from 0.96 to 1.

New Appendix Figure S7

Furthermore, in response to questions regarding TMT labeling the authors present a correlation between TMT and MS1 quantification as a measure of the accuracy and reliability of their measurements. An R square of 0.6 between TMT and MS1 quantification is

moderate and a higher correlation coefficient would be expected to indicate a stronger linear relationship between the two methods.

Please also see our comments above. The availability of the additional data (triplicates of four cell lines) enabled us to use MS1 quantification throughout the manuscript. Because we no longer needed to rely on the MS1 vs TMT comparison, we removed it from the manuscript.

References

- Bairoch A (2018) The Cellosaurus, a Cell-Line Knowledge Resource. *J Biomol Tech* 29: 25-38
- Frejno M, Meng C, Ruprecht B, Oellerich T, Scheich S, Kleigrew K, Drecoll E, Samaras P, Hogrebe A, Helm D *et al* (2020) Proteome activity landscapes of tumor cell lines determine drug responses. *Nat Commun* 11: 3639
- Ghandi M, Huang FW, Jane-Valbuena J, Kryukov GV, Lo CC, McDonald ER, 3rd, Barretina J, Gelfand ET, Bielski CM, Li H *et al* (2019) Next-generation characterization of the Cancer Cell Line Encyclopedia. *Nature* 569: 503-508
- Kelly V, Al-Rawi A, Lewis D, Kustatscher G, Ly T (2022) Low Cell Number Proteomic Analysis Using In-Cell Protease Digests Reveals a Robust Signature for Cell Cycle State Classification. *Mol Cell Proteomics* 21: 100169
- Ly T, Whigham A, Clarke R, Brenes-Murillo AJ, Estes B, Madhessian D, Lundberg E, Wadsworth P, Lamond AI (2017) Proteomic analysis of cell cycle progression in asynchronous cultures, including mitotic subphases, using PRIMMUS. *Elife* 6
- Tate JG, Bamford S, Jubb HC, Sondka Z, Beare DM, Bindal N, Boutselakis H, Cole CG, Creatore C, Dawson E *et al* (2019) COSMIC: the Catalogue Of Somatic Mutations In Cancer. *Nucleic Acids Res* 47: D941-D947

30th Nov 2023

Manuscript number: MSB-2022-11520RR

Title: Illuminating phenotypic drug responses of sarcoma cells to kinase inhibitors by phosphoproteomics

Dear Bernhard,

Thank you again for sending us your revised manuscript. We have now heard back from the two reviewers who were asked to evaluate the study. As you will see below, they are both satisfied with the additional analyses and they support publication. I am therefore pleased to inform you that your study has been accepted for publication.

Kind regards,

Maria

Maria Polychronidou, PhD
Senior Editor
Molecular Systems Biology

Reviewer #1:

The authors have done a great job in addressing my remaining concerns. The new data and explanations have improved the manuscript and are important additions that help the reader to use and interpret the data. I congratulate the authors on their work!

Reviewer #3:

I have thoroughly evaluated the revised version of the manuscript and appreciate the authors' diligent efforts to address the concerns raised during the review process. The changes made to the main text improved the manuscript, and the additional information is more evident. The inclusion of data for four cell lines in biological triplicates is of importance and effectively addresses my major concern about the reproducibility of the results. This addition demonstrates a valuable determination of both technical and biological variance of the approach.
